# Anytime Detection of Strategic Deviations in Multi-Agent Systems

**Etienne Gauthier** [1]   **Francis Bach** [1]   **Michael I. Jordan** [1,2]

## Abstract

In many multi-agent systems, agents interact repeatedly and are expected to settle into stable, rational behavior over time. Yet in practice, behavior often drifts, and detecting such deviations in real time remains an open challenge. We introduce a sequential testing framework that monitors whether observed play is consistent with a benchmark of strategic behavior, without assuming a fixed sample size. Our approach builds on the e-value framework for safe anytime-valid inference: by "betting" against the benchmark, we construct a test supermartingale that accumulates evidence whenever observed payoffs systematically violate the expected conditions. For repeated normal-form games, we take equilibrium as the benchmark, yielding a statistically sound, interpretable measure of departure from equilibrium that can be monitored online; our framework unifies the treatment of Nash, correlated, and coarse correlated equilibria, offering finite-time guarantees and a detailed analysis of detection times. We also leverage Benjamini-Hochberg-type procedures to increase detection power in large games while rigorously controlling the false discovery rate. Finally, we extend our method to stochastic games, verifying online whether observed trajectories adhere to a specified target policy, such as a computed equilibrium, broadening the framework's applicability to dynamic, state-dependent settings.

## 1. Introduction

Game theory provides a powerful lens for understanding the collective behavior of interacting decision-makers, or agents, in shared environments. Each agent selects actions to achieve its own objectives, while the resulting interactions create complex patterns of incentives that shape the system as a whole. Such patterns can be characterized in terms of equilibrium concepts, such as Nash equilibrium (Nash, 1951), correlated equilibrium (Aumann, 1974), or coarse correlated equilibrium (Moulin & Vial, 1978). In all of these cases, no individual agent can improve its expected payoff by unilaterally deviating. These equilibrium concepts apply both to single-shot games and to repeated games, thereby providing a general foundation for analyzing the collective dynamics of strategic behavior.

This foundation is supported by powerful results that link game theory and online learning. For example, if all agents follow no-regret learning dynamics, their joint empirical play converges to an equilibrium set. Specifically, vanishing external regret guarantees convergence to the set of coarse correlated equilibria, while vanishing internal regret yields convergence to the set of correlated equilibria. In special cases, such as two-player zero-sum games, these equilibria coincide with Nash equilibrium. For a thorough discussion of such links between learning and games, we refer the reader to Cesa-Bianchi & Lugosi (2006).

If we take a broader viewpoint on learning-based multi-agent systems, however, we see that the stationarity assumptions underlying equilibrium concepts are unrealistic. Learning agents may update their objectives, face changing environments, or adopt new strategies as learning continues. As a result, the collective behavior can drift away from equilibrium, sometimes gradually and sometimes abruptly (Rand & Nowak, 2013; Galla & Farmer, 2013). For equilibrium concepts to still be useful, a basic challenge is that of detecting departures from equilibrium. This would allow agents (or external entities) to verify whether agents are playing rationally (Fudenberg & Levine, 1998), detect strategic misalignment or collusion (Hardt et al., 2023; Gauthier et al., 2025b), and preempt instability in networked systems (Parshani et al., 2010).

Yet monitoring equilibrium behavior as it unfolds is a challenging statistical problem. Traditional equilibrium testing assumes a fixed dataset (Babichenko et al., 2017). In contrast, practitioners observing ongoing interactions need methods that can assess, in real time, whether the system remains consistent with equilibrium assumptions, or whether

[1]Inria, Ecole Normale Supérieure, PSL Research University, Paris [2]Departments of EECS and Statistics, University of California, Berkeley. Correspondence to: Etienne Gauthier <etienne.gauthier@inria.fr>.

*Proceedings of the 43rd International Conference on Machine Learning*, Seoul, South Korea. PMLR 306, 2026. Copyright 2026 by the author(s).

something fundamental has changed.

To address these challenges, we first introduce a sequential testing framework for repeated normal-form games, monitoring whether observed play aligns with equilibrium, without relying on fixed sample sizes. Our approach leverages the e-value framework for safe, anytime-valid inference (Shafer & Vovk, 2019; Ramdas et al., 2023; Waudby-Smith & Ramdas, 2023; Grünwald et al., 2024; Ramdas & Wang, 2025; Gauthier et al., 2025a). By "betting" against equilibrium, we construct a test supermartingale that accumulates evidence whenever observed payoffs systematically violate equilibrium conditions. This yields a statistically sound, interpretable measure of departure from equilibrium that can be monitored online.

Our framework extends existing equilibrium testing approaches such as that of Babichenko et al. (2017) in several key aspects. First, rather than assuming access to a fixed dataset of historical play, we develop an online procedure that enables sequential monitoring as interactions unfold. This captures the practical goal of detecting shifts from equilibrium as they happen, something that offline methods are inherently ill-suited for. Simply reapplying static tests to a growing dataset is akin to a form of sequential p-hacking (Simmons et al., 2011). Second, our method does not require knowledge of the specific payoff evaluations at the equilibrium under test. It instead operates under weaker information assumptions, requiring only that players' behavior be consistent with some (possibly unknown) equilibrium, thereby expanding the range of settings where testing remains feasible. Additionally, we sharpen the theoretical characterization of correlated equilibrium, obtaining bounds that are linear in the number of actions per player rather than super-exponential as is the case for existing methods. Finally, we leverage a variant of the Benjamini-Hochberg procedure (Benjamini & Hochberg, 1995) for controlling the false discovery rate (FDR) while enhancing power in large games. This method builds on earlier work using e-values for FDR control (Wang & Ramdas, 2022) and related online multiple testing methods (Wang et al., 2025). Unlike prior approaches that address online multiple testing where hypotheses themselves arrive over time (Xu & Ramdas, 2024; Fischer et al., 2025), our setting involves a fixed collection of hypotheses that are continuously updated as evidence accumulates.

We also extend our sequential testing framework to stochastic games (Neyman & Sorin, 2003), where the environment's state evolves in response to players' joint actions. Mathematically, this generalizes Wald's framework (Wald, 1945) to state-dependent Markovian dynamics. We monitor whether observed trajectories remain consistent with a specified target policy (which may itself be a computed equilibrium) even under nonstationary payoffs. Our framework

captures dynamic, state-dependent interactions common in real-world multi-agent systems, where detecting deviations from intended strategies is critical. Examples include verifying adherence to risk policies in automated trading or ensuring compliance with coordination rules in autonomous driving. Our method enables online detection of such departures, supporting safety, compliance, and reliability in complex adaptive environments.

## 2. Monitoring Repeated Normal-Form Games

In many multi-agent systems, agents repeatedly interact over time, producing sequences of actions and payoffs. A central question in these settings is whether observed behavior aligns with rational strategic play. In this section, we take equilibrium as the benchmark of strategic behavior against which play is monitored. We consider a repeated, finite game setting where, at each round, players choose actions and receive payoffs according to the game's payoff structure. We assume that the joint action profile is drawn identically and independently from a fixed (possibly unknown) strategy at each round, enabling martingale and sequential inference techniques.

**Notation and setup.** We consider a finite multi-player game with $n$ players, and for convenience we write $[n] := \{1, \ldots, n\}$. For each player $i \in [n]$, let $A_i$ be the finite action set available to that player, yielding the joint action space $\mathbf{A} := A_1 \times \cdots \times A_n$. Let $u_i : \mathbf{A} \to [0, 1]$ be the *payoff function* of player $i$; the restriction to the unit interval is without loss of generality, as any bounded payoff function can be rescaled to $[0, 1]$. We write $\mathbf{a} = (a_1, \ldots, a_n)$ for an *action profile*, and, following standard game-theoretic notation, we use $a_{-i}$ to denote the actions of all players other than $i$, so that $u_i(a_i, a_{-i})$ is the payoff of player $i$ when the action profile is $\mathbf{a}$. We evaluate the payoffs of randomized strategies using the standard expected utility, i.e., the expectation of $u_i$ taken under the joint distribution over action profiles induced by the players' (possibly randomized) strategies. For any finite set $S$, we write $\Delta(S)$ for the probability simplex over $S$, i.e., the set of all probability measures on $S$. Throughout, we use the shorthand $a \wedge b := \min(a, b)$ and $a \vee b := \max(a, b)$.

**Equilibria in games.** Equilibrium concepts provide a formal way to capture situations in which players' strategies are mutually consistent: an equilibrium captures a state of strategic stability in which each agent's behavior is optimal given the behavior of others, providing a useful benchmark for predicting or analyzing multi-agent interactions. For each player $i$, let $\pi_i \in \Delta(A_i)$ denote a candidate (possibly mixed) strategy. A *joint strategy profile* specifies a strategy for each player; we write $\pi = (\pi_1, \ldots, \pi_n)$ in general, and $\pi = \pi_1 \times \cdots \times \pi_n$ when the players' strategies are inde-

pendent (i.e., the joint distribution $\pi$ is the product measure of the individual strategies $\pi_i$), which is the form assumed for Nash equilibrium.

**Definition 2.1** (Nash Equilibrium)**.** A joint strategy profile $\pi = \pi_1 \times \cdots \times \pi_n$ is a *Nash equilibrium* if, for each player $i \in [n]$ and all alternative actions $a_i' \in A_i$, we have

$$\mathbb{E}_{a \sim \pi}[u_i(a_i, a_{-i})] \geq \mathbb{E}_{a \sim \pi}[u_i(a_i', a_{-i})].$$

In a Nash equilibrium, no unilateral deviation is profitable. Observed deviations are thus interpreted as evidence of non-equilibrium play.

For clarity and concreteness, in the main text we focus on Nash equilibria; the analysis of other equilibrium concepts, including coarse, coarse correlated, and approximate equilibria, is deferred to Appendix D.

**Hypothesis testing in repeated games.** We observe repeated rounds of interaction indexed by $t = 1, 2, \ldots$, generating action profiles $\mathbf{a}_t = (a_{1,t}, \ldots, a_{n,t})$. We assume that the players' action profile in a given round is drawn identically and independently from some joint distribution.

Let $\pi$ denote the (unknown) joint strategy according to which the action profiles $\mathbf{a}_t$ are generated. The null hypothesis assumes that the generating strategy $\pi$ is a Nash equilibrium, as defined in Definition 2.1. Formally, we write

$$\mathcal{H}_0 : \forall i, \, \forall a_i', \;\; \mathbb{E}[u_i(a_i, a_{-i})] \geq \mathbb{E}[u_i(a_i', a_{-i})], \quad (1)$$

where the expectation is taken over $a \sim \pi$.

For the alternative hypothesis, we consider deviations taking the form of an $\eta$-approximate Nash equilibrium for some fixed $\eta > 0$:

$$\mathcal{H}_1 : \exists i, \, \exists a_i', \;\; \mathbb{E}[u_i(a_i, a_{-i})] \leq \mathbb{E}[u_i(a_i', a_{-i})] - \eta, \quad (2)$$

where again the expectation is over $a \sim \pi$. Note that the boundedness condition $u_i \in [0, 1]$ implies that we can select $\eta \in (0, 1]$. This formulation mirrors the approach by Babichenko et al. (2017), where testing is defined with respect to a fixed slack parameter $\eta > 0$.

We note that one cannot simply define $\mathcal{H}_1$ as "the strategy is not a Nash equilibrium," because then the deviations could be arbitrarily small, and the sequential test may never reliably reject. Appendix E provides a concrete example illustrating why a positive error margin $\eta > 0$ is necessary.

### 2.1. Sequential testing with e-values

We assess whether observed play is consistent with equilibrium using a sequential testing framework based on *e-values*. An e-value for $\mathcal{H}_0$ is simply a nonnegative random variable $E$ satisfying $\mathbb{E}_{\mathcal{H}_0}[E] \leq 1$ under the null hypothesis $\mathcal{H}_0$.

Intuitively, an e-value can be interpreted as the wealth raised via a betting strategy: if the null is true, one cannot systematically "make money," so the expected wealth never exceeds the initial stake.

For each player $i$ and each potential deviation $a_i' \in A_i$, we define a per-round *regret-like increment*

$$X_{i,t}^{(a_i')} := u_i(a_{i,t}, a_{-i,t}) - u_i(a_i', a_{-i,t}), \quad (3)$$

which measures how much worse the observed action $a_{i,t}$ performs compared to the deviation $a_i'$ given the opponents' actions. Note that testing all pure-action deviations is without loss of generality: utility is linear in the mixed strategy, so a profitable mixed deviation implies a profitable pure one.

To monitor whether any deviation would have been profitable, we turn the per-round regret-like increments into a simple e-value for each player and each potential deviation. Intuitively, the e-value grows when the observed action falls short of the deviation, providing evidence against the null hypothesis. The following straightforward lemma formalizes this construction:

**Lemma 2.2.** *For any $\lambda \in (0, 1]$, the per-round quantity*

$$E_{i,t}^{(a_i')}(\lambda) := 1 - \lambda X_{i,t}^{(a_i')} \quad (4)$$

*where $X_{i,t}^{(a_i')}$ is defined in Equation (3) is an e-value for $\mathcal{H}_0$.*

This construction is tailored to bounded-outcome random variables: it relies on $X_{i,t}^{(a_i')} \in [-1, 1]$, which holds since the payoffs $u_i$ take values in $[0, 1]$. Within this class, it is known to be optimal in the sense of Clerico (2024; 2025); Larsson et al. (2026), and similar constructions have appeared in previous work by Waudby-Smith & Ramdas (2023); Orabona & Jun (2024); Larsson et al. (2025). Other e-value constructions are possible, each with its own trade-offs (for instance exponential tilting, or the likelihood-ratio e-values we develop in Appendix G). We adopt the form in Equation (4) for concreteness.

The parameter $\lambda$ scales the per-round increment in the e-value, controlling how aggressively the gambler "bets" against the null: larger $\lambda$ amplifies the effect of each deviation, while smaller $\lambda$ is more conservative.

To interpret these processes sequentially, we view each per-round e-value $E_{i,t}^{(a_i')}(\lambda)$ as the multiplicative factor by which a gambler's wealth is updated when betting against the null at round $t$. The cumulative product of these e-values therefore represents the gambler's wealth after $t$ rounds of repeatedly staking a fraction of wealth according to the chosen rule. Under a valid null this wealth process cannot be expected to grow. Lemma 2.3 formalizes this intuition and provides a rigorous guarantee that this cumulative process

is controlled under the null, which is key for constructing valid sequential tests.

**Lemma 2.3.** *Let $E_{i,t}^{(a_i')}(\lambda)$ be the e-value defined in Equation (4) for some $\lambda \in (0,1]$. Define the cumulative process*

$$M_{i,t}^{(a_i')}(\lambda) := \prod_{s=1}^{t} E_{i,s}^{(a_i')}(\lambda), \quad t \geq 0, \qquad (5)$$

*with $M_{i,0}^{(a_i')}(\lambda) = 1$ by the empty product convention. Let $(\mathcal{F}_t)_{t\geq 0}$ denote the filtration representing all information available up to time $t$, i.e., $\mathcal{F}_t = \sigma(\mathbf{a}_1, \ldots, \mathbf{a}_t)$. Then, under the null hypothesis $\mathcal{H}_0$, $(M_{i,t}^{(a_i')}(\lambda))_{t\geq 0}$ is a nonnegative supermartingale with respect to $(\mathcal{F}_t)_{t\geq 0}$.*

To maintain the flow of exposition, we defer the proofs of this section to Appendix C.

While Lemma 2.3 guarantees control under $\mathcal{H}_0$, the choice of $\lambda$ is critical for maximizing the e-value's growth rate (i.e., its *power*) under the alternative hypothesis $\mathcal{H}_1$. The standard approach in sequential testing and portfolio management is to select the $\lambda$ that maximizes the expected logarithmic growth of the wealth process (Kelly, 1956; Cover & Thomas, 2006). This is known as the log-optimal or Kelly criterion.

**Proposition 2.4.** *Assume $\mathcal{H}_1$, implying $\mathbb{E}_{\mathcal{H}_1}[X_{i,1}^{(a_i')}] \leq -\eta$ for some pair $(i, a_i')$. Define $\mathcal{P}(\mathcal{H}_1)$ as the set of distributions $P$ with $\mathbb{E}_P[X] \leq -\eta$. The parameter $\lambda^*$ maximizing the worst-case expected logarithmic growth is given by:*

$$\lambda^* := \operatorname*{argmax}_{\lambda \in (0,1]} \min_{P \in \mathcal{P}(\mathcal{H}_1)} \mathbb{E}_P[\log(E_{i,1}^{(a_i')}(\lambda))] = \eta.$$

Proposition 2.4 gives a clean theoretical insight: the optimal betting fraction $\lambda$ exactly matches the slack parameter $\eta$ that specifies the alternative hypothesis. In practice, however, $\eta$ is usually unknown: it is precisely what we aim to detect. As a result, fixing a single value of $\lambda$ that does not align with the true $\eta$ can lead to a substantial loss of power.

This motivates considering a *mixture* of e-values corresponding to different $\lambda$ values, rather than committing to a single one. This approach, introduced by Robbins (1970) in its modern form, automatically balances sensitivity across a range of potential deviations without requiring a prior selection of a single tuning parameter. Variants of this approach have been widely used in recent work on sequential testing and e-values (De la Peña et al., 2009; Kaufmann & Koolen, 2021; Howard et al., 2021; Waudby-Smith et al., 2024). An alternative is a *plug-in* approach, in which $\lambda$ is set at each round using a data-driven estimate computed from past observations; this preserves the supermartingale property as long as $\lambda_t$ is $\mathcal{F}_{t-1}$-measurable (Waudby-Smith & Ramdas, 2023). We adopt the mixture approach in what follows, but the procedure extends directly to plug-in choices of $\lambda$.

**Corollary 2.5.** *Let $\nu \in \Delta((0,1])$ be any probability distribution on $(0,1]$, and define the mixture process*

$$M_{i,t}^{(a_i')}(\nu) := \int_0^1 M_{i,t}^{(a_i')}(\lambda)\, d\nu(\lambda), \qquad (6)$$

*where $M_{i,t}^{(a_i')}(\lambda)$ is the supermartingale defined in Equation (5). Then, under the null hypothesis $\mathcal{H}_0$, $(M_{i,t}^{(a_i')}(\nu))_{t\geq 0}$ is also a nonnegative supermartingale.*

The supermartingale property motivates a sequential procedure that flags unusually large cumulative values, as we describe below.

**Family-wise error rate control.** Intuitively, if the null holds, no player can systematically "profit" by deviating from the equilibrium. We formalize this by combining a union bound over all possible deviations with Ville's inequality, which guarantees that the corresponding supermartingales are unlikely to reach unusually large values:

**Theorem 2.6.** *Let $M_{i,t}^{(a_i')}(\nu)$ be the supermartingales defined in Equation (6) for each player $i \in [n]$ and each deviation $a_i' \in A_i$, for some $\nu \in \Delta((0,1])$. Then for any $\alpha \in (0,1)$,*

$$\mathbb{P}_{\mathcal{H}_0}\left( \exists i, \exists a_i' : \sup_{t \geq 0} M_{i,t}^{(a_i')}(\nu) \geq \frac{\sum_{i=1}^{n} |A_i|}{\alpha} \right) \leq \alpha.$$

Therefore, tracking the supermartingales $M_{i,t}^{(a_i')}(\nu)$ provides a concrete sequential monitoring rule: we flag a deviation from the equilibrium whenever any process exceeds the threshold $\sum_{i=1}^{n} |A_i|/\alpha$, with a controlled *family-wise error rate (FWER)* of at most $\alpha$ under the null. We summarize the testing procedure in Algorithm 1.

---

**Algorithm 1** FWER Control – Normal-Form Games

---

Input: significance level $\alpha \in (0,1)$, players $\{1, \ldots, n\}$, action sets $\mathbf{A}$, mixture distribution $\nu$
Set threshold $b = \sum_{i=1}^{n} |A_i|/\alpha$
Initialize $M_{i,0}^{(a_i')}(\nu) \leftarrow 1$ for all $i$ and deviations $a_i' \in A_i$
**Repeat for each round** $t = 1, 2, \ldots$:
    For each player $i$ and deviation $a_i'$:
        Observe $X_{i,t}^{(a_i')}$
        Update $M_{i,t}^{(a_i')}(\nu) = \int_0^1 \prod_{s=1}^{t}(1 - \lambda X_{i,s}^{(a_i')})\, d\nu(\lambda)$
        If $M_{i,t}^{(a_i')}(\nu) \geq b$, flag deviation: reject $\mathcal{H}_0$ and stop

---

Having established control under $\mathcal{H}_0$, we now turn to the complementary question of detection. Under $\mathcal{H}_1$, true deviations should eventually trigger the test.

**Expected rejection time under the alternative.** Under the alternative hypothesis $\mathcal{H}_1$, there exists at least one player $i \in [n]$ and deviation $a_i' \in A_i$ such that the deviation is profitable, i.e., $\mathbb{E}_{\mathcal{H}_1}[X_{i,1}^{(a_i')}] \leq -\eta < 0$. In this case, the corresponding supermartingale tends to grow over time. If the margin $\eta$ were known a priori, one could tune the supermartingale parameter to obtain a sharp bound on the expected stopping time matching that of Babichenko et al. (2017); we detail this derivation in Appendix C. However, a key advantage of the mixture approach is that it does not require knowledge of $\eta$. The following theorem demonstrates that even when $\eta$ is unknown, using a uniform mixture $\nu$ allows us to recover the same asymptotic behavior, mirroring fundamental bounds in sequential analysis (Garivier & Kaufmann, 2016; Kaufmann et al., 2016; Agrawal et al., 2021; Shekhar & Ramdas, 2023a; Chugg et al., 2023; Podkopaev & Ramdas, 2023; Podkopaev et al., 2023; Waudby-Smith et al., 2025):

**Theorem 2.7.** *For any $b > 1$ and mixture $\nu \in \Delta((0,1])$, let*

$$\tau_i^{(a_i')}(\nu) := \inf\{t \geq 0 : M_{i,t}^{(a_i')}(\nu) \geq b\}. \quad (7)$$

*If $\nu$ is the uniform distribution on $(0,1]$, i.e., $d\nu(\lambda) = d\lambda$, then:*

$$\mathbb{E}_{\mathcal{H}_1}[\tau_i^{(a_i')}(\nu)] \leq \frac{12(\log(b) + \log(4/\eta) + \log(3/2))}{\eta^2}.$$

Combining Theorems 2.6 and 2.7, by monitoring all players $i \in [n]$ and deviations $a_i' \in A_i$ with threshold $b = \sum_{i=1}^{n} |A_i|/\alpha$, the sequential test controls the FWER at level $\alpha$ under $\mathcal{H}_0$. Moreover, under $\mathcal{H}_1$ there exists at least one player and deviation for which the expected time to rejection is bounded by

$$\mathbb{E}_{\mathcal{H}_1}[\tau_i^{(a_i')}(\nu)] = O(\log(\sum_{i=1}^{n} |A_i|/\alpha)),$$

which matches the asymptotic behavior of the testing bound in Babichenko et al. (2017) when the sample size is fixed.

**False discovery rate control.** Theorem 2.6 provides a conservative guarantee on the FWER, leveraging the supermartingale property along with a union bound over the number $m = \sum_{i=1}^{n} |A_i|$ of hypotheses. However, this approach treats the collection of all $m$ hypotheses,

$$\mathcal{H} = \{H_i^{(a_i')} : i \in [n], a_i' \in A_i\}, \quad (8)$$

as a single composite event, controlling only the probability of *any* false alarm. In large games where each player may have many potential deviations, FWER control can be overly strict, substantially delaying detection of genuine deviations.

To obtain a less conservative yet valid procedure, we apply the standard e-BH procedure (Wang & Ramdas, 2022) to

e-values stopped at a single global stopping time, valid by optional stopping. Our goal is to control the *false discovery rate (FDR)*, which is the expected fraction of falsely rejected null hypotheses among all rejections:

$$\mathrm{FDR} := \mathbb{E}\left[\frac{|R \cap \mathcal{H}_0|}{1 \vee |R|}\right], \quad (9)$$

where $R$ is the set of rejected hypotheses and $\mathcal{H}_0 \subseteq \mathcal{H}$ denotes the true nulls. Intuitively, controlling the FDR at level $\alpha$ ensures that, on average, at most an $\alpha$ fraction of the rejections are false alarms, providing a principled balance between discovery and reliability. Concretely, our procedure monitors all e-processes in parallel and applies the e-BH rule at each round to their *current* values, producing a candidate rejection set $R_t(\nu)$. The experiment is stopped at the single global stopping time

$$\tau^{\mathrm{FDR}}(\nu) := \inf\{t \geq 0 : |R_t(\nu)| > 0\}, \quad (10)$$

the first round at which e-BH rejects at least one hypothesis. All e-processes are stopped simultaneously at this common time $\tau^{\mathrm{FDR}}(\nu)$ (we do not assign a separate stopping time to each hypothesis) and the e-BH procedure is applied to the resulting stopped e-values $M_{i,\tau^{\mathrm{FDR}}(\nu)}^{(a_i')}(\nu)$. Since each $M_{i,t}^{(a_i')}(\nu)$ is a nonnegative supermartingale with unit initial value, the optional stopping theorem guarantees that these stopped values are themselves valid e-values, so the standard e-BH guarantee applies. We summarize the procedure in Algorithm 2.

The e-BH procedure allows the $\alpha$ budget to be distributed across hypotheses via weights $\gamma_{i,a_i'} \in (0,1]$, which satisfy $\sum_{i,a_i'} \gamma_{i,a_i'} = 1$. These weights can reflect prior importance or desired allocation of the error budget; a simple default is uniform weighting, $\gamma_{i,a_i'} = 1/m$, giving equal priority to each deviation.

For completeness, we recall the e-BH procedure in Algorithm 2, with the conventions $\max(\varnothing) = 0$ and $1/0 = \infty$.

**Theorem 2.8.** *Let $\tau^{\mathrm{FDR}}(\nu)$ be the global stopping time at which the procedure first raises an alarm defined in Equation (10). Then the rejection set $R_{\tau^{\mathrm{FDR}}(\nu)}(\nu)$ produced by Algorithm 2 satisfies*

$$\mathbb{E}\left[\frac{|R_{\tau^{\mathrm{FDR}}(\nu)}(\nu) \cap \mathcal{H}_0|}{1 \vee |R_{\tau^{\mathrm{FDR}}(\nu)}(\nu)|}\right] \leq \alpha,$$

*where $\mathcal{H}_0$ is the set of true nulls.*

Practically, this allows the experimenter to track all supermartingales $M_{i,t}^{(a_i')}(\nu)$ in parallel and declare a violation of equilibrium whenever the e-BH rule rejects one or more nulls. While the FDR guarantee is less strict than controlling the family-wise error rate, this relaxation translates into

---

**Algorithm 2** FDR Control – Normal-Form Games

Input: significance level $\alpha \in (0,1)$, players $\{1,\ldots,n\}$, action sets $\mathbf{A}$, weights $\{\gamma_{i,a_i'}\}$, mixture $\nu$

Initialize $M_{i,0}^{(a_i')}(\nu) \leftarrow 1$ for all $i$ and $a_i' \in A_i$

**Repeat for each round** $t = 1, 2, \ldots$:

For each player $i$ and deviation $a_i'$:

Observe $X_{i,t}^{(a_i')}$ and update

$M_{i,t}^{(a_i')}(\nu) = \int_0^1 \prod_{s=1}^t (1 - \lambda X_{i,s}^{(a_i')}) \, d\nu(\lambda)$

Compute for each $k \in [m]$:

$N_t(k) = \#\{(i, a_i') : M_{i,t}^{(a_i')}(\nu) \geq \frac{1}{k\alpha\gamma_{i,a_i'}}\}$

Set $k_t = \max\{k : N_t(k) \geq k\}$

Define rejection set

$R_t(\nu) = \{(i, a_i') : M_{i,t}^{(a_i')}(\nu) \geq \frac{1}{k_t\alpha\gamma_{i,a_i'}}\}$

---

a gain in statistical power, as formalized by the following result:

**Proposition 2.9.** *Let $\tau_i^{(a_i')}(\nu)$ be the FWER detection time defined in Equation (7) with threshold $b = \sum_{i=1}^n |A_i|/\alpha$ and any mixture $\nu \in \Delta((0,1])$. Consider the FDR procedure with uniform weights $\gamma_{i,a_i'} = 1/m$. Then, almost surely,*

$$\tau^{\mathrm{FDR}}(\nu) \leq \tau_i^{(a_i')}(\nu).$$

*In other words, when using uniform weights, the FDR procedure flags a deviation at least as early as the conservative FWER procedure.*

*Remark* 2.10. Our framework assumes access to the payoff evaluations $u_i(a_i', a_{-i})$ for all players $i \in [n]$ and all possible deviations $a_i' \in A_i$, but it does not require any knowledge of the equilibrium joint strategy $\pi$. This requirement is less restrictive than in Babichenko et al. (2017), where the testing framework relies not only on payoff evaluations for all possible deviations but also on the knowledge of the joint strategy $\pi$. In Appendix G, we present an alternative testing framework that operates in the complementary setting where $\pi$ is known, but the payoff functions are not directly observable.

# 3. Monitoring Stochastic Games

In this section, we extend the monitoring framework from repeated normal-form games to *stochastic games*, which generalize repeated interactions by incorporating state-dependent dynamics. In a stochastic game, the environment occupies one of finitely many states, and the players' joint actions determine both their immediate payoffs and the probability distribution over the next state. This added structure captures many realistic multi-agent settings, where the consequences of an action depend on a context that itself evolves over time. The shift to state-dependent dynamics,

however, also changes the nature of the testing problem: as we explain below, verifying equilibrium directly becomes intractable, and we therefore move from equilibrium testing to compliance testing against a specified target policy.

## 3.1. Setting and problem formulation

**Dynamics.** A finite stochastic game is defined by the tuple $(\mathcal{S}, \mathbf{A}, \{r_i\}_{i=1}^n, P)$, where

- $\mathcal{S}$ is the finite set of states.
- $\mathbf{A} = A_1 \times \cdots \times A_n$ is the joint finite action set.
- $r_i : \mathcal{S} \times \mathbf{A} \to [0,1]$ is the reward function for player $i$.
- $P : \mathcal{S} \times \mathbf{A} \to \Delta(\mathcal{S})$ is the state transition kernel.

Time proceeds in discrete rounds $t = 1, 2, \ldots$, with each round proceeding as follows. At the beginning of round $t$, the game is in state $s_t \in \mathcal{S}$ and each player $i$ selects an action $a_{i,t} \in A_i$ according to their (possibly mixed) strategy $\pi_i(\cdot \mid s_t) \in \Delta(A_i)$. The joint action profile is denoted $\mathbf{a}_t = (a_{1,t}, \ldots, a_{n,t})$. Given the current state-action pair $(s_t, \mathbf{a}_t)$, the next state $s_{t+1}$ is sampled from the transition kernel $s_{t+1} \sim P(\cdot \mid s_t, \mathbf{a}_t)$, and each player receives an immediate reward $r_{i,t} = r_i(s_t, \mathbf{a}_t)$. This recursive structure induces a stochastic process over states and actions $(s_t, \mathbf{a}_t)_{t \geq 1}$, which reflects both the strategic choices of the players and the stochastic evolution of the environment.

**Problem formulation.** A Nash equilibrium in this setting requires that no player can improve their cumulative discounted reward by deviating. Unfortunately, verifying these global optimality conditions is computationally intractable as it requires access to full counterfactual trajectories. Therefore, instead of testing for equilibrium properties directly (as in Section 2), we focus on the tractable problem of *compliance testing*: verifying whether agents adhere to a specific candidate strategy profile $\pi$ (which may be a computed equilibrium), or whether they deviate towards an alternative $\pi^{(1)}$.

## 3.2. Sequential testing via likelihood ratios

In this subsection, we develop sequential tests based on likelihood ratios between the candidate policy $\pi$ and a hypothesized alternative $\pi^{(1)}$. We fix $\pi^{(1)} = (\pi_1^{(1)}, \ldots, \pi_n^{(1)})$, which denotes a hypothesized deviation from the strategy profile $\pi$, where each $\pi_i^{(1)} : \mathcal{S} \to \Delta(A_i)$ specifies a state-dependent alternative strategy for player $i \in [n]$.

For a given player $i \in [n]$ and round $t$, define the per-round likelihood ratio

$$E_{i,t} := \frac{\pi_i^{(1)}(a_{i,t} \mid s_t)}{\pi_i(a_{i,t} \mid s_t)}. \tag{11}$$

Such likelihood ratios constitute canonical examples of e-values used in safe, anytime-valid inference (Wasserman

et al., 2020; Shafer, 2021; Grünwald et al., 2024). Intuitively, $E_{i,t}$ measures the evidence provided by the observed action $a_{i,t}$ in favor of the alternative strategy $\pi_i^{(1)}$ over the candidate $\pi_i$, given the current state $s_t$. The ratio indicates how much more (or less) likely the action was under the hypothesized deviation compared to the candidate policy.

**Lemma 3.1.** *For any player $i$, we have*

$$\mathbb{E}_{\mathcal{H}_0}[E_{i,t} \mid \mathcal{F}_{t-1}] = 1,$$

*where $\mathcal{F}_{t-1}$ is the filtration generated by all states and actions up to round $t-1$. Hence, $E_{i,t}$ is a valid e-value conditional on the history under the null hypothesis $\mathcal{H}_0$.*

For clarity of exposition, we defer the proofs for all theoretical results in this section to Appendix F.

As in the normal-form case, we can accumulate these per-round e-values over time to form a test statistic that reflects the total cumulative evidence. Define

$$M_{i,t} := \prod_{s=1}^{t} E_{i,s}, \quad t \geq 0, \tag{12}$$

with $M_{i,0} := 1$ by the empty product convention.

**Lemma 3.2.** *Under $\mathcal{H}_0$, the sequence $(M_{i,t})_{t \geq 0}$ defined in Equation (12) is a nonnegative martingale with respect to the filtration $(\mathcal{F}_t)_{t \geq 0}$.*

Since the processes $(M_{i,t})_{t \geq 0}$ are martingales (and hence supermartingales) under $\mathcal{H}_0$, they cannot grow too large in expectation. This observation allows us to construct a sequential testing procedure, following the same principle as in Theorem 2.6 in the case of normal-form games. The resulting test follows the same structure as Algorithm 1, replacing the regret-based update with the likelihood update. We omit the pseudocode for brevity. This ensures that the FWER is controlled at level $\alpha$ by employing a stopping rule that rejects $\mathcal{H}_0$ whenever any martingale exceeds $n/\alpha$:

**Theorem 3.3.** *For any threshold $b > 1$, we have:*

$$\mathbb{P}_{\mathcal{H}_0}\left(\exists i \in [n] : \sup_{t \geq 0} M_{i,t} \geq b\right) \leq \frac{n}{b}.$$

*In particular, setting $b = n/\alpha$ ensures that the FWER is controlled at level $\alpha$.*

Having ensured that the FWER is controlled under the null hypothesis $\mathcal{H}_0$, we next quantify the detection efficiency under the alternative hypothesis $\mathcal{H}_1$. As in the normal-form setting, our key performance metric is the detection time $\tau_i$ for a given threshold $b > 1$:

$$\tau_i := \inf\{t \geq 1 : M_{i,t} \geq b\}. \tag{13}$$

The expected duration of this detection phase depends on the distinguishability of the deviation $\pi_i^{(1)}$ from the baseline $\pi_i$.

To quantify this, we first define the conditional Kullback-Leibler (KL) divergence at each state $s \in \mathcal{S}$:

$$\mathrm{KL}_i(s) := \sum_{a_i \in A_i} \pi_i^{(1)}(a_i \mid s) \log \frac{\pi_i^{(1)}(a_i \mid s)}{\pi_i(a_i \mid s)}. \tag{14}$$

In a dynamic setting, the rate at which evidence accumulates depends on the frequency with which states are visited. Assuming the Markov chain induced by $\pi^{(1)}$ is ergodic with stationary distribution $\mu \in \Delta(\mathcal{S})$, the relevant measure of signal strength is the state-averaged KL divergence:

$$\bar{\mathrm{KL}}_i(\mu) := \sum_{s \in \mathcal{S}} \mu(s) \, \mathrm{KL}_i(s). \tag{15}$$

Intuitively, $\bar{\mathrm{KL}}_i(\mu)$ represents the asymptotic rate of information accumulation; a larger divergence implies that the martingale drifts upward more rapidly, leading to faster detection.

**Theorem 3.4.** *Suppose that the Markov chain induced by $\pi^{(1)}$ is ergodic, with unique stationary distribution $\mu \in \Delta(\mathcal{S})$, and that the process is started in stationarity, i.e., $s_0 \sim \mu$. Assume the support condition $\pi_i(a \mid s) > 0$ whenever $\pi_i^{(1)}(a \mid s) > 0$ holds for all $s \in \mathcal{S}$ and $a \in A_i$. Then, we have*

$$\mathbb{E}_{\mathcal{H}_1}[\tau_i] \leq \frac{\log b + C_i}{\bar{\mathrm{KL}}_i(\mu)},$$

*where $C_i$ is the bounded overshoot constant for player $i$:*

$$C_i := \max_{s \in \mathcal{S}} \max_{a \in A_i} \left| \log \frac{\pi_i^{(1)}(a \mid s)}{\pi_i(a \mid s)} \right| < \infty.$$

If the process does not start exactly in stationarity but the Markov chain induced by the alternative policy $\pi^{(1)}$ is ergodic and mixes rapidly, then after a burn-in period the state distribution is close to $\mu$. In that case one can obtain a similar bound up to an additive constant accounting for the burn-in.

*Remark* 3.5. In practical settings, the precise alternative strategy $\pi^{(1)}$ used by a deviating agent is rarely known in advance. To address this uncertainty, we can construct a robust test statistic by averaging the likelihood ratios over a prior distribution $\nu$ of plausible deviations. Analogous to the mixture-of-$\lambda$ approach in Section 2, we define the mixture process:

$$M_{i,t}(\nu) := \int \left( \prod_{s=1}^{t} \frac{\pi_i^{(1)}(a_{i,s} \mid s_s)}{\pi_i(a_{i,s} \mid s_s)} \right) d\nu(\pi^{(1)}). \tag{16}$$

Since the integrand is non-negative, Tonelli's theorem ensures that the expectation and the integral can be exchanged.

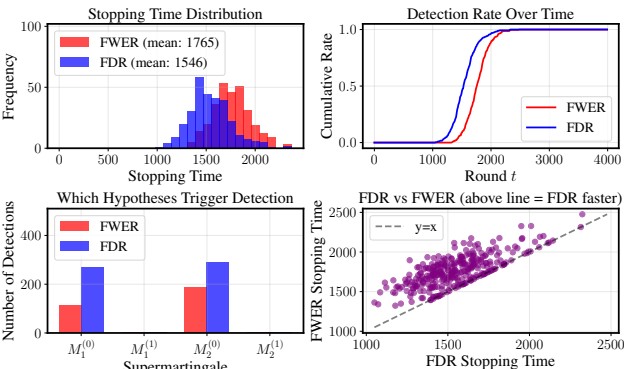

*Figure 1.* Comparison of FWER and FDR detection performance over 300 runs in the $2 \times 2$ normal-form game. The FDR procedure consistently detects deviations earlier. **(Top-Left)** Distribution of stopping times. **(Top-Right)** Cumulative detection rate over time. **(Bottom-Left)** Frequency of detection for each specific hypothesis, showing the procedure correctly targets the two true alternative signals. **(Bottom-Right)** Run-by-run scatter plot.

Consequently, the martingale property under $\mathcal{H}_0$ is preserved, maintaining valid FWER control. This formulation grants robustness against unknown deviations while, as proven in Appendix F, fully preserving the asymptotic detection scaling law established in Theorem 3.4.

Note that this procedure naturally extends to FDR control via the e-BH framework (as in Section 2), enhancing detection power in large games; we omit the details for brevity.

# 4. Experiments

## 4.1. Detection power in normal-form games

We evaluate our framework on a simple $2 \times 2$ normal-form game engineered with two weak, balanced deviation signals ($\eta \approx 0.033$). This scenario challenges the conservative FWER procedure (Algorithm 1), which requires a single supermartingale to cross a high threshold ($b = 20$ for $\alpha = 0.2$). Figure 1 demonstrates that the FDR procedure (Algorithm 2) yields a consistent speedup by utilizing the aggregate signal to trigger rejection at a lower threshold ($k = 2$). The run-by-run comparison confirms Proposition 2.9 empirically: in every trial, the FDR procedure detects the deviation no later than the FWER baseline, proving its advantage in multi-signal settings. To show that these gains persist beyond the minimal $2 \times 2$ setting, Appendix H.2 reports a larger-scale experiment with more players and bigger action spaces, where the FDR procedure's advantage grows as the number of hypotheses increases. See Appendix H for full experimental details. The code is publicly available at https://github.com/GauthierE/anytime-detection-deviation.

## 4.2. Detection power in stochastic games

We now evaluate our framework in dynamic environments with state transitions. We consider two distinct gridworld scenarios to validate the scaling laws of the detection time $\tau$ and the robustness of the mixture martingale approach.

**Scaling law validation (grid soccer).** We begin with a $5 \times 4$ stochastic grid-soccer game in which an Attacker attempts to reach a goal while evading a partially unreliable Defender. We obtain a Nash equilibrium policy $\pi^*$ (details in Appendix H) and introduce deviations via the mixture policy $\pi^{(\varepsilon)} = (1 - \varepsilon)\pi^* + \varepsilon\pi^{\text{afraid}}$, where $\pi^{\text{afraid}}$ shifts probability mass away from aggressive moves.

A key consequence of Theorem 3.4 is that the expected detection time is asymptotically bounded by a constant divided by the KL divergence between the null and alternative policies. In our setting, we show in Appendix H.3.5 that this KL divergence grows quadratically with the deviation magnitude $\varepsilon$. Thus, the theory predicts that the detection time should scale no better than $1/\varepsilon^2$. Figure 2 shows that the empirical detection times follow exactly this $1/\varepsilon^2$ trend, demonstrating that our detector matches the scaling law of the theoretical upper bound and achieves the corresponding optimal asymptotic efficiency.

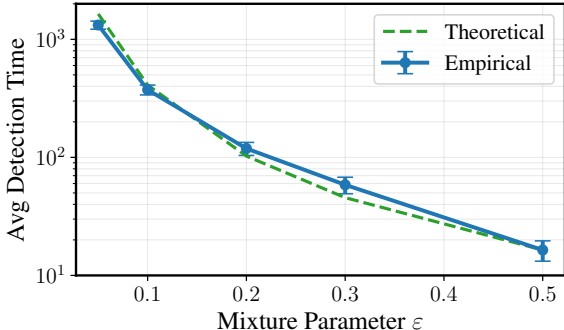

*Figure 2.* **Asymptotic scaling in Grid Soccer.** Empirical detection times (blue) align with the theoretical $O(1/\varepsilon^2)$ curve (green), confirming optimal scaling, with both curves anchored at the final point to compare asymptotic scaling independent of constants.

**Robustness to unknown deviations (predator-prey).** The grid-soccer experiment assumed the deviation magnitude $\varepsilon$ was known to the monitor. We now relax this assumption: the monitor must detect a deviation *without knowing* the alternative policy in advance. In a $10 \times 10$ predator-prey environment, a suspect predator plays

$$\pi^{(\varepsilon)}(a \mid s) = (1 - \varepsilon)\frac{1}{|\mathcal{A}|} + \varepsilon\,\pi^{\text{chase}}(a \mid s),$$

mixing a random-walk baseline with a chase heuristic $\pi^{\text{chase}}$ using a mixture weight $\varepsilon$ that is *unknown* to the monitor. Since the true $\varepsilon$ is unavailable, the monitor cannot form

a single likelihood ratio; instead, following Remark 3.5, it constructs a *mixture* that averages likelihood ratios over a grid of candidate values of $\varepsilon$, thereby marginalizing its uncertainty about the alternative.

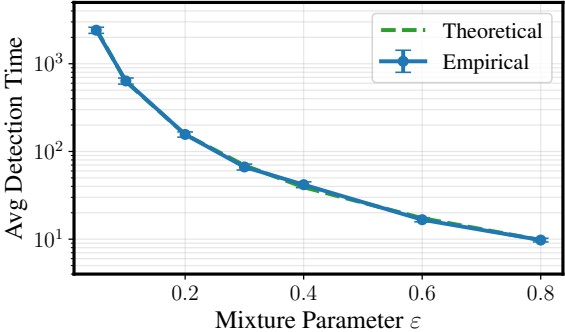

*Figure 3.* **Robustness of the mixture martingale.** The mixture detector achieves the optimal $O(1/\varepsilon^2)$ scaling of Theorem 3.4 despite the deviation magnitude $\varepsilon$ being unknown to the monitor.

As shown in Figure 3, the mixture detector retains the $O(1/\varepsilon^2)$ scaling of Theorem 3.4 despite not knowing the true $\varepsilon$, confirming the robustness claim of Remark 3.5: averaging over a prior preserves valid error control while maintaining asymptotically optimal detection power.

## 5. Discussion

We have presented a unified framework for the sequential detection of strategic deviations in multi-agent systems, enabling anytime-valid monitoring without fixed sample sizes. For repeated normal-form games, we developed an online test for departures from equilibrium that relies solely on payoff evaluations, requiring no knowledge of the equilibrium strategy itself, and we sharpened its power in large games by integrating e-BH procedures for false discovery rate control. For stochastic games, we addressed the problem of compliance testing, using likelihood ratios to verify online whether observed trajectories adhere to a specified target policy under dynamic, state-dependent interactions.

**Limitations and future work.** Our normal-form framework relies on observing counterfactual payoffs, which is standard in equilibrium testing but may require a model of the environment in practice. Conversely, our stochastic framework relies on knowing the target equilibrium $\pi$ to compute likelihoods. Bridging these gaps remains a challenging open problem, likely requiring the integration of value estimation techniques. Furthermore, while we analyzed detection times under stationary deviations, analyzing the response of learning agents to these detections constitutes a promising direction for creating closed-loop, self-correcting multi-agent systems.

## Acknowledgements

The authors thank the anonymous reviewers for their helpful feedback, which improved this work and highlighted a mistake in the original e-BH procedure. EG acknowledges support from the Google PhD Fellowship.

Funded by the European Union (ERC-2022-SYG-OCEAN-101071601). Views and opinions expressed are however those of the author(s) only and do not necessarily reflect those of the European Union or the European Research Council Executive Agency. Neither the European Union nor the granting authority can be held responsible for them.

This publication is part of the Chair "Markets and Learning," supported by Air Liquide, BNP PARIBAS ASSET MANAGEMENT Europe, EDF, Orange and SNCF, sponsors of the Inria Foundation.

This work has also received support from the French government, managed by the National Research Agency, under the France 2030 program with the reference "PR[AI]RIE-PSAI" (ANR-23-IACL-0008).

## Impact Statement

This paper presents work whose goal is to advance the field of Machine Learning. There are many potential societal consequences of our work, none which we feel must be specifically highlighted here.

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

# A. Further Related Work

Our work lies at the intersection of game theory, sequential statistical inference, and multi-agent monitoring. While our primary focus is on game-theoretic stability, our methodology directly tackles the statistical challenge of detecting strategic shifts in real time.

At the same time, our approach connects to the econometric literature on testing equilibrium behavior in games, which develops nonparametric and semiparametric methods to evaluate whether observed strategic interactions are consistent with equilibrium concepts such as Nash or rationalizability. Aradillas-López (2019) proposes nonparametric tests for strategic interaction effects with rationalizability, offering a distribution-free approach to detect deviations from rationalizable behavior. In related work, Aradillas-López et al. (2016) develop a simple test for moment inequality models and demonstrate its application to English auctions, providing practical tools for inference under equilibrium restrictions. Earlier, Aradillas-Lopez & Tamer (2008) analyze the identification power of equilibrium in simple games, illustrating how equilibrium and rationalizability assumptions shape the identified set for structural parameters. Together, these contributions establish a foundation for testing equilibrium behavior in static and auction environments. Beyond static settings, Otsu & Pesendorfer (2023) study equilibrium multiplicity in dynamic games, proposing testing and estimation methods that accommodate multiple equilibria and dynamic strategic interactions. Complementing these approaches, Melo et al. (2019) develop econometric tests of the Quantal Response Hypothesis, bridging behavioral game theory and structural estimation. Our work departs from these econometric approaches by developing a sequential, anytime-valid testing framework that continuously monitors equilibrium consistency over time, offering finite-sample guarantees in dynamic multi-agent environments.

Our framework is conceptually rooted in the broader field of sequential analysis, yet it is distinct from classical change-point detection methods such as CUSUM (Page, 1954) and the Shiryaev-Roberts procedure (Shiryaev, 1961; 1963; Roberts, 1966). While those frameworks are optimized to detect a transition from a baseline to a disordered state at an unknown random time, our problem formulation assumes that the alternative strategy (if present) is active from the onset of monitoring. Mathematically, this aligns our approach with Wald's Sequential Probability Ratio Test (SPRT) (Wald, 1945), rather than the additive recursions of change-point detection. By constructing a multiplicative martingale based on likelihood ratios, our method generalizes the SPRT to the setting of stochastic games, extending the classical optimality guarantees of the Wald-Wolfowitz theorem (Wald & Wolfowitz, 1948) to environments with state-dependent dynamics. Moreover, several recent works have considered detection procedures based on e-values (Shekhar & Ramdas, 2023b; 2024; Shin et al., 2024; Csillag et al., 2025; Fischer & Ramdas, 2025), with some of these works also giving bounds inversely proportional to the KL divergence, though in different contexts and problem formulations. Instead of merely flagging a shift in the distribution of actions, our procedure specifically targets the violation of incentives that define equilibrium. By translating the loss of strategic stability into a monitorable statistical measure, we provide a tool that is highly interpretable in a game-theoretic context and directly relevant for systems where safety and compliance depend on adherence to equilibrium strategies.

Recent work in reinforcement learning (RL) has also begun to integrate statistical inference and hypothesis testing into algorithmic evaluation and policy monitoring. For instance, Colas et al. (2019) discuss the importance of proper statistical testing when comparing RL algorithms, highlighting issues of variance, reproducibility, and false discoveries. Similarly, Ramprasad et al. (2023) and Syrgkanis & Zhan (2023) develop inference procedures for policy evaluation in adaptive, Markovian settings, addressing the challenges of dependence and nonstationarity inherent in online learning. A recent survey by Shi (2025) provides a comprehensive overview of this emerging intersection between statistics and reinforcement learning. While these works share our goal of embedding rigorous statistical reasoning within sequential decision processes, they primarily focus on policy evaluation and algorithm comparison. In contrast, our framework targets strategic monitoring in multi-agent environments, where the objective is not to assess performance but to detect loss of equilibrium and incentive alignment. This distinction places our contribution at the interface of statistical testing and game theory, extending inference tools from RL toward the continuous detection of strategic behavior.

More directly, our approach relates to a growing literature using betting-based ideas for sequential testing. While we apply this framework to detect deviations from game-theoretic equilibria, similar methods have appeared in other strategic and algorithmic settings. For example, Velasco et al. (2025) use e-values to audit pay-per-token schemes in large language models, where detecting tokenization misreporting can also be seen as a form of strategic deviation. There are close technical parallels: our e-value construction in Lemma 2.2 is closely related to their estimator, and our bounds in Theorem 2.7 mirror the guarantees in their main theorem. More broadly, this highlights the generality of betting-based methods: whenever a hypothesis can be tested sequentially via a valid betting construction, e-values provide a natural tool for online monitoring, with the simplest case being tests of expected values, as in both our work and that of Velasco et al. (2025).

# B. Background on Martingale Theory

This appendix provides a concise overview of the basic martingale concepts used throughout the paper. We recall standard definitions from probability theory and sequential analysis, including filtrations, martingales and their variants, stopping times, and three classical results: Ville's inequality, the optional stopping theorem, and Wald's identity. These tools form the foundation for modern e-value and anytime-valid inference frameworks.

## B.1. Filtrations and adapted processes

**Definition B.1** (Filtration). Let $(\Omega, \mathcal{F}, \mathbb{P})$ be a probability space. A *filtration* $(\mathcal{F}_t)_{t \geq 0}$ is a non-decreasing sequence of sub-$\sigma$-algebras of $\mathcal{F}$ such that

$$\mathcal{F}_s \subseteq \mathcal{F}_t \subseteq \mathcal{F}, \quad \text{for all } s \leq t.$$

Intuitively, $(\mathcal{F}_t)_{t \geq 0}$ represents the collection of all events whose outcomes are known by time $t$; that is, the information available up to time $t$.

A stochastic process $(X_t)_{t \geq 0}$ is said to be *adapted* to the filtration $(\mathcal{F}_t)_{t \geq 0}$ if each random variable $X_t$ is $\mathcal{F}_t$-measurable. Adaptedness ensures that the process does not "look into the future": its value at time $t$ depends only on information available up to that time.

## B.2. Martingales, supermartingales, and submartingales

**Definition B.2** (Martingale, Supermartingale, Submartingale). An integrable adapted process $(M_t)_{t \geq 0}$ is a *martingale* with respect to $(\mathcal{F}_t)_{t \geq 0}$ if

$$\mathbb{E}[|M_t|] < \infty, \quad \text{and} \quad \mathbb{E}[M_t \mid \mathcal{F}_{t-1}] = M_{t-1}, \quad \text{for all } t \geq 1.$$

If instead

$$\mathbb{E}[M_t \mid \mathcal{F}_{t-1}] \leq M_{t-1} \quad (\text{resp. } \geq M_{t-1}),$$

then $(M_t)_{t \geq 0}$ is called a *supermartingale* (resp. *submartingale*).

A martingale represents the formal mathematical model of a *fair game*: conditional on all current information, the expected future value is equal to the present value. Supermartingales and submartingales capture systematic negative or positive drifts, respectively. These distinctions are essential when constructing sequential tests. Supermartingales are often used to accumulate evidence against a null hypothesis while maintaining valid type-I error control.

When the underlying filtration $(\mathcal{F}_t)_{t \geq 0}$ is clear from context, we may drop it and simply refer to $(M_t)_{t \geq 0}$ as a martingale, supermartingale, or submartingale without explicitly mentioning the filtration.

## B.3. Stopping times

**Definition B.3** (Stopping Time). A random variable $\tau : \Omega \to \{0, 1, 2, \dots\} \cup \{\infty\}$ is a *stopping time* with respect to $(\mathcal{F}_t)_{t \geq 0}$ if

$$\{\tau \leq t\} \in \mathcal{F}_t, \quad \text{for all } t \geq 0.$$

That is, at time $t$ one can determine whether the stopping time has occurred using only the information available up to $t$.

Stopping times formalize the notion of a random time at which a decision is made based on observed data; for example, the first time a process crosses a pre-specified threshold. They play a central role in sequential analysis and are key to defining valid anytime inference procedures.

Again, when the underlying filtration $(\mathcal{F}_t)_{t \geq 0}$ is clear from context, we may drop it and simply refer to $\tau$ as a stopping time without explicitly mentioning the filtration.

## B.4. Ville's inequality (Ville, 1939)

**Theorem B.4** (Ville's Inequality). *Let $(M_t)_{t \geq 0}$ be a nonnegative supermartingale with $M_0 = 1$. Then, for any $\alpha > 0$,*

$$\mathbb{P}\left(\sup_{t \geq 0} M_t \geq \frac{1}{\alpha}\right) \leq \alpha.$$

Ville's inequality is a cornerstone of modern sequential testing theory. It ensures that a nonnegative supermartingale, started at 1, rarely grows very large under the null hypothesis. Intuitively, if $M_t$ represents a "betting process" that grows when evidence accumulates against the null, Ville's inequality guarantees that the probability of falsely accumulating excessive evidence is controlled at level $\alpha$, uniformly over time. This property underlies the *anytime-validity* of e-value-based statistical tests.

### B.5. Wald's identity

**Theorem B.5** (Wald's Identity). *Let $(X_t)_{t \geq 1}$ be i.i.d. random variables with $\mathbb{E}[X_1] = \mu$, and let $\tau$ be a stopping time satisfying $\mathbb{E}[\tau] < \infty$. Then*

$$\mathbb{E}\left[\sum_{t=1}^{\tau} X_t\right] = \mathbb{E}[\tau]\,\mu.$$

Wald's identity links expectations over random (but well-behaved) stopping times to expectations over deterministic horizons. In sequential settings, it provides a bridge between random stopping procedures and expected cumulative quantities such as rewards, costs, or log-likelihood ratios. It is a fundamental result in sequential analysis and optimal stopping theory.

### B.6. Optional stopping theorem

The optional stopping theorem (OST) generalizes Wald's identity to martingales, allowing for dependent increments. Here, we state a simple version that is sufficient for the purposes of this paper:

**Theorem B.6** (Optional Stopping Theorem). *Let $(M_t)_{t \geq 0}$ be a martingale with respect to a filtration $(\mathcal{F}_t)_{t \geq 0}$, and let $\tau$ be a stopping time that is almost surely bounded. Then*

$$\mathbb{E}[M_\tau] = \mathbb{E}[M_0].$$

*Analogously, if $(M_t)_{t \geq 0}$ is a supermartingale, then $\mathbb{E}[M_\tau] \leq \mathbb{E}[M_0]$, and if it is a submartingale, then $\mathbb{E}[M_\tau] \geq \mathbb{E}[M_0]$.*

Together, these results provide the mathematical foundation for constructing sequential tests and monitoring procedures with finite-time guarantees. They formalize how evidence can be accumulated and bounded over time in multi-agent or learning systems.

## C. Proofs for Section 2

### C.1. Sequential testing via coin betting

**Lemma C.1** (Lemma 2.2). *For any $\lambda \in (0, 1]$, the per-round quantity*

$$E_{i,t}^{(a_i')}(\lambda) := 1 - \lambda X_{i,t}^{(a_i')}$$

*where $X_{i,t}^{(a_i')}$ is defined in Equation (3) is an e-value for $\mathcal{H}_0$.*

*Proof.* By definition, $X_{i,t}^{(a_i')} \in [-1, 1]$ since $u_i \in [0, 1]$, so $E_{i,t}^{(a_i')}(\lambda) = 1 - \lambda X_{i,t}^{(a_i')} \geq 0$ for $\lambda \in (0, 1]$. Under $\mathcal{H}_0$, no deviation is profitable, so $\mathbb{E}_{\mathcal{H}_0}[X_{i,t}^{(a_i')}] \geq 0$, which implies $\mathbb{E}_{\mathcal{H}_0}[E_{i,t}^{(a_i')}(\lambda)] \leq 1$. $\qquad\square$

### C.2. Supermartingale construction

**Lemma C.2** (Lemma 2.3). *Let $E_{i,t}^{(a_i')}(\lambda)$ be the e-value defined in Equation (4) for some $\lambda \in (0, 1]$. Define the cumulative process*

$$M_{i,t}^{(a_i')}(\lambda) := \prod_{s=1}^{t} E_{i,s}^{(a_i')}(\lambda), \quad t \geq 0,$$

*with $M_{i,0}^{(a_i')}(\lambda) = 1$ by the empty product convention. Let $(\mathcal{F}_t)_{t \geq 0}$ denote the filtration representing all information available up to time t, i.e., $\mathcal{F}_t = \sigma(\mathbf{a}_1, \ldots, \mathbf{a}_t)$. Then, under the null hypothesis $\mathcal{H}_0$, $(M_{i,t}^{(a_i')}(\lambda))_{t \geq 0}$ is a nonnegative supermartingale with respect to $(\mathcal{F}_t)_{t \geq 0}$.*

*Proof.* By Lemma 2.2, each factor $E_{i,s}^{(a_i')}(\lambda)$ is nonnegative so every finite product $M_{i,t}^{(a_i')}(\lambda)$ is nonnegative. Each $E_{i,t}^{(a_i')}(\lambda)$ is $\mathcal{F}_t$-measurable by construction, hence $M_{i,t}^{(a_i')}(\lambda)$ is adapted to $(\mathcal{F}_t)_{t \geq 0}$. Using the fact that $M_{i,t}^{(a_i')}(\lambda)$ is $\mathcal{F}_t$-measurable,

$$\mathbb{E}_{\mathcal{H}_0}[M_{i,t+1}^{(a_i')}(\lambda) \mid \mathcal{F}_t] = \mathbb{E}_{\mathcal{H}_0}[M_{i,t}^{(a_i')}(\lambda) \cdot E_{i,t+1}^{(a_i')}(\lambda) \mid \mathcal{F}_t]$$
$$= M_{i,t}^{(a_i')}(\lambda) \cdot \mathbb{E}_{\mathcal{H}_0}[E_{i,t+1}^{(a_i')}(\lambda) \mid \mathcal{F}_t].$$

Since each player's action is assumed to be independent of previous actions, we have

$$\mathbb{E}[E_{i,t+1}^{(a_i')}(\lambda) \mid \mathcal{F}_t] = \mathbb{E}[E_{i,t+1}^{(a_i')}(\lambda)] \leq 1,$$

where the last inequality comes from Lemma 2.2. Substituting the above inequality into the previous expression yields

$$\mathbb{E}[M_{i,t+1}^{(a_i')}(\lambda) \mid \mathcal{F}_t] \leq M_{i,t}^{(a_i')}(\lambda),$$

which proves that $(M_{i,t}^{(a_i')}(\lambda))_{t \geq 0}$ is a nonnegative supermartingale. $\qquad\square$

**Proposition C.3** (Proposition 2.4). *Assume $\mathcal{H}_1$, implying $\mathbb{E}_{\mathcal{H}_1}[X_{i,1}^{(a_i')}] \leq -\eta$ for some pair $(i, a_i')$. Define $\mathcal{P}(\mathcal{H}_1)$ as the set of distributions $P$ with $\mathbb{E}_P[X] \leq -\eta$. The parameter $\lambda^*$ maximizing the worst-case expected logarithmic growth is given by:*

$$\lambda^* := \operatorname*{argmax}_{\lambda \in (0,1]} \min_{P \in \mathcal{P}(\mathcal{H}_1)} \mathbb{E}_P[\log(E_{i,1}^{(a_i')}(\lambda))] = \eta.$$

*Proof.* We solve the inner minimization first.

**Inner minimization.** We seek the Least Favorable Distribution (LFD) $P^*$ that solves the minimization:

$$\min_{P \in \mathcal{P}(\mathcal{H}_1)} \mathbb{E}_P[\log(1 - \lambda X)], \quad \text{where } \mathcal{P}(\mathcal{H}_1) = \{P \text{ on } [-1,1] \mid \mathbb{E}_P[X] \leq -\eta\}$$

Let $f(X) = \log(1 - \lambda X)$. This function is strictly concave on $[-1, 1]$ and strictly decreasing in $X$.

**Step 1: Justifying the boundary ($\mu = -\eta$).** Because $f(X)$ is a decreasing function of $X$, the expected utility $\mathbb{E}_P[f(X)]$ is a decreasing function of its mean, $\mu = \mathbb{E}_P[X]$. To minimize $\mathbb{E}_P[f(X)]$ over the set of allowed means $\mu \leq -\eta$, we must choose the distribution with the *largest* possible mean. The largest allowed mean in $\mathcal{P}(\mathcal{H}_1)$ is $\mu = -\eta$. Therefore, the minimum must occur at the boundary of the set, and our problem simplifies to finding the LFD $P^*$ in the smaller set $\{P \text{ on } [-1,1] \mid \mathbb{E}_P[X] = -\eta\}$.

**Step 2: Justifying the two-point distribution.** Now we must solve $\min_{P:\mathbb{E}_P[X]=-\eta} \mathbb{E}_P[f(X)]$. By the concavity of $f(x)$, it is bounded below by the secant line $L(x)$ connecting its endpoints on the interval $[-1, 1]$. Taking the expectation over any $P$ with mean $\mu = -\eta$:
$$\mathbb{E}_P[f(X)] \geq \mathbb{E}_P[L(X)]$$

Due to the linearity of $L(x)$, $\mathbb{E}_P[L(X)] = L(\mathbb{E}_P[X]) = L(-\eta)$, which is a constant lower bound. This bound is achieved if and only if the distribution's mass is placed *only* on the points where $f(x) = L(x)$, which are precisely the endpoints $\{-1, 1\}$.

**Step 3: Calculating the LFD probabilities.** From Steps 1 and 2, the LFD $P^*$ is the unique two-point distribution on $\{-1, 1\}$ with mean $-\eta$. Let $P^*(X = -1) = p$ and $P^*(X = 1) = 1 - p$. The mean constraint is:

$$\mathbb{E}_{P^*}[X] = p(-1) + (1 - p)(1) = 1 - 2p = -\eta.$$

Solving for $p$: $2p = 1 + \eta$, which gives $p = \frac{1+\eta}{2}$. Thus, $P^*(X = -1) = \frac{1+\eta}{2}$ and $P^*(X = 1) = \frac{1-\eta}{2}$.

**Outer maximization (finding $\lambda^*$).** We now maximize the expected log-growth rate $g(\lambda) = \mathbb{E}_{P^*}[\log(1 - \lambda X)]$ with respect to the LFD $P^*$:

$$g(\lambda) = \frac{1 + \eta}{2} \log(1 - \lambda(-1)) + \frac{1 - \eta}{2} \log(1 - \lambda(1))$$
$$= \frac{1 + \eta}{2} \log(1 + \lambda) + \frac{1 - \eta}{2} \log(1 - \lambda)$$

To find the maximum, we set the first derivative $\frac{dg}{d\lambda}$ to zero:

$$\frac{dg}{d\lambda} = \frac{1 + \eta}{2} \frac{1}{1 + \lambda} + \frac{1 - \eta}{2} \frac{-1}{1 - \lambda} = 0$$

Multiplying by $2(1 + \lambda)(1 - \lambda)$ and rearranging terms yields:

$$(1 + \eta)(1 - \lambda) = (1 - \eta)(1 + \lambda)$$

Expanding both sides:

$$1 - \lambda + \eta - \eta\lambda = 1 + \lambda - \eta - \eta\lambda$$

Canceling 1 and $-\eta\lambda$ from both sides:

$$-\lambda + \eta = \lambda - \eta$$

$$2\eta = 2\lambda$$

Thus, the log-optimal betting fraction is $\lambda^* = \eta$. Since the slack parameter $\eta$ is defined as $\eta \in (0, 1]$ ($\eta > 0$ can always be assumed to be $\leq 1$ since payoffs lie in $[0, 1]$), this solution lies in the required domain $\lambda \in (0, 1]$. $\qquad\square$

**Corollary C.4** (Corollary 2.5). *Let $\nu \in \Delta((0, 1])$ be any probability distribution on $(0, 1]$, and define the mixture process*

$$M_{i,t}^{(a_i')}(\nu) := \int_0^1 M_{i,t}^{(a_i')}(\lambda) \, d\nu(\lambda),$$

*where $M_{i,t}^{(a_i')}(\lambda)$ is the supermartingale defined in Equation (5). Then, under the null hypothesis $\mathcal{H}_0$, $(M_{i,t}^{(a_i')}(\nu))_{t \geq 0}$ is also a nonnegative supermartingale.*

*Proof.* The only nontrivial step is justifying that we can interchange the integral and conditional expectation. Since each $M_{i,t}^{(a_i')}(\lambda)$ is a nonnegative supermartingale, we have

$$\mathbb{E}[M_{i,t+1}^{(a_i')}(\lambda) \mid \mathcal{F}_t] \leq M_{i,t}^{(a_i')}(\lambda), \quad \forall \lambda \in [0, 1].$$

Integrating both sides with respect to $\nu$ and using Tonelli's theorem for nonnegative integrands, we obtain

$$\mathbb{E}[M_{i,t+1}^{(a_i')}(\nu) \mid \mathcal{F}_t] = \mathbb{E}\Big[\int_0^1 M_{i,t+1}^{(a_i')}(\lambda) \, d\nu(\lambda) \,\Big|\, \mathcal{F}_t\Big] \leq \int_0^1 M_{i,t}^{(a_i')}(\lambda) \, d\nu(\lambda) = M_{i,t}^{(a_i')}(\nu),$$

so $M_{i,t}^{(a_i')}(\nu)$ is a nonnegative supermartingale. $\qquad\square$

### C.3. FWER control

**Theorem C.5** (Theorem 2.6). *Let $M_{i,t}^{(a_i')}(\nu)$ be the supermartingales defined in Equation (6) for each player $i \in [n]$ and each deviation $a_i' \in A_i$, for some $\nu \in \Delta((0, 1])$. Then for any $\alpha \in (0, 1)$,*

$$\mathbb{P}_{\mathcal{H}_0}\left(\exists \, i, \exists \, a_i' : \sup_{t \geq 0} M_{i,t}^{(a_i')}(\nu) \geq \frac{\sum_{i=1}^n |A_i|}{\alpha}\right) \leq \alpha.$$

*Proof.* By Corollary 2.5, each $M_{i,t}^{(a_i')}(\nu)$ is a nonnegative supermartingale under $\mathcal{H}_0$. Ville's inequality implies that for any fixed $i$ and $a_i'$:

$$\mathbb{P}_{\mathcal{H}_0}\left(\sup_{t \geq 0} M_{i,t}^{(a_i')}(\nu) \geq \frac{\sum_{i=1}^n |A_i|}{\alpha}\right) \leq \frac{\alpha}{\sum_{i=1}^n |A_i|}.$$

Applying a union bound over all players $j \in [n]$ and all deviations $a_j' \in A_j$ gives

$$\mathbb{P}_{\mathcal{H}_0}\left(\exists i, \exists a_i' : \sup_{t \geq 0} M_{i,t}^{(a_i')}(\nu) \geq \frac{\sum_{i=1}^n |A_i|}{\alpha}\right) \leq \sum_{j=1}^n \sum_{a_j' \in A_j} \frac{\alpha}{\sum_{i=1}^n |A_i|} = \left(\sum_{j=1}^n \frac{|A_j|}{\sum_{i=1}^n |A_i|}\right) \alpha = \alpha.$$

$\square$

## C.4. Expected detection time under the alternative

**Theorem C.6** (Theorem 2.7). *For any $b > 1$ and mixture $\nu \in \Delta((0,1])$, let*

$$\tau_i^{(a_i')}(\nu) := \inf\{t \geq 0 : M_{i,t}^{(a_i')}(\nu) \geq b\}.$$

*If $\nu$ is the uniform distribution on $(0,1]$, i.e., $d\nu(\lambda) = d\lambda$, then:*

$$\mathbb{E}_{\mathcal{H}_1}[\tau_i^{(a_i')}(\nu)] \leq \frac{12(\log(b) + \log(4/\eta) + \log(3/2))}{\eta^2}.$$

*Proof.* In the proof, we will write $\tau$ instead of $\tau_i^{(a_i')}(\nu)$ for convenience.

The mixture supermartingale is defined as $M_{i,t}^{(a_i')}(\nu) = \int_0^1 M_{i,t}^{(a_i')}(\lambda)d\lambda$. We construct a lower bound process to control the stopping time. Define the interval $J := [\eta/4, \eta/2]$. Since $\eta \in (0,1]$, we have $J \subset [0, 1/2]$. The length of this interval is $|J| = \eta/4$.

By positivity, we lower bound the mixture integral by restricting it to $J$:

$$M_{i,t}^{(a_i')}(\nu) \geq \int_J M_{i,t}^{(a_i')}(\lambda)d\lambda = |J| \cdot \frac{1}{|J|} \int_J M_{i,t}^{(a_i')}(\lambda)d\lambda.$$

Since the logarithm is concave, Jensen's inequality implies:

$$\log M_{i,t}^{(a_i')}(\nu) \geq \log |J| + \frac{1}{|J|} \int_J \log M_{i,t}^{(a_i')}(\lambda)d\lambda.$$

Let $Y_t := \frac{1}{|J|} \int_J \log(1 - \lambda X_{i,t}^{(a_i')})d\lambda$ denote the average log-increment over $J$, and let $S_t := \sum_{s=1}^t Y_s$ (we omit the player and deviation subscripts for convenience). The inequality can be rewritten as:

$$\log M_{i,t}^{(a_i')}(\nu) \geq \log(\eta/4) + S_t.$$

We define a surrogate stopping time $\tau'$:

$$\tau' := \inf\{t \geq 0 : S_t \geq \log(b) - \log(\eta/4)\}.$$

If the condition for $\tau'$ is met, then $\log M_{i,t}^{(a_i')}(\nu) \geq \log(b)$, so $M_{i,t}^{(a_i')}(\nu) \geq b$ and $\tau \leq \tau'$. Thus, it suffices to bound $\mathbb{E}_{\mathcal{H}_1}[\tau']$.

**Step 1: Lower bounding the drift.** Let $\mu(\lambda) := \mathbb{E}_{\mathcal{H}_1}[\log(1 - \lambda X_t)]$. For any $x \in [-1, 1]$ and $\lambda \in (0, 1/2]$, Taylor's theorem with Lagrange remainder guarantees the existence of $\xi$ that lies strictly between $0$ and $\lambda x$ such that:

$$\log(1 - \lambda x) = -\lambda x - \frac{\lambda^2 x^2}{2(1-\xi)^2}.$$

Since $\lambda \in J \subset [0, 1/2]$ and $|x| \leq 1$, we have $|\lambda x| \leq 1/2$, which implies $|1 - \xi| \geq |1 - 1/2| = 1/2$. Therefore $\log(1 - \lambda x) \geq -\lambda x - 2\lambda^2 x^2$. Taking expectations under $\mathcal{H}_1$ (where $\mathbb{E}_{\mathcal{H}_1}[X_t] \leq -\eta$ and $\mathbb{E}_{\mathcal{H}_1}[X_t^2] \leq 1$ since $|X_t| \leq 1$):

$$\mu(\lambda) \geq -\lambda(-\eta) - 2\lambda^2(1) = \lambda\eta - 2\lambda^2.$$

We lower bound the average drift $\Delta := \mathbb{E}_{\mathcal{H}_1}[Y_t] = \frac{1}{|J|} \int_J \mu(\lambda) d\lambda$. Using the bound $\mu(\lambda) \geq \lambda\eta - 2\lambda^2$ for $\lambda \in [\eta/4, \eta/2]$:

$$\int_{\eta/4}^{\eta/2} (\lambda\eta - 2\lambda^2) d\lambda = \left[\frac{\eta\lambda^2}{2} - \frac{2\lambda^3}{3}\right]_{\eta/4}^{\eta/2} = \eta^3 \left(\left(\frac{1}{8} - \frac{1}{12}\right) - \left(\frac{1}{32} - \frac{1}{96}\right)\right) = \eta^3 \left(\frac{1}{24} - \frac{2}{96}\right) = \frac{\eta^3}{48}.$$

Dividing by the length $|J| = \eta/4$, the average drift is:

$$\Delta \geq \frac{\eta^3/48}{\eta/4} = \frac{\eta^2}{12}.$$

Thus, $\Delta \geq \eta^2/12$ for all $\eta \in (0, 1]$. Note that this drift is strictly positive.

**Step 2: Wald's identity.** Let $B := \log(b) - \log(\eta/4)$. The stopping time is $\tau' = \inf\{t \geq 0 : S_t \geq B\}$. The increments $Y_t$ are i.i.d. random variables with strictly positive mean $\Delta > 0$.

Consider the truncated stopping time $\tau' \wedge T$ for all $T \geq 1$, which is bounded and integrable. First, observe that $Y_t$ is bounded from above:

$$Y_t \leq \frac{1}{|J|} \int_J \log(3/2) d\lambda = \log(3/2).$$

Therefore, by definition of $\tau'$, we have $S_{\tau' \wedge T} = S_{(\tau' \wedge T)-1} + Y_{\tau' \wedge T} \leq B + \log(3/2)$.

Now, applying Wald's identity to the i.i.d. increments $Y_s$ gives

$$\mathbb{E}_{\mathcal{H}_1}[S_{\tau' \wedge T}] = \mathbb{E}_{\mathcal{H}_1}[\tau' \wedge T] \cdot \Delta,$$

so that

$$\mathbb{E}_{\mathcal{H}_1}[\tau' \wedge T] = \frac{\mathbb{E}_{\mathcal{H}_1}[S_{\tau' \wedge T}]}{\Delta} \leq \frac{B + \log(3/2)}{\Delta}.$$

Finally, taking the limit $T \to \infty$ and applying the monotone convergence theorem gives

$$\mathbb{E}_{\mathcal{H}_1}[\tau'] = \mathbb{E}_{\mathcal{H}_1}\left[\lim_{T \to \infty} \tau' \wedge T\right] = \lim_{T \to \infty} \mathbb{E}_{\mathcal{H}_1}[\tau' \wedge T] \leq \frac{B + \log(3/2)}{\Delta}.$$

Rearranging and using the fact that $\Delta \geq \eta^2/12$, we obtain:

$$\mathbb{E}_{\mathcal{H}_1}[\tau'] \leq \frac{12(\log(b) + \log(4/\eta) + \log(3/2))}{\eta^2}.$$

Since $\mathbb{E}_{\mathcal{H}_1}[\tau] \leq \mathbb{E}_{\mathcal{H}_1}[\tau']$, the result follows. $\qquad\square$

If the parameter $\eta$ is known, then the monitor can construct a mixture $\nu$ that depends explicitly on $\eta$, yielding an even tighter bound.

**Theorem C.7.** *Fix some threshold $b > 1$ and, for each $\nu \in \Delta((0, 1])$, define the stopping time*

$$\tau_i^{(a_i')}(\nu) := \inf\{t \geq 0 : M_{i,t}^{(a_i')}(\nu) \geq b\}.$$

*Then, there exists $\nu \in \Delta((0, 1])$ such that:*

$$\mathbb{E}_{\mathcal{H}_1}[\tau_i^{(a_i')}(\nu)] \leq \frac{9(\log(b) + \log(1 + \eta))}{\eta^2}.$$

*Proof.* Let $\lambda := \eta \wedge \frac{1}{4}$ and define $\nu := \delta_\lambda$, the probability measure that assigns mass 1 to $\lambda$.[1]

Define the function $f(x) := \log(1 - \lambda x)$ for $|x| \leq 1$, and set $Y_t := f(X_{i,t}^{(a_i')})$ for all $t \geq 1$.

First, we lower bound $\mathbb{E}_{\mathcal{H}_1}[Y_1]$. Using Taylor's theorem around $x = 0$ with Lagrange remainder, we have

$$f(x) = -\lambda x - \frac{\lambda^2 x^2}{2(1 - \lambda \xi)^2}, \quad \text{for some } \xi \in (0, x).$$

Since $\lambda \leq 1/4$ and $|x| \leq 1$, we have $1 - \lambda \xi \geq 3/4$, which implies

$$f(x) \geq -\lambda x - \frac{8}{9}\lambda^2.$$

Taking expectation under $\mathcal{H}_1$ gives

$$\mathbb{E}_{\mathcal{H}_1}[Y_1] = \mathbb{E}_{\mathcal{H}_1}[f(X_{i,1}^{(a_i')})] \geq -\lambda \mathbb{E}_{\mathcal{H}_1}[X_{i,1}^{(a_i')}] - \frac{8}{9}\lambda^2 \geq \frac{\eta^2}{9}.$$

Next, define the truncated stopping time $\tau_{i,T}^{(a_i')}(\nu) := \tau_i^{(a_i')}(\nu) \wedge T$ for all $T \geq 1$, which is bounded and integrable. Let $S_t := \sum_{s=1}^t Y_s$, so that

$$\tau_i^{(a_i')}(\nu) = \inf\{t \geq 0 : S_t \geq \log(b)\}.$$

By definition of $\tau_i^{(a_i')}(\nu)$, we have $S_{\tau_{i,T}^{(a_i')}(\nu)-1} < \log(b)$. Moreover, $Y_{\tau_{i,T}^{(a_i')}(\nu)} \leq \log(1 + \eta)$ because $\lambda \leq \eta$, which implies

$$S_{\tau_{i,T}^{(a_i')}(\nu)} \leq \log(b) + \log(1 + \eta).$$

Applying Wald's identity to the i.i.d. increments $Y_s$ gives

$$\mathbb{E}_{\mathcal{H}_1}\left[S_{\tau_{i,T}^{(a_i')}(\nu)}\right] = \mathbb{E}_{\mathcal{H}_1}[\tau_{i,T}^{(a_i')}(\nu)] \cdot \mathbb{E}_{\mathcal{H}_1}[Y_1],$$

so that

$$\mathbb{E}_{\mathcal{H}_1}[\tau_{i,T}^{(a_i')}(\nu)] = \frac{\mathbb{E}_{\mathcal{H}_1}\left[S_{\tau_{i,T}^{(a_i')}(\nu)}\right]}{\mathbb{E}_{\mathcal{H}_1}[Y_1]} \leq \frac{9(\log(b) + \log(1 + \eta))}{\eta^2}.$$

Finally, taking the limit $T \to \infty$ and applying the monotone convergence theorem gives

$$\mathbb{E}_{\mathcal{H}_1}[\tau_i^{(a_i')}(\nu)] = \mathbb{E}_{\mathcal{H}_1}\left[\lim_{T \to \infty} \tau_{i,T}^{(a_i')}(\nu)\right] = \lim_{T \to \infty} \mathbb{E}_{\mathcal{H}_1}[\tau_{i,T}^{(a_i')}(\nu)] \leq \frac{9(\log(b) + \log(1 + \eta))}{\eta^2},$$

which concludes the proof. $\qquad\square$

### C.5. False discovery rate control

**Theorem C.8** (Theorem 2.8)**.** *Let $\tau^{\mathrm{FDR}}(\nu)$ be the global stopping time at which the procedure first raises an alarm defined in Equation* (10)*. Then the rejection set $R_{\tau^{\mathrm{FDR}}(\nu)}(\nu)$ produced by Algorithm 2 satisfies*

$$\mathbb{E}\left[\frac{|R_{\tau^{\mathrm{FDR}}(\nu)}(\nu) \cap \mathcal{H}_0|}{1 \vee |R_{\tau^{\mathrm{FDR}}(\nu)}(\nu)|}\right] \leq \alpha,$$

*where $\mathcal{H}_0$ is the set of true nulls.*

---

[1]We choose $\lambda = \eta \wedge 1/4$ to obtain a simple constant in the lower bound on $\mathbb{E}_{\mathcal{H}_1}[Y_1]$. This choice is not optimal: the optimal $\lambda^*$ would maximize $\lambda \eta - \frac{\lambda^2}{2(1-\lambda)^2}$, but there is no simple closed form, so we select $\lambda = \eta \wedge 1/4$ for clarity.

*Proof.* Write $\tau := \tau^{\mathrm{FDR}}(\nu)$ and $R := R_\tau(\nu)$. Assume $|\mathcal{H}_0| > 0$, the case $|\mathcal{H}_0| = 0$ being trivial.

Fix $(i, a_i') \in \mathcal{H}_0$. By Corollary 2.5, $(M_{i,t}^{(a_i')}(\nu))_{t \geq 0}$ is a nonnegative supermartingale with $M_{i,0}^{(a_i')}(\nu) = 1$. For any $T \geq 0$, optional stopping at the bounded time $\tau \wedge T$ and Fatou's lemma give

$$\mathbb{E}\big[M_{i,\tau}^{(a_i')}(\nu)\big] \leq \liminf_{T \to \infty} \mathbb{E}\big[M_{i,\tau \wedge T}^{(a_i')}(\nu)\big] \leq 1.$$

If $k_\tau = 0$ then $R = \varnothing$ and the ratio is zero. Otherwise $k_\tau \geq 1$, and by definition of $R$, for $(i, a_i') \in R$,

$$M_{i,\tau}^{(a_i')}(\nu) \geq \frac{1}{k_\tau \, \alpha \, \gamma_{i,a_i'}},$$

which implies that

$$\mathbb{1}\{(i, a_i') \in R\} \leq k_\tau \, \alpha \, \gamma_{i,a_i'} \, M_{i,\tau}^{(a_i')}(\nu).$$

Since $|R| \geq k_\tau$,

$$\frac{|R \cap \mathcal{H}_0|}{1 \vee |R|} \leq \frac{1}{k_\tau} \sum_{(i,a_i') \in \mathcal{H}_0} \mathbb{1}\{(i, a_i') \in R\} \leq \alpha \sum_{(i,a_i') \in \mathcal{H}_0} \gamma_{i,a_i'} \, M_{i,\tau}^{(a_i')}(\nu).$$

Taking expectations, using $\mathbb{E}[M_{i,\tau}^{(a_i')}(\nu)] \leq 1$ and $\sum_{(i,a_i')} \gamma_{i,a_i'} = 1$,

$$\mathbb{E}\left[\frac{|R \cap \mathcal{H}_0|}{1 \vee |R|}\right] \leq \alpha \sum_{(i,a_i') \in \mathcal{H}_0} \gamma_{i,a_i'} \leq \alpha. \qquad \square$$

**Proposition C.9** (Proposition 2.9). *Let $\tau_i^{(a_i')}(\nu)$ be the FWER detection time defined in Equation (7) with threshold $b = \sum_{i=1}^n |A_i|/\alpha$ and any mixture $\nu \in \Delta((0,1])$. Consider the FDR procedure with uniform weights $\gamma_{i,a_i'} = 1/m$. Then, almost surely,*

$$\tau^{\mathrm{FDR}}(\nu) \leq \tau_i^{(a_i')}(\nu).$$

*In other words, when using uniform weights, the FDR procedure flags a deviation at least as early as the conservative FWER procedure.*

*Proof.* Write $\tau_i := \tau_i^{(a_i')}(\nu)$ for simplicity. Let $(i, a_i')$ be such that $M_{i,\tau_i}^{(a_i')}(\nu) \geq m/\alpha$. Under uniform weights $\gamma_{i,a_i'} = 1/m$, this is equivalent to $M_{i,\tau_i}^{(a_i')}(\nu) \geq 1/(\alpha \gamma_{i,a_i'})$. Hence, in the e-BH procedure, at time $\tau_i$ we must have $k_{\tau_i} \geq 1$ (since at least one hypothesis exceeds its threshold). The e-BH threshold for any hypothesis $(i, a_i')$ is $1/(k_{\tau_i} \alpha \gamma_{i,a_i'}) \leq 1/(\alpha \gamma_{i,a_i'})$. Therefore, $(i, a_i')$ also exceeds the e-BH threshold at time $\tau_i$, meaning that the FDR procedure would have rejected it no later than the union-bound procedure. Consequently, $\tau^{\mathrm{FDR}}(\nu) \leq \tau_i$. $\qquad \square$

# D. Extension of Section 2 to Alternative Equilibria

## D.1. Alternative equilibrium concepts

While the main text focused on Nash equilibria, many strategic settings are naturally captured by broader equilibrium concepts. These equilibria allow for a richer set of strategic behaviors by permitting dependencies between players' actions.

**Definition D.1** (Correlated Equilibrium). A joint strategy profile $\pi$ is a *correlated equilibrium* if, for each player $i \in [n]$, for all actions $a_i \in A_i$, and all alternative actions $a_i' \in A_i$, we have

$$\mathbb{E}_{a \sim \pi}[u_i(a_i, a_{-i})|a_i] \geq \mathbb{E}_{a \sim \pi}[u_i(a_i', a_{-i})|a_i].$$

The interpretation is that given a recommended action, no player can improve their expected payoff by unilaterally deviating.

Correlated equilibria are generally easier to compute than Nash equilibria, because the joint distribution $\pi$ can be found by solving a linear feasibility problem rather than searching for a fixed point of best responses. An even more tractable concept is the coarse correlated equilibrium, which relaxes the conditional deviation requirement: players only need to consider unconditional deviations, ignoring the recommended action.

**Definition D.2** (Coarse Correlated Equilibrium). A joint strategy profile $\pi$ is a *coarse correlated equilibrium* if, for each player $i \in [n]$ and all alternative actions $a_i' \in A_i$, we have

$$\mathbb{E}_{a \sim \pi}[u_i(a_i, a_{-i})] \geq \mathbb{E}_{a \sim \pi}[u_i(a_i', a_{-i})].$$

Definition D.2 closely resembles that of Definition 2.1, with the key difference that the joint distribution is not required to factor as a product of independent strategies.

It is straightforward to see that every Nash equilibrium is also a correlated equilibrium: when the joint distribution is a product of independent strategies, the conditional expectations in Definition D.1 reduce to the unconditional expectations used in Definition 2.1. Moreover, every correlated equilibrium is also a coarse correlated equilibrium, as the law of total expectation shows that the conditional no-deviation requirement in Definition D.1 implies the unconditional no-deviation requirement of Definition D.2.

In many practical or computational settings, it is natural to allow for approximate rather than exact equilibrium conditions. Players may tolerate small deviations from optimality, or equilibrium strategies may be computed only approximately due to stochasticity or bounded rationality. This leads to the notion of an $\varepsilon$-equilibrium, where each player's incentive to deviate is bounded by a small parameter $\varepsilon > 0$:

**Definition D.3** ($\varepsilon$-Approximate Nash Equilibrium). A joint strategy profile $\pi = \pi_1 \times \cdots \times \pi_n$ is an $\varepsilon$-*Nash equilibrium* if, for each player $i \in [n]$ and all alternative actions $a_i' \in A_i$, we have

$$\mathbb{E}_{a \sim \pi}[u_i(a_i, a_{-i})] \geq \mathbb{E}_{a \sim \pi}[u_i(a_i', a_{-i})] - \varepsilon.$$

The parameter $\varepsilon$ quantifies the maximum incentive any player has to deviate: when $\varepsilon = 0$, the definition reduces to the standard Nash equilibrium. Analogous definitions apply to $\varepsilon$-correlated and $\varepsilon$-coarse correlated equilibria, obtained by relaxing the corresponding inequalities in Definitions D.1 and D.2. These approximate notions are useful when equilibria are learned empirically, computed via iterative dynamics, or estimated from noisy data, in contexts where exact equilibrium conditions are rarely satisfied but small deviations are acceptable.

### D.2. Extending the framework beyond Nash

The framework presented in Section 2 focuses on Nash equilibria, though it can be extended naturally to broader equilibrium notions. First, it is easy to see that the same methodology applies directly to coarse correlated equilibria. Second, the framework also covers correlated equilibria. Formally, at each time step $t$, we can model the recommendation $a_{i,t}$ as being drawn first, followed by the realization of the remaining actions $a_{-i,t}$. By working with the enlarged filtration $\mathcal{G}_{t-1} := \sigma(\mathcal{F}_{t-1}, a_{i,t})$, the same sequential testing arguments and FDR guarantees apply in this setting as well. In particular, this refinement yields an expected detection delay under the alternative satisfying $O(\log(\sum_{i=1}^{n} (|A_i|(|A_i| - 1))/\alpha)) = O(\log(\sum_{i=1}^{n} |A_i|/\alpha))$ which constitutes a strict improvement over the $O(\log(\sum_{i=1}^{n} |A_i|^{|A_i|}/\alpha))$ bound obtained by Babichenko et al. (2017). Finally, the approach extends to $\varepsilon$-approximate equilibria. In that case, it suffices to shift the per-round regret-like increments by $\varepsilon$, that is,

$$X_{i,t}^{(\varepsilon)} = u_i(a_{i,t}, a_{-i,t}) - u_i(a_i', a_{-i,t}) + \varepsilon,$$

so that the resulting supermartingales remain valid under the $\varepsilon$-equilibrium hypothesis.

## E. Role of the Slack Parameter $\eta$ in Section 2

In our sequential testing framework, the alternative hypothesis $\mathcal{H}_1$ is defined with respect to a fixed slack parameter $\eta > 0$. This parameter quantifies the minimal payoff gain that constitutes a detectable deviation from the equilibrium. Specifically, a deviation $a_i'$ by player $i$ is considered profitable if it increases the expected payoff by at least $\eta$:

$$\mathbb{E}[u_i(a_i', a_{-i})] - \mathbb{E}[u_i(a_i, a_{-i})] \geq \eta.$$

Smaller values of $\eta$ correspond to detecting finer deviations from equilibrium, but require more rounds of observation to confidently reject $\mathcal{H}_0$. Conversely, larger values of $\eta$ lead to faster detection of strong deviations.

To illustrate the necessity of a positive slack parameter $\eta > 0$, we provide a simple deterministic two-player example where the stopping time can be arbitrarily large if $\eta$ is very small.

**Proposition E.1.** *Consider a 2-player game with action sets $A_1 = A_2 = \{a, b\}$, where the observed action profile is always $(a, a)$, and the payoffs are given by the following matrices:*

$$U_1 = \begin{bmatrix} 0.5 & 0.5 \\ 0.5 + \eta & 0.5 \end{bmatrix}, \qquad U_2 = \begin{bmatrix} 0.5 & 0.5 \\ 0.5 & 0.5 \end{bmatrix}.$$

*The deviation payoff difference for player 1 is*

$$X_{1,t}^{(b)} := u_1(a, a) - u_1(b, a) = 0.5 - (0.5 + \eta) = -\eta < 0,$$

*and the log-increment for a fixed $\lambda \in (0, 1)$ is*

$$Y_t = \log(1 - \lambda X_{1,t}^{(b)}) = \log(1 + \lambda \eta) > 0.$$

*The sequential stopping time for threshold $b > 1$ is*

$$\tau_\lambda = \inf \left\{ t \geq 1 : \sum_{s=1}^{t} Y_s \geq \log(b) \right\}.$$

*Then, the stopping time satisfies the deterministic lower bound*

$$\tau_\lambda \geq \frac{\log(b)}{\log(1 + \lambda \eta)}.$$

*In particular, as $\eta \to 0$, $\tau_\lambda \to \infty$, showing that arbitrarily small deviations can lead to arbitrarily large stopping times.*

*Proof.* Since the payoff differences $X_{1,t}^{(b)} = -\eta$ are deterministic and identical across rounds, the log-increments $Y_t = \log(1 + \lambda \eta)$ are also constant. Therefore, the cumulative log process grows linearly:

$$\sum_{s=1}^{t} Y_s = t \cdot \log(1 + \lambda \eta).$$

The stopping time is the first $t$ such that this sum reaches the threshold $\log(b)$:

$$\tau_\lambda = \inf \{t \geq 1 : t \cdot \log(1 + \lambda \eta) \geq \log(b)\} = \left\lceil \frac{\log(b)}{\log(1 + \lambda \eta)} \right\rceil \geq \frac{\log(b)}{\log(1 + \lambda \eta)}.$$

This shows that the expected stopping time grows unboundedly as $\eta \to 0$, illustrating the necessity of a strictly positive slack parameter. $\square$

This deterministic example demonstrates that even when a deviation exists ($\mathcal{H}_1$ is true), if the minimal payoff improvement $\eta$ is extremely small, the sequential test may require an arbitrarily large number of rounds to reject the null. Introducing a positive $\eta$ guarantees a minimal expected growth of the log cumulative process and ensures a finite expected stopping time, as formalized in Theorem 2.7.

# F. Proofs for Section 3

## F.1. Sequential testing via likelihood ratios

**Lemma F.1** (Lemma 3.1). *For any player $i$, we have*

$$\mathbb{E}_{\mathcal{H}_0}[E_{i,t} \mid \mathcal{F}_{t-1}] = 1,$$

*where $\mathcal{F}_{t-1}$ is the filtration generated by all states and actions up to round $t-1$. Hence, $E_{i,t}$ is a valid e-value conditional on the history under the null hypothesis $\mathcal{H}_0$.*

*Proof.* By the law of total expectation, we first condition on the state $s_t$:

$$\mathbb{E}_{\mathcal{H}_0}[E_{i,t} \mid \mathcal{F}_{t-1}] = \sum_{s \in \mathcal{S}} \mathbb{P}_{\mathcal{H}_0}(s_t = s \mid \mathcal{F}_{t-1}) \, \mathbb{E}_{\mathcal{H}_0}[E_{i,t} \mid \mathcal{F}_{t-1}, s_t = s].$$

Given that $s_t = s$, the action $a_{i,t}$ is drawn according to $\pi_i(\cdot \mid s)$ under $\mathcal{H}_0$, so

$$\mathbb{E}_{\mathcal{H}_0}[E_{i,t} \mid \mathcal{F}_{t-1}, s_t = s] = \sum_{a_i \in A_i} \pi_i(a_i \mid s) \frac{\pi_i^{(1)}(a_i \mid s)}{\pi_i(a_i \mid s)} = \sum_{a_i \in A_i} \pi_i^{(1)}(a_i \mid s) = 1.$$

Plugging this back, we get

$$\mathbb{E}_{\mathcal{H}_0}[E_{i,t} \mid \mathcal{F}_{t-1}] = \sum_{s \in \mathcal{S}} \mathbb{P}_{\mathcal{H}_0}(s_t = s \mid \mathcal{F}_{t-1}) \cdot 1 = 1.$$

Hence, $E_{i,t}$ is a valid e-value conditional on the history. $\qquad\square$

### F.2. (Super)martingale construction

**Lemma F.2** (Lemma 3.2). *Under $\mathcal{H}_0$, the sequence $(M_{i,t})_{t \geq 0}$ defined in Equation (12) is a nonnegative martingale with respect to the filtration $(\mathcal{F}_t)_{t \geq 0}$.*

*Proof.* By Lemma 3.1,

$$\mathbb{E}_{\mathcal{H}_0}[M_{i,t} \mid \mathcal{F}_{t-1}] = M_{i,t-1} \cdot \mathbb{E}_{\mathcal{H}_0}[E_{i,t} \mid \mathcal{F}_{t-1}] = M_{i,t-1}.$$

Hence $(M_{i,t})_{t \geq 0}$ is a nonnegative martingale. $\qquad\square$

### F.3. FWER control

**Theorem F.3** (Theorem 3.3). *For any threshold $b > 1$, we have:*

$$\mathbb{P}_{\mathcal{H}_0}\Big(\exists i \in [n] : \sup_{t \geq 0} M_{i,t} \geq b\Big) \leq \frac{n}{b}.$$

*In particular, setting $b = n/\alpha$ ensures that the FWER is controlled at level $\alpha$.*

*Proof.* By Lemma 3.2, each $(M_{i,t})_{t \geq 0}$ is a nonnegative supermartingale with $M_{i,0} = 1$ under $\mathcal{H}_0$. Applying Ville's inequality to each player $i$ and then taking a union bound over $i \in [n]$ gives

$$\mathbb{P}_{\mathcal{H}_0}\Big(\exists i \in [n] : \sup_{t \geq 0} M_{i,t} \geq b\Big) \leq \sum_{i=1}^{n} \frac{1}{b} = \frac{n}{b}.$$

$\qquad\square$

### F.4. Expected detection time under the alternative

**Theorem F.4** (Theorem 3.4). *Suppose that the Markov chain induced by $\pi^{(1)}$ is ergodic, with unique stationary distribution $\mu \in \Delta(\mathcal{S})$, and that the process is started in stationarity, i.e., $s_0 \sim \mu$. Assume the support condition $\pi_i(a \mid s) > 0$ whenever $\pi_i^{(1)}(a \mid s) > 0$ holds for all $s \in \mathcal{S}$ and $a \in A_i$. Then, we have*

$$\mathbb{E}_{\mathcal{H}_1}[\tau_i] \leq \frac{\log b + C_i}{\bar{\mathrm{KL}}_i(\mu)},$$

*where $C_i$ is the bounded overshoot constant for player $i$:*

$$C_i := \max_{s \in \mathcal{S}} \max_{a \in A_i} \left| \log \frac{\pi_i^{(1)}(a \mid s)}{\pi_i(a \mid s)} \right| < \infty.$$

*Proof.* Because $\pi_i^{(1)}(a \mid s) > 0$ implies $\pi_i(a \mid s) > 0$ and the state/action spaces are finite, $C_i$ is finite and the log-ratio $Y_{i,t} := \log \frac{\pi_i^{(1)}(a_{i,t}\mid s_t)}{\pi_i(a_{i,t}\mid s_t)}$ is almost surely finite and uniformly bounded:

$$|Y_{i,t}| \leq C_i \quad \text{a.s.}$$

In particular $Y_{i,t} \leq C_i$ a.s. Under the alternative and stationarity, the conditional marginal $\mathbb{P}_{\mathcal{H}_1}(s_t = s \mid \mathcal{F}_{t-1}) = \mu(s)$ for every history. Thus, for any $t \geq 1$,

$$\mathbb{E}_{\mathcal{H}_1}[Y_{i,t} \mid \mathcal{F}_{t-1}] = \sum_{s\in\mathcal{S}} \mathbb{P}_{\mathcal{H}_1}(s_t = s \mid \mathcal{F}_{t-1})\mathbb{E}_{\mathcal{H}_1}[Y_{i,t} \mid \mathcal{F}_{t-1}, s_t = s]$$

$$= \sum_{s\in\mathcal{S}} \mathbb{P}_{\mathcal{H}_1}(s_t = s \mid \mathcal{F}_{t-1}) \sum_{a\in A_i} \pi_i^{(1)}(a \mid s) \log \frac{\pi_i^{(1)}(a \mid s)}{\pi_i(a \mid s)}$$

$$= \sum_{s\in\mathcal{S}} \mu(s)\,\mathrm{KL}_i(s)$$

$$= \bar{\mathrm{KL}}_i(\mu).$$

Hence the process $Z_t := S_{i,t} - \bar{\mathrm{KL}}_i(\mu)\, t$ satisfies

$$\mathbb{E}_{\mathcal{H}_1}[Z_t \mid \mathcal{F}_{t-1}] = Z_{t-1},$$

i.e. $(Z_t)_{t\geq 0}$ is a martingale with $Z_0 = 0$. Fix $T \geq 1$ and define the truncated stopping time $\tau_{i,T} := \tau_i \wedge T$. This stopping time is bounded. By definition of $\tau_i$ we have

$$S_{i,\tau_{i,T}} \leq \log b + C_i \quad \text{a.s.,}$$

since $\tau_{i,T} \leq \tau_i$. The optional stopping theorem applied to the martingale $Z_t$ and the bounded stopping time $\tau_{i,T}$ yields:

$$\mathbb{E}_{\mathcal{H}_1}[Z_{\tau_{i,T}}] = 0$$

since $Z_0 = 0$. Expanding $Z_{\tau_{i,T}}$ gives

$$0 = \mathbb{E}_{\mathcal{H}_1}[S_{i,\tau_{i,T}}] - \bar{\mathrm{KL}}_i(\mu)\, \mathbb{E}_{\mathcal{H}_1}[\tau_{i,T}].$$

Rearranging and using the a.s. bound on $S_{i,\tau_{i,T}}$ yields

$$\mathbb{E}_{\mathcal{H}_1}[\tau_{i,T}] = \frac{\mathbb{E}_{\mathcal{H}_1}[S_{i,\tau_{i,T}}]}{\bar{\mathrm{KL}}_i(\mu)} \leq \frac{\log b + C_i}{\bar{\mathrm{KL}}_i(\mu)}.$$

Finally let $T \to \infty$. By monotone convergence, we may pass the limit inside the expectation:

$$\mathbb{E}_{\mathcal{H}_1}[\tau_i] = \mathbb{E}_{\mathcal{H}_1}\Big[ \lim_{T\to\infty} \tau_{i,T}\Big] = \lim_{T\to\infty} \mathbb{E}_{\mathcal{H}_1}[\tau_{i,T}] \leq \frac{\log b + C_i}{\bar{\mathrm{KL}}_i(\mu)},$$

which proves the claim. $\qquad\square$

### F.5. Mixture detection time

Firstly, we note that according to Tonelli's theorem for non-negative integrands, the process defined in Equation (16) is indeed a martingale, and thus it successfully controls the FWER under the null hypothesis. We will now show that under the alternative, the detection time scales the same way with a mixture as without a mixture, as explained in Remark 3.5.

#### F.5.1. FINITE CASE

**Theorem F.5.** *Consider a finite set of alternative strategies* $\mathcal{Q} = \{\pi^{(1)}, \ldots, \pi^{(K)}\}$ *and a prior distribution* $\nu$ *assigning probability* $w_k > 0$ *to each* $\pi^{(k)}$. *Let* $M_{i,t}(\nu) = \sum_{k=1}^{K} w_k M_{i,t}^{(k)}$ *be the mixture martingale, and let* $\tau_i(\nu) = \inf\{t \geq 1 : M_{i,t}(\nu) \geq b\}$ *be the detection time.*

*Assume the true generating strategy is $\pi^{(k^*)} \in \mathcal{Q}$ for some index $k^*$. Under the same regularity conditions as Theorem 3.4 (ergodicity and stationarity), we have:*

$$\mathbb{E}_{\pi^{(k^*)}}[\tau_i(\nu)] \leq \frac{\log b + \log(1/w_{k^*}) + C_{i,k^*}}{\bar{\mathrm{KL}}_i(\mu_{k^*})},$$

*where $C_{i,k^*}$ is the overshoot constant associated with the specific alternative $\pi^{(k^*)}$, and $\bar{\mathrm{KL}}_i(\mu_{k^*})$ is the KL divergence averaged over the stationary distribution $\mu_{k^*}$ of $\pi^{(k^*)}$.*

*Proof.* By definition, the mixture martingale is a sum of non-negative terms (since likelihood ratios are non-negative). Therefore, at any time $t$, the mixture value is lower-bounded by the contribution of the true alternative $\pi^{(k^*)}$:

$$M_{i,t}(\nu) = \sum_{k=1}^{K} w_k M_{i,t}^{(k)} \geq w_{k^*} M_{i,t}^{(k^*)}.$$

The stopping condition for the mixture is $M_{i,t}(\nu) \geq b$. Based on the inequality above, a sufficient condition for the mixture to cross $b$ is for the single component $w_{k^*} M_{i,t}^{(k^*)}$ to cross $b$. That is:

$$w_{k^*} M_{i,t}^{(k^*)} \geq b \iff M_{i,t}^{(k^*)} \geq \frac{b}{w_{k^*}}.$$

Let $\tau^*$ be the stopping time for the single martingale $M_{i,t}^{(k^*)}$ with the adjusted threshold $b' = b/w_{k^*}$. Since $M_{i,t}^{(k^*)} \geq b/w_{k^*}$ implies $M_{i,t}(\nu) \geq b$, it follows almost surely that:

$$\tau_i(\nu) \leq \tau^*.$$

We can now apply the bound from Theorem 3.4 directly to $\tau^*$:

$$\mathbb{E}_{\pi^{(k^*)}}[\tau^*] \leq \frac{\log(b/w_{k^*}) + C_{i,k^*}}{\bar{\mathrm{KL}}_i(\mu_{k^*})} = \frac{\log b + \log(1/w_{k^*}) + C_{i,k^*}}{\bar{\mathrm{KL}}_i(\mu_{k^*})}.$$

Since $\mathbb{E}_{\pi^{(k^*)}}[\tau_i(\nu)] \leq \mathbb{E}_{\pi^{(k^*)}}[\tau^*]$, the result follows. $\qquad\square$

### F.5.2. CONTINUOUS CASE

In this setting, we assume the mixture is defined over a compact metric space of strategies $\Theta$ equipped with a prior probability measure $\nu$. Let $\theta^*$ denote the parameter of the true generating strategy.

**Assumption F.6.** We assume that:

1. **Continuity of Drift:** Let $D(\theta^* \| \theta)$ denote the expected log-likelihood ratio of strategy $\pi_i^{(\theta)}$ when the data is generated by $\theta^*$:
$$D(\theta^* \| \theta) := \mathbb{E}_{s \sim \mu_{\theta^*}}\left[ \sum_{a_i \in A_i} \pi_i^{(\theta^*)}(a_i|s) \log \frac{\pi_i^{(\theta)}(a_i|s)}{\pi_i(a_i|s)} \right].$$
   We assume the mapping $\theta \mapsto D(\theta^* \| \theta)$ is continuous at $\theta^*$. Specifically, for any $\epsilon > 0$, there exists $\delta > 0$ such that for all $\theta \in B_\delta(\theta^*)$, we have $D(\theta^* \| \theta) \geq \bar{\mathrm{KL}}_i(\mu_{\theta^*}) - \epsilon$.

2. **Prior Mass:** The prior $\nu$ assigns positive mass to balls around the truth: $w_\delta := \nu(B_\delta(\theta^*)) > 0$ for all $\delta > 0$.

3. **Bounded Likelihood Ratios:** The log-likelihood ratios are locally bounded. That is, there exists a neighborhood $B_{\delta_0}(\theta^*)$ such that:
$$\sup_{\theta \in B_{\delta_0}(\theta^*)} \max_{s \in \mathcal{S}, a_i \in A_i} \left| \log \frac{\pi_i^{(\theta)}(a_i \mid s)}{\pi_i(a_i \mid s)} \right| < \infty.$$

   (This condition is satisfied automatically if $\pi_i(a_i|s) > 0$ everywhere and the mapping $\theta \mapsto \pi_i^{(\theta)}$ is continuous).

**Theorem F.7.** *Let $M_{i,t}(\nu) = \int_\Theta M_{i,t}(\theta)\, d\nu(\theta)$ be the mixture martingale. Suppose that the Markov chain induced by the true parameter $\theta^*$ is ergodic with unique stationary distribution $\mu_{\theta^*}$, and that the process is started in stationarity. For any $\epsilon \in (0, \bar{\mathrm{KL}}_i(\mu_{\theta^*}))$, let $\delta \leq \delta_0$ be the radius satisfying Assumption F.6.1, and let $w_\delta = \nu(B_\delta(\theta^*))$.*

*The expected detection time $\tau_i(\nu) = \inf\{t \geq 1 : M_{i,t}(\nu) \geq b\}$ satisfies:*

$$\mathbb{E}_{\theta^*}[\tau_i(\nu)] \leq \frac{\log b + \log(1/w_\delta) + C_{i,\delta}}{\bar{\mathrm{KL}}_i(\mu_{\theta^*}) - \epsilon},$$

*where $C_{i,\delta} := \sup_{\theta \in B_\delta(\theta^*)} \max_{s \in \mathcal{S}} \max_{a_i \in A_i} \left| \log \frac{\pi_i^{(\theta)}(a_i|s)}{\pi_i(a_i|s)} \right| < \infty.$*

*Proof.* Since the likelihood ratios are non-negative, restricting the integral to the ball $B_\delta(\theta^*)$ yields a lower bound. Let $\nu_\delta$ be the conditional prior restricted to this ball, defined by $d\nu_\delta(\theta) = w_\delta^{-1} \mathbb{I}(\theta \in B_\delta) d\nu(\theta)$. Then:

$$M_{i,t}(\nu) \geq \int_{B_\delta(\theta^*)} M_{i,t}(\theta)\, d\nu(\theta) = w_\delta \int_{B_\delta(\theta^*)} M_{i,t}(\theta)\, d\nu_\delta(\theta).$$

Let $\tau_{local}$ be the stopping time for the restricted mixture martingale $M_{i,t}^{loc} := \int_{B_\delta} M_{i,t}(\theta) d\nu_\delta(\theta)$ with threshold $b' = b/w_\delta$. The inequality above implies that if $M_{i,t}^{loc} \geq b'$, then $M_{i,t}(\nu) \geq b$. Thus, almost surely:

$$\tau_i(\nu) \leq \tau_{local}.$$

We analyze the log-process of the restricted mixture. Since the logarithm is concave, Jensen's inequality applied to the probability measure $\nu_\delta$ yields:

$$\log M_{i,t}^{loc} = \log \left( \int_{B_\delta(\theta^*)} M_{i,t}(\theta)\, d\nu_\delta(\theta) \right) \geq \int_{B_\delta(\theta^*)} \log M_{i,t}(\theta)\, d\nu_\delta(\theta).$$

Let $L_t$ denote this lower bounding process on the RHS. By linearity of the integral, we can write $L_t$ as a sum of average log-increments:

$$L_t = \sum_{s=1}^{t} \bar{Y}_{i,s}, \quad \text{where } \bar{Y}_{i,s} := \int_{B_\delta(\theta^*)} \log \left( \frac{\pi_i^{(\theta)}(a_{i,s}|s_s)}{\pi_i(a_{i,s}|s_s)} \right) d\nu_\delta(\theta).$$

We examine the expected drift of the increment $\bar{Y}_{i,t}$ under the true generating process $\theta^*$ and stationarity (marginal distribution $\mu_{\theta^*}$). By Tonelli's theorem:

$$\mathbb{E}_{\theta^*}[\bar{Y}_{i,t} \mid \mathcal{F}_{t-1}] = \int_{B_\delta(\theta^*)} \mathbb{E}_{\theta^*} \left[ \log \frac{\pi_i^{(\theta)}(a_{i,t}|s_t)}{\pi_i(a_{i,t}|s_t)} \,\middle|\, \mathcal{F}_{t-1} \right] d\nu_\delta(\theta).$$

Using the stationarity of $\theta^*$, the inner expectation is exactly the drift function $D(\theta^*\|\theta)$ defined in Assumption F.6.1. Note that $D(\theta^*\|\theta^*) = \bar{\mathrm{KL}}_i(\mu_{\theta^*})$. By the continuity condition from Assumption F.6.1, for all $\theta \in B_\delta(\theta^*)$, we have $D(\theta^*\|\theta) \geq \bar{\mathrm{KL}}_i(\mu_{\theta^*}) - \epsilon$. Therefore:

$$\mathbb{E}_{\theta^*}[\bar{Y}_{i,t} \mid \mathcal{F}_{t-1}] \geq \int_{B_\delta(\theta^*)} (\bar{\mathrm{KL}}_i(\mu_{\theta^*}) - \epsilon)\, d\nu_\delta(\theta) = \bar{\mathrm{KL}}_i(\mu_{\theta^*}) - \epsilon.$$

Let $\Delta = \bar{\mathrm{KL}}_i(\mu_{\theta^*}) - \epsilon$. We have shown that $Z_t := L_t - t\Delta$ is a submartingale.

Now, define a new stopping time $\tau_L$ based on the lower bound process $L_t$:

$$\tau_L := \inf \{t \geq 1 : L_t \geq \log(b/w_\delta)\}.$$

Recall from our application of Jensen's inequality that $\log M_{i,t}^{loc} \geq L_t$. Therefore, whenever $L_t$ crosses the threshold $\log(b/w_\delta)$, the log-mixture $\log M_{i,t}^{loc}$ must also have crossed it. This implies that $\tau_{local} \leq \tau_L$ almost surely.

It remains to bound $\mathbb{E}_{\theta^*}[\tau_L]$. Since $Z_t$ is a submartingale, we can apply the optional stopping theorem. For a truncated stopping time $\tau_L \wedge T$, we have $\mathbb{E}[Z_{\tau_L \wedge T}] \geq \mathbb{E}[Z_0] = 0$, which implies:

$$\mathbb{E}[L_{\tau_L \wedge T}] \geq \Delta \mathbb{E}[\tau_L \wedge T].$$

Since $\bar{Y}_{i,t}$ is an average of log-likelihood ratios, it is bounded by the supremum of those ratios in the neighborhood $B_\delta(\theta^*)$, which is exactly $C_{i,\delta}$, so by the definition of $\tau_L$:

$$L_{\tau_L \wedge T} \leq \log(b/w_\delta) + C_{i,\delta}.$$

Note that $C_{i,\delta}$ is finite by Assumption F.6.3. Rearranging and taking the limit $T \to \infty$ (justified by the monotone convergence theorem as in the proof of Theorem 3.4), we obtain:

$$\mathbb{E}_{\theta^*}[\tau_i(\nu)] \leq \mathbb{E}_{\theta^*}[\tau_L] \leq \frac{\log b + \log(1/w_\delta) + C_{i,\delta}}{\bar{\mathrm{KL}}_i(\mu_{\theta^*}) - \epsilon},$$

which completes the proof. $\qquad\square$

## G. Alternative Testing Framework in Repeated Normal-Form Games

In this appendix, we consider an alternative monitoring framework for repeated normal-form games that operates under different informational assumptions than in Section 2. Here, we assume that the joint strategy profile $\pi = \pi_1 \times \cdots \times \pi_n$ of the Nash equilibrium is known to the monitor, but the payoff functions $u_i$ are not directly observable. This setting arises naturally in scenarios where the equilibrium has been theoretically derived or previously estimated, but precise payoff measurements are unavailable or noisy. While we focus on Nash equilibrium for concreteness, as in Section 2, the framework naturally extends to other equilibrium concepts.

The testing procedure developed in this section is based on likelihood ratio statistics, which quantify deviations of the observed play from the hypothesized equilibrium strategy profile. The idea is similar to in our analysis of equilibrium testing in stochastic games in Section 3. We omit most proofs, as they closely resemble those presented in that section.

### G.1. Sequential testing via likelihood ratios

Let $\mathbf{a}_t = (a_{1,t}, \ldots, a_{n,t})$ denote the observed action profile at round $t$. Suppose we have a candidate alternative strategy $\pi^{(1)} = (\pi_1^{(1)}, \ldots, \pi_n^{(1)})$ specifying, for each player $i$, a distribution over actions that represents a plausible deviation from the null strategy $\pi$.

For player $i \in [n]$, define the per-round likelihood ratio

$$E_{i,t} := \frac{\pi_i^{(1)}(a_{i,t})}{\pi_i(a_{i,t})}. \tag{17}$$

**Lemma G.1.** *For each player $i \in [n]$ and any alternative strategy $\pi_i^{(1)}$,*

$$\mathbb{E}_{\mathcal{H}_0}[E_{i,t}] = 1,$$

*so $E_{i,t}$ is a valid e-value for the null hypothesis $\mathcal{H}_0$.*

*Proof.* By definition of expectation,

$$\mathbb{E}_{\mathcal{H}_0}[E_{i,t}] = \sum_{a_i \in A_i} \pi_i(a_i) \frac{\pi_i^{(1)}(a_i)}{\pi_i(a_i)} = \sum_{a_i \in A_i} \pi_i^{(1)}(a_i) = 1.$$

$\qquad\square$

### G.2. Supermartingale construction

Analogously to the payoff-based framework, we define a cumulative product over rounds to construct a supermartingale.

**Lemma G.2.** *Define*

$$M_{i,t} := \prod_{s=1}^{t} E_{i,s} = \prod_{s=1}^{t} \frac{\pi_i^{(1)}(a_{i,s})}{\pi_i(a_{i,s})}, \quad M_{i,0} := 1.$$

*Then $(M_{i,t})_{t \geq 0}$ is a nonnegative supermartingale with respect to the filtration $(\mathcal{F}_t)_{t \geq 0}$ under $\mathcal{H}_0$.*

*Proof.* Since the $E_{i,t}$ are independent and satisfy $\mathbb{E}_{\mathcal{H}_0}[E_{i,t} \mid \mathcal{F}_{t-1}] = 1$ by Lemma G.1, the product $M_{i,t}$ satisfies

$$\mathbb{E}_{\mathcal{H}_0}[M_{i,t} \mid \mathcal{F}_{t-1}] = M_{i,t-1} \cdot \mathbb{E}_{\mathcal{H}_0}[E_{i,t} \mid \mathcal{F}_{t-1}] = M_{i,t-1}.$$

Hence, $(M_{i,t})_{t \geq 0}$ is a nonnegative martingale (and thus a supermartingale) with initial value $M_{i,0} = 1$. $\qquad \square$

### G.3. FWER control

We can immediately obtain a sequential test with controlled FWER by applying Ville's inequality to the supermartingales.

**Theorem G.3.** *For any threshold $b > 1$, under $\mathcal{H}_0$ we have*

$$\mathbb{P}_{\mathcal{H}_0}\left( \exists i \in [n] : \sup_{t \geq 0} M_{i,t} \geq b \right) \leq \frac{n}{b}.$$

*In particular, choosing $b = n/\alpha$ ensures FWER control at level $\alpha$:*

$$\mathbb{P}_{\mathcal{H}_0}\left( \exists i : \sup_{t \geq 0} M_{i,t} \geq \tfrac{n}{\alpha} \right) \leq \alpha.$$

### G.4. Expected detection time under the alternative

**Theorem G.4** (Expected detection time under the alternative)**.** *Fix a player $i \in [n]$. Assume that the alternative strategy $\pi_i^{(1)}$ has support contained in that of the null strategy $\pi_i$, i.e., $\pi_i(a) > 0$ whenever $\pi_i^{(1)}(a) > 0$. This ensures that the Kullback-Leibler divergence*

$$\mathrm{KL}(\pi_i^{(1)} \| \pi_i) := \sum_{a_i \in A_i} \pi_i^{(1)}(a_i) \log \frac{\pi_i^{(1)}(a_i)}{\pi_i(a_i)}$$

*is finite. For a threshold $b > 1$, define the detection time*

$$\tau_i := \inf\{t \geq 1 : M_{i,t} \geq b\}.$$

*Further define the overshoot constant*

$$C_i := \max_{a \in A_i} \left\{ \left| \log \frac{\pi_i^{(1)}(a)}{\pi_i(a)} \right| : \pi_i^{(1)}(a) > 0 \right\} < +\infty.$$

*Then the expected detection time under the alternative satisfies*

$$\mathbb{E}_{\mathcal{H}_1}[\tau_i] \leq \frac{\log b + C_i}{\mathrm{KL}(\pi_i^{(1)} \| \pi_i)},$$

*where the denominator is strictly positive provided that $\pi_i^{(1)}$ is not equal to $\pi_i$ almost surely.*

### G.5. Mixtures over alternative strategies

When a precise alternative is unknown, it is often preferable to test against a mixture of alternatives, which provides robustness to uncertainty about the true deviation. As in Section 3, we may define a mixture e-value

$$E_{i,t}(\nu) := \int \frac{\pi_i^{(1)}(a_{i,t})}{\pi_i(a_{i,t})} \, d\nu_i(\pi_i^{(1)}),$$

where $\nu$ is a prior distribution over plausible deviations. Under $\mathcal{H}_0$, $E_{i,t}(\nu)$ remains an e-value, and the corresponding cumulative product

$$M_{i,t}(\nu) := \prod_{s=1}^{t} E_{i,s}(\nu)$$

defines a supermartingale that can be continuously monitored.

## H. Additional Experimental Details

### H.1. Repeated normal-form games

#### H.1.1. EXPERIMENTAL SETUP

**Game definition.** We use a $2 \times 2$ normal-form game with $n = 2$ players and action set $A_i = \{0, 1\}$ for player $i \in \{1, 2\}$. The payoff matrices $U_1$ and $U_2$ are given by:

$$U_1 = \begin{pmatrix} 0.9 & 0.2 \\ 0.3 & 0.7 \end{pmatrix}, \qquad U_2 = \begin{pmatrix} 0.5 & 0.3 \\ 0.2 & 0.7 \end{pmatrix}.$$

It can be easily checked that this game has a unique mixed Nash equilibrium $\pi = (\pi_1, \pi_2)$ where $\pi_1 = (5/7, 2/7)$ and $\pi_2 = (5/11, 6/11)$.

**Parameters.** For all experiments, we set the significance level $\alpha = 0.2$ and the betting parameter $\lambda = 0.05$. For the $\mathcal{H}_1$ scenario, we design an alternative strategy profile $\pi_{\text{alt}} = (\pi_1^{\text{alt}}, \pi_2^{\text{alt}})$ where $\pi_1^{\text{alt}} = (17/20, 3/20)$ and $\pi_2^{\text{alt}} = (13/20, 7/20)$. This setup creates two weak, balanced signals $\eta_{1,0} \approx 0.032$ for $H_1^{(0)}$ and $\eta_{2,0} \approx 0.033$ for $H_2^{(0)}$, satisfying the alternative hypothesis $\mathcal{H}_1$.

#### H.1.2. VALIDATION OF FWER CONTROL (UNDER $\mathcal{H}_0$)

To validate our framework's control of the FWER, we first conduct a simulation under the null hypothesis $\mathcal{H}_0$. We simulate $R = 300$ independent runs, each for $T = 4000$ rounds, where both players sample actions according to the true Nash equilibrium $\pi$. We monitor all $m = 4$ supermartingales using the FWER rejection threshold $b = m/\alpha = 4/0.2 = 20$.

Figure 4 shows the results. Across all 300 runs, only 3 runs resulted in a false rejection (a rate of 0.01). This empirical rejection rate is far below our nominal $\alpha = 0.2$ level, confirming that our procedure provides rigorous FWER control.

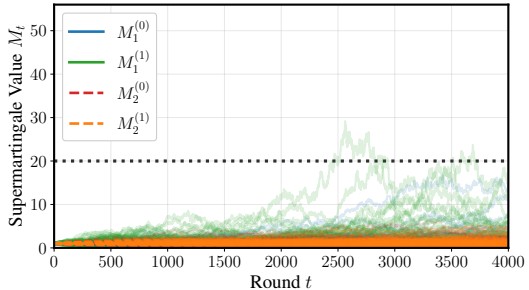

*Figure 4.* Evolution of all four test supermartingales across 300 independent runs under the null hypothesis $\mathcal{H}_0$. The horizontal dotted line is the FWER rejection threshold $b = 20$. The empirical FWER (0.01) is well below the target level $\alpha = 0.2$.

This empirical FWER rate (0.01) is far below the nominal $\alpha = 0.2$ level, which is partly attributed to the small value of the betting parameter $\lambda = 0.05$. A small $\lambda$ reduces the overall variance of the supermartingales, making large, chance

excursions above the threshold highly improbable. We analyze the behavior of more balanced betting mixtures over the parameter $\lambda$ in detail later.

### H.1.3. ANALYSIS OF SAMPLE DETECTION TRAJECTORIES (UNDER $\mathcal{H}_1$): FWER VS FDR

In a $2 \times 2$ normal-form game, there are $m = 4$ total hypotheses to test: $H_1^{(0)}$ and $H_1^{(1)}$ for Player 1, and $H_2^{(0)}$ and $H_2^{(1)}$ for Player 2, using notations from Equation (8). However, a player's strategy $\pi_i$ cannot be simultaneously outperformed by *both* alternative actions. Formally, let $\Delta_1 = \mathbb{E}[u_1(0, a_2)] - \mathbb{E}[u_1(1, a_2)]$. The conditions for $H_1^{(0)}$ and $H_1^{(1)}$ to be profitable deviations (i.e., true alternatives) are $\pi_1(1) \cdot \Delta_1 \geq \eta$ and $-\pi_1(0) \cdot \Delta_1 \geq \eta$, respectively. As $\pi_1(0), \pi_1(1) > 0$, the first condition requires $\Delta_1 > 0$ while the second requires $\Delta_1 < 0$. These conditions are mutually exclusive. Therefore, at most one deviation per player can be profitable. This implies that for our $m = 4$ hypotheses, at most $|\mathcal{H}_1| = 2$ can be true alternatives. This has two important consequences for our plots:

1. **Plotted martingales:** We only plot the two supermartingales $M_{1,t}^{(0)}$ and $M_{2,t}^{(0)}$, which correspond to the two true alternatives in our $\mathcal{H}_1$ scenario. The other two martingales (for $a' = 1$) are true nulls and remaining below 1 in expectation.

2. **Plotted thresholds:** The FDR procedure (Algorithm 2) uses the maximum index $k_t = \max\{k : N_t(k) \geq k\}$. We know that the maximum number of true alternative signals is 2. Therefore, in practice, the rejection is almost always triggered by the simultaneous accumulation of evidence from the two true signals. We consequently focus the visualization on the two key effective rejection boundaries: the FWER threshold ($k = 1$) and the lower FDR threshold for $k = 2$.

Figure 5 shows six sample trajectories from our experiment, illustrating the comparative behavior of the FWER and FDR procedures. The FWER threshold ($k = 1$) is $u_1 = m/\alpha = 4/0.2 = 20$. The FDR $k = 2$ threshold is $u_2 = m/(2\alpha) = 4/(2 \cdot 0.2) = 10$.

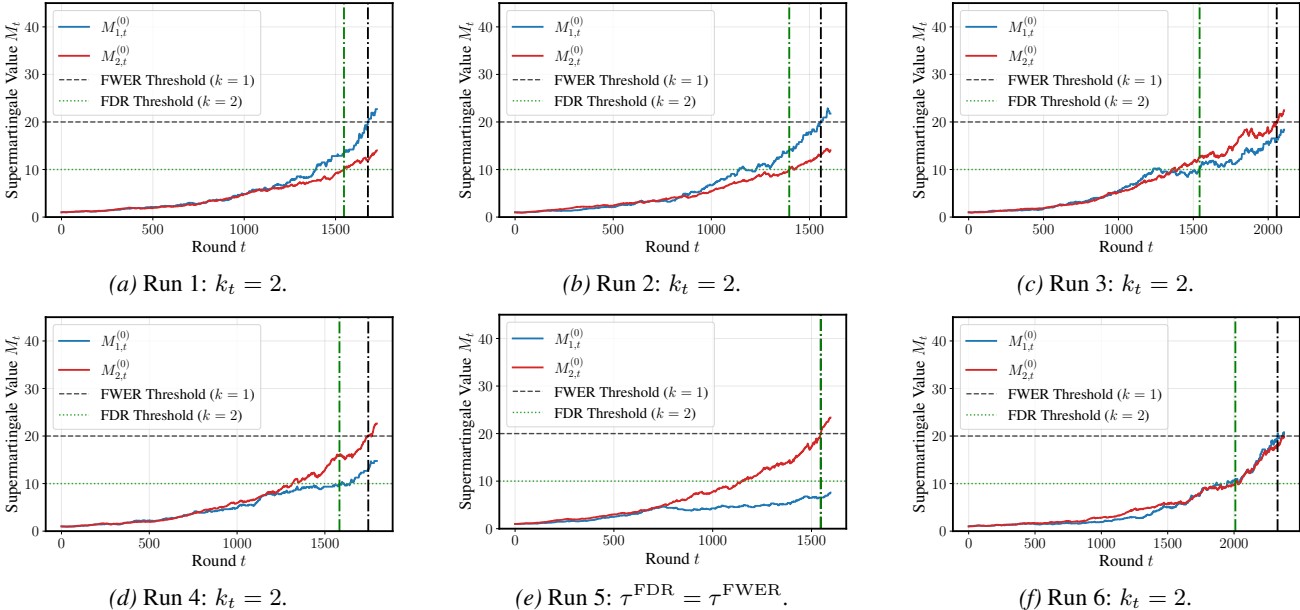

*(a)* Run 1: $k_t = 2$.     *(b)* Run 2: $k_t = 2$.     *(c)* Run 3: $k_t = 2$.

*(d)* Run 4: $k_t = 2$.     *(e)* Run 5: $\tau^{\text{FDR}} = \tau^{\text{FWER}}$.     *(f)* Run 6: $k_t = 2$.

*Figure 5.* Six sample trajectories from the $\mathcal{H}_1$ experiment ($\alpha = 0.2$, $\lambda = 0.05$). Each plot shows the two true supermartingales $M_{1,t}^{(0)}$ and $M_{2,t}^{(0)}$ against the FWER ($b = 20$, dashed black) and FDR $k = 2$ ($b = 10$, dotted green) thresholds. Vertical lines indicate stopping times.

**Analysis of sample runs.** The sample runs illustrate the two primary outcomes of our experiment.

**FDR speedup (e.g., Run 1, 2, 3, 4, 6):** In most runs, the two balanced signals grow at a similar rate. They both cross the $k = 2$ threshold ($b = 10$) long before either one can reach the FWER threshold ($b = 20$). This results in an FDR detection ($\tau^{\text{FDR}}$, green vertical line) that is significantly earlier than the FWER detection ($\tau^{\text{FWER}}$, black vertical line). On average across all 300 runs, this translates to an FDR speedup of $1.15\times$ over the FWER procedure.

**Tied case (Run 5):** This run shows an example of a "tie," where $\tau^{\text{FDR}} = \tau^{\text{FWER}}$. Due to random variance, the $M_{1,t}^{(0)}$ martingale (blue) grew much faster than the $M_{2,t}^{(0)}$ martingale. It successfully crossed the FWER threshold ($b = 20$) before $M_{2,t}^{(0)}$ cross the $k = 2$ threshold ($b = 10$). At the moment of FWER rejection, $N_t(1) = 1$, so $k_t = 1$. This results in both procedures stopping at the exact same time.

### H.1.4. PARAMETER SENSITIVITY ANALYSIS

To further analyze the interplay between the betting parameter $\lambda$, the significance level $\alpha$, and the signal strength $\eta$, we conduct a grid-search experiment. We test with $\alpha \in \{0.2, 0.1, 0.05\}$, Dirac-mixture betting parameters $\lambda \in \{0.05, 0.1, 0.15, 0.4\}$, and three distinct $\mathcal{H}_1$ scenarios with increasing signal strength, $\eta \in \{0.05, 0.1, 0.15\}$.[2]

**FWER control (under $\mathcal{H}_0$).** First, we analyze the FWER control under $\mathcal{H}_0$. Table 1 summarizes the empirical FWER from $R = 300$ runs for each $(\lambda, \alpha)$ pair. In all 12 scenarios, the observed false rejection rate remains strictly below the nominal significance level $\alpha$, validating the statistical control of our sequential framework, as predicted by Theorem 2.6.

Figure 6 visualizes these simulations. As $\lambda$ increases from $0.05$ (far left) to $0.40$ (far right), the variance of the null supermartingales visibly increases. For $\lambda = 0.05$, the processes are highly stable. For $\lambda = 0.15$, the trajectories are far more volatile, with several runs experiencing large, transient spikes. This is even more pronounced for $\lambda = 0.40$, where the process is extremely volatile; most trajectories collapse towards zero due to the aggressive betting, and several exhibit large, early spikes before collapsing. Despite this volatility, the empirical FWER remains controlled across all settings.

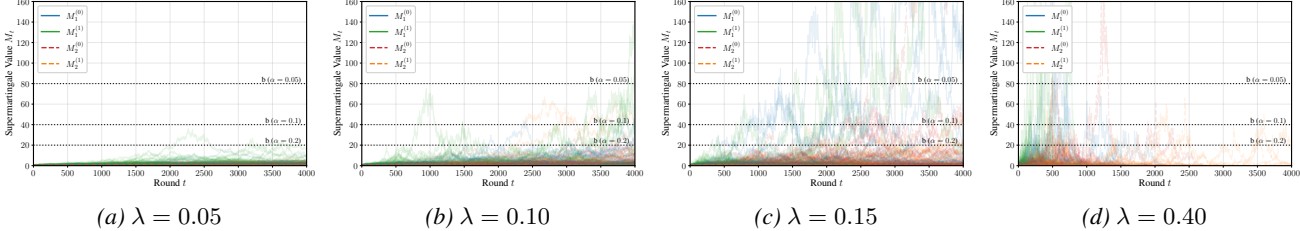

*(a) $\lambda = 0.05$*     *(b) $\lambda = 0.10$*     *(c) $\lambda = 0.15$*     *(d) $\lambda = 0.40$*

*Figure 6.* FWER control (under $\mathcal{H}_0$) trajectories for different fixed $\lambda$. As $\lambda$ increases, the variance of the null supermartingales increases, but the empirical FWER remains controlled.

Table 1 summarizes the empirical FWER observed under the null hypothesis $\mathcal{H}_0$. In all 12 scenarios, the observed false rejection rate remains strictly below the nominal significance level $\alpha$, validating the statistical control of our sequential framework. This result aligns with the theoretical prediction of Theorem 2.6.

**Detection time (under $\mathcal{H}_1$).** Second, we analyze the average FWER stopping time under $\mathcal{H}_1$. Figure 7 plots the average stopping time (solid lines) and the interquartile range (25th-75th percentile, shaded region) on a log scale against $\lambda$ for each $\alpha$ and $\eta$.

- **Impact of $\eta$:** As expected by the theoretical bound provided by Theorem 2.7, for any fixed $\alpha$ and $\lambda$, detection time decreases as the signal strength $\eta$ increases. In all three plots, the $\eta = 0.15$ line (strongest signal) is uniformly below the other lines.

- **Impact of $\alpha$:** A smaller $\alpha$ increases the rejection threshold $b$, which uniformly increases the stopping time. This is visible by comparing the $y$-axis values across the three plots (e.g., for $\eta = 0.05, \lambda = 0.10$, the average stopping time is $\approx 775$ for $\alpha = 0.1$ but $\approx 911$ for $\alpha = 0.05$).

- **Impact of $\lambda$:** A key observation from the experimental results in Figure 7 is that the average stopping time is a monotonically decreasing function of $\lambda$ for all tested $\eta$. This indicates that a larger $\lambda$ (e.g., $\lambda = 0.4$) is empirically superior to the min-max optimal $\lambda^* = \eta$ (e.g., $\lambda = 0.1$ for the $\eta = 0.1$ case). This is not an error. Proposition 2.4 provides the optimal $\lambda$ against a *worst-case* distribution. However, the *actual* distribution of the increment $X_{i,t}^{(a_i')}$ in our

---

[2]The $\mathcal{H}_1$ scenarios are constructed using the same fixed $U_1, U_2$ payoff matrices. We fix Player 2's strategy at $\pi_2 = (10/11, 1/11)$ and adjust Player 1's mixed strategy $\pi_1 = (p, 1-p)$ to generate the target $\eta$. The specific strategies used are: $\eta = 0.05 \implies \pi_1 = (0.9, 0.1)$; $\eta = 0.1 \implies \pi_1 = (0.8, 0.2)$; and $\eta = 0.15 \implies \pi_1 = (0.7, 0.3)$.

*Table 1.* Empirical FWER (300 runs) under $\mathcal{H}_0$ for different $\lambda$ and $\alpha$.

| $\lambda$ | $\alpha$ | Threshold ($b$) | Empirical FWER |
|---|---|---|---|
| 0.05 | 0.20 | 20.0 | 0.010 |
| 0.05 | 0.10 | 40.0 | 0.000 |
| 0.05 | 0.05 | 80.0 | 0.000 |
| 0.10 | 0.20 | 20.0 | 0.077 |
| 0.10 | 0.10 | 40.0 | 0.037 |
| 0.10 | 0.05 | 80.0 | 0.007 |
| 0.15 | 0.20 | 20.0 | 0.100 |
| 0.15 | 0.10 | 40.0 | 0.050 |
| 0.15 | 0.05 | 80.0 | 0.027 |
| 0.40 | 0.20 | 20.0 | 0.153 |
| 0.40 | 0.10 | 40.0 | 0.077 |
| 0.40 | 0.05 | 80.0 | 0.030 |

specific game is far from this worst-case. Below, we find the *true* log-optimal $\lambda$ by solving the optimization for our specific game, rather than the min-max problem.

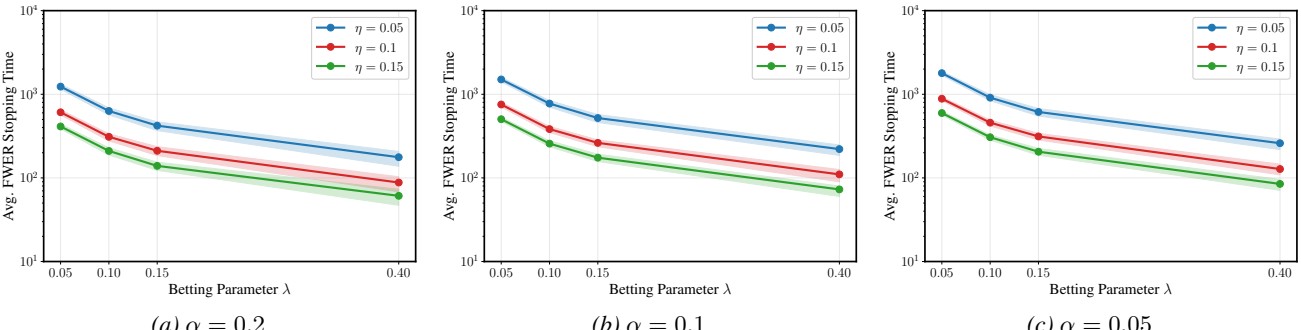

*(a)* $\alpha = 0.2$        *(b)* $\alpha = 0.1$        *(c)* $\alpha = 0.05$

*Figure 7.* Average FWER stopping time (solid lines) and 25th-75th percentile (shaded region) under $\mathcal{H}_1$ as a function of $\lambda$, for fixed $\alpha$ and varying $\eta$. The convex decreasing trend illustrates that more aggressive betting leads to faster, more consistent detection in this regime.

The log-optimal $\lambda$ is the one that maximizes the expected log-growth rate, $g(\lambda) = \mathbb{E}_{\mathcal{H}_1}[\log(1 - \lambda X)]$. Let us analyze the distribution of the increment $X_{i,t}^{(a_i')}$ for the $\eta = 0.1$ alternative from the simulation code.

**1. Identifying the violating martingale**    The alternative strategy is $\pi_1 = (0.8, 0.2)$ and $\pi_2 = (10/11, 1/11)$. The utility matrix for Player 1 is $U_1 = \begin{pmatrix} 0.9 & 0.2 \\ 0.3 & 0.7 \end{pmatrix}$.

- **Deviation to $a_1' = 1$:** The increment is $X = u_1(a_1, a_2) - u_1(1, a_2)$. The mean is $\mathbb{E}[X] = +0.4$. This is $\geq -\eta$, so this player-action pair is not responsible for detection.

- **Deviation to $a_1' = 0$:** This is the violating deviation. The increment is $X = u_1(a_1, a_2) - u_1(0, a_2)$.

**2. Deriving the distribution of $X$.**    The distribution of $X$ for the $a_1' = 0$ deviation is a 3-point distribution:

- $\mathbb{P}(a = (0, \cdot)) = \mathbb{P}(a_1 = 0) = 0.8$. In this case, $X = u_1(0, a_2) - u_1(0, a_2) = 0$.

- $\mathbb{P}(a = (1, 0)) = \mathbb{P}(a_1 = 1)\mathbb{P}(a_2 = 0) = 0.2 \times (10/11) = 2/11$. In this case, $X = u_1(1, 0) - u_1(0, 0) = 0.3 - 0.9 = -0.6$.

- $\mathbb{P}(a = (1,1)) = \mathbb{P}(a_1 = 1)\mathbb{P}(a_2 = 1) = 0.2 \times (1/11) = 0.2/11$. In this case, $X = u_1(1,1) - u_1(0,1) = 0.7 - 0.2 = +0.5$.

The mean is $\mathbb{E}[X] = 0.8(0) + \frac{2}{11}(-0.6) + \frac{0.2}{11}(0.5) = \frac{-1.2+0.1}{11} = \frac{-1.1}{11} = -0.1$, which is exactly $-\eta$.

**3. Solving for the game-specific $\lambda^*$.** We maximize the expected log-growth $g(\lambda)$ for this specific distribution:

$$
\begin{aligned}
g(\lambda) &= \mathbb{E}[\log(1 - \lambda X)] \\
&= 0.8 \cdot \log(1 - \lambda \cdot 0) + \frac{2}{11}\log(1 - \lambda(-0.6)) + \frac{0.2}{11}\log(1 - \lambda(0.5)) \\
&= \frac{2}{11}\log(1 + 0.6\lambda) + \frac{0.2}{11}\log(1 - 0.5\lambda)
\end{aligned}
$$

To find the maximum, we take the derivative with respect to $\lambda$:

$$
g'(\lambda) = \frac{2}{11}\left(\frac{0.6}{1 + 0.6\lambda}\right) + \frac{0.2}{11}\left(\frac{-0.5}{1 - 0.5\lambda}\right) = \frac{1}{11}\left(\frac{1.2}{1 + 0.6\lambda} - \frac{0.1}{1 - 0.5\lambda}\right)
$$

We set $g'(\lambda) = 0$ to find the critical point:

$$
\frac{1.2}{1 + 0.6\lambda} = \frac{0.1}{1 - 0.5\lambda} \implies 1.2(1 - 0.5\lambda) = 0.1(1 + 0.6\lambda)
$$

$$
1.2 - 0.6\lambda = 0.1 + 0.06\lambda \implies 1.1 = 0.66\lambda \implies \lambda^* = \frac{1.1}{0.66} = \frac{5}{3} \approx 1.67
$$

The true log-optimal betting fraction for this specific game's $\eta = 0.1$ alternative is $\lambda^* \approx 1.67$. This value is outside the valid parameter range $\lambda \in [0, 1]$.

Since $g'(\lambda) > 0$ for all $\lambda \in [0, 1]$, the expected log-growth rate $g(\lambda)$ is a strictly monotonically increasing function on the entire valid parameter space.

This analytical finding explains the experimental results in Figure 7. The plots correctly show that for this game, the stopping time (a proxy for $1/g(\lambda)$) is a strictly decreasing function of $\lambda$ on the tested interval $[0.05, 0.4]$. The min-max choice $\lambda = \eta$ is a robust, safe choice, but it is "too conservative" for this specific, favorable distribution.

### H.2. Large-Scale Experiment: Population Rock-Paper-Scissors

The experiments in the main text use a minimal $2 \times 2$ game to isolate the FDR-versus-FWER comparison. Here we verify that the conclusions persist in a substantially larger game, with more players, a larger action space, and many more hypotheses.

We consider a population Rock-Paper-Scissors game with $n = 20$ players, each with action set $A_i = \{\text{Rock, Paper, Scissors}\}$, so $|A_i| = 3$. Each player's payoff is the average payoff obtained by playing the standard Rock-Paper-Scissors matrix

$$
U = \begin{pmatrix} 0.5 & 0.0 & 1.0 \\ 1.0 & 0.5 & 0.0 \\ 0.0 & 1.0 & 0.5 \end{pmatrix}
$$

against each of the other $n - 1$ players, rescaled to $[0, 1]$. The total number of monitored hypotheses is $m = \sum_{i=1}^{n} |A_i| = 20 \times 3 = 60$, a fifteen-fold increase over the $m = 4$ hypotheses of the main-text game.

Under $\mathcal{H}_1$, 15 of the 20 players are *compliant* and play the uniform strategy $(1/3, 1/3, 1/3)$, while 5 players are biased toward Rock, playing $(0.6, 0.2, 0.2)$. Because each player's payoff is evaluated against the entire population, a minority over-playing Rock skews the opponent pool: against a Rock-heavy field, switching to Paper becomes a genuinely profitable deviation for the players still playing uniformly. The biased subgroup therefore does not merely deviate itself; it breaks the equilibrium for the compliant majority as well. Recall that the null $\mathcal{H}_0$ asserts that no profitable deviation exists; the experiment thus tests whether the population as a whole remains at equilibrium, and a single biased subgroup is enough to violate this for many players at once.

We run $R = 200$ independent trials, each over a horizon of $T = 3000$ rounds, at significance level $\alpha = 0.2$ and betting parameter $\lambda = 0.05$. The FWER procedure uses the threshold $b = m/\alpha = 60/0.2 = 300$; the e-BH (FDR) procedure uses the data-dependent thresholds $1/(k\alpha\gamma_{i,a'_i})$ with uniform weights $\gamma_{i,a'_i} = 1/m$, as in Algorithm 2.

With $m = 60$ hypotheses, both the FWER and FDR procedures pay a multiple-testing cost that scales with the full collection of hypotheses. A natural way to reduce this cost is to first *screen* the hypotheses on a held-out window of rounds, and then test only the most promising ones. We include such a procedure as an additional baseline.

The data-splitting procedure operates in two phases. In the *screening phase* (rounds $t = 1, \ldots, T_0$, with $T_0 = 50$), no testing is performed; instead, for each hypothesis $(i, a'_i)$ we accumulate the regret-like increments $\sum_{t=1}^{T_0} X_{i,t}^{(a'_i)}$. At the end of the window, we select the $K = 10$ hypotheses with the most negative accumulated increments (those that look most like genuine profitable deviations) and discard the remaining $m - K = 50$. In the *testing phase* (rounds $t > T_0$), the sequential test of Section 2 is run *only* on the $K$ selected hypotheses, using the reduced threshold $b = K/\alpha = 50$ for the FWER variant and the e-BH thresholds over $K$ hypotheses for the FDR variant.

The procedure is statistically valid because the selection rule depends only on rounds $1, \ldots, T_0$, whereas the e-processes used for testing are built exclusively from rounds $t > T_0$. Since the action profiles are drawn i.i.d. across rounds, the screening and testing windows are independent: the selected set of $K$ hypotheses is fixed before any test data is observed, so no selective-inference bias is introduced, and the supermartingale and FDR guarantees of Section 2 apply verbatim to the reduced family of $K$ hypotheses. The cost of the procedure is the $T_0$ discarded rounds and the risk of screening out a true deviation when $T_0$ is too small; this is a loss of power, not a violation of error control.

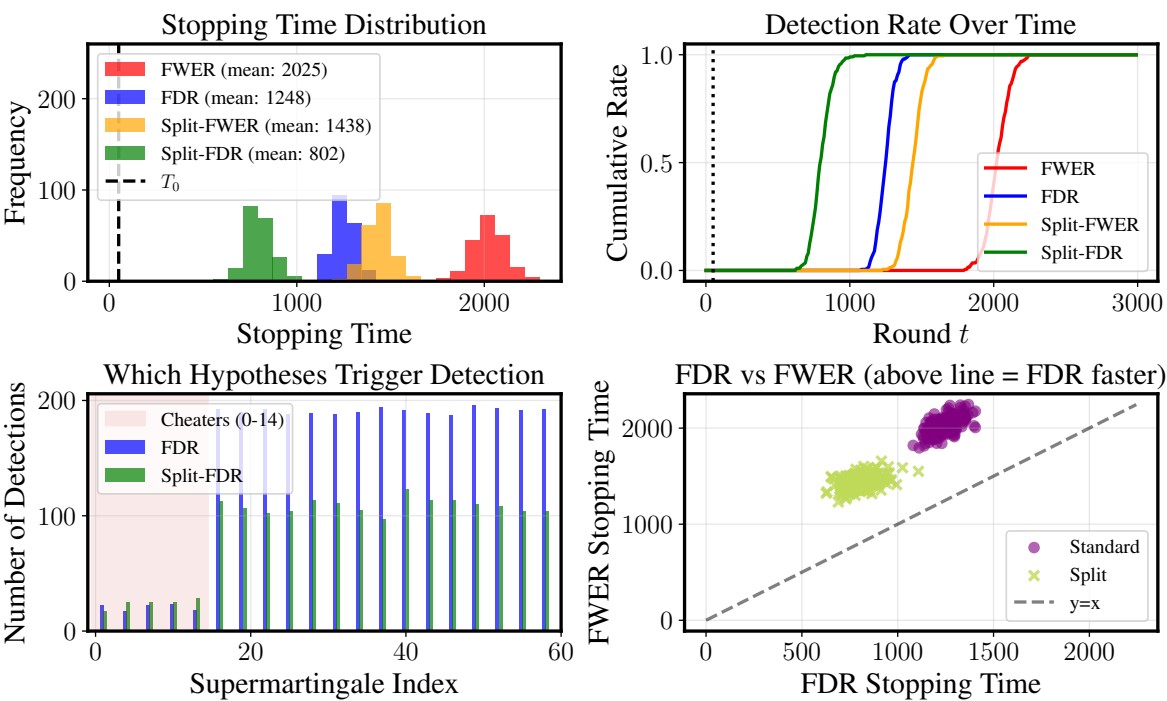

*Figure 8.* **Large-scale population Rock-Paper-Scissors** ($n = 20$ players, $|A_i| = 3$, $m = 60$ hypotheses, $\alpha = 0.2$, $\lambda = 0.05$, $R = 200$ runs). Comparison of FWER, FDR, and their data-split variants (Split-FWER, Split-FDR with $T_0 = 50$, $K = 10$). **(Top-Left)** Distribution of stopping times; the dashed line marks the end of the screening window $T_0$. **(Top-Right)** Cumulative detection rate over rounds. **(Bottom-Left)** Detection frequency per hypothesis; the shaded band marks the hypothesis indices of the 5 players biased toward Rock. Detections concentrate *outside* this band, on the compliant players' profitable best responses against the Rock-skewed population. **(Bottom-Right)** Run-by-run scatter of FWER versus FDR stopping times; points above the diagonal indicate runs where FDR detected sooner.

Figure 8 compares the four procedures: FWER and FDR on the full family of $m = 60$ hypotheses, and their Split-FWER and Split-FDR counterparts on the screened family of $K = 10$. Three observations stand out.

First, the FDR procedure consistently detects deviations earlier than the FWER procedure, confirming Proposition 2.9 at scale: the mean detection time drops from 2025 rounds under FWER to 1248 under FDR.

Second, both data-splitting variants detect substantially earlier still. By restricting attention to $K = 10$ hypotheses, they lower the testing threshold and concentrate the error budget on the hypotheses most likely to correspond to true deviations, bringing the mean detection time down to $1438$ rounds for Split-FWER and $802$ for Split-FDR. The run-by-run scatter (Figure 8, bottom-right panel) confirms that the FDR-over-FWER ordering holds in essentially every individual trial, for both the standard and the split procedures.

Third, the per-hypothesis panel (Figure 8, bottom-left panel) shows *which* hypotheses fire, and the pattern is informative: detections concentrate on the hypotheses of the $15$ *compliant* players, not the $5$ biased ones. This is correct behavior. The test flags every $(i, a_i')$ that constitutes a profitable deviation, and in Rock-Paper-Scissors a population skewed toward Rock makes "deviate to Paper" genuinely profitable for the players still playing uniformly. The detector therefore identifies loss of equilibrium wherever incentives are violated, rather than singling out the agents who caused the skew.

Overall, the experiment confirms that the advantage of FDR control over FWER control persists (and that simple screening offers a further, complementary speedup) as the number of players, the action space, and the number of hypotheses all grow.

## H.3. Stochastic games

### H.3.1. IMPLEMENTATION DETAILS—GRID SOCCER GAME

The Grid Soccer environment simulates a zero-sum game played on a discrete grid where an Attacker (Player A) attempts to carry a ball to a goal zone while avoiding a Defender (Player B).

**Game dynamics and parameters.** The game is played on a grid of size $W \times H = 5 \times 4$.

- **State space:** The state is defined by the coordinates of Player A $(x_A, y_A)$, Player B $(x_B, y_B)$, and a binary indicator for ball possession.

- **Action space:** Each agent chooses from 5 actions: {North, South, East, West, Wait}.

- **Transitions:** Player A has deterministic movement dynamics. Player B is "slippery" to simulate a less agile defender: with probability $p_{\text{slip}} = 0.25$, Player B's move fails and they remain in their previous position. If the agents collide, the ball possession may swap (probabilistic bounce). When the two players attempt to enter each other's cell simultaneously, or when their intended moves place them on the same cell, both actions are cancelled and they bounce back to their previous positions; in this rebound situation, ball possession switches with probability 0.5. If Player A moves into Player B's cell while B chooses the "Wait" action, possession transfers to Player B; symmetrically, if Player B moves into A's cell while A waits, possession transfers to Player A. In all other cases the players simply execute their resulting moves (accounting for walls), and possession remains unchanged.

- **Rewards:** The Attacker receives $+100$ for reaching the goal column ($y = W - 1$) and $-100$ if the Defender steals the ball and reaches the Attacker's goal ($y = 0$). To encourage active play and avoid infinite looping behavior during solver training, we apply a small step cost of $-0.05$ per turn.

- **Discount factor:** Although no value functions were introduced earlier, the computation of the Nash equilibrium relies on them. We use the discounted infinite-horizon value function

$$V_i^\pi(s) := \mathbb{E}\Big[ \sum_{t=0}^{\infty} \gamma^t r_i(s_t, \mathbf{a}_t) \,\Big|\, s_0 = s \Big],$$

where $\gamma = 0.95$ determines how future rewards are weighted.

**Nash equilibrium solver.** We compute the exact Nash Equilibrium policies $(\pi_A^*, \pi_B^*)$ using Linear Programming. We iteratively solve the matrix game induced by the Q-values for each state until the value function converges (tolerance $10^{-3}$, max iterations 100).

**Smoothing and support condition.** A critical implementation detail is the satisfaction of the support condition required by Theorem 3.4. In a raw Nash equilibrium derived from linear programming, optimal strategies often assign exactly zero probability to sub-optimal actions. If the alternative strategy assigns positive probability to an action where $\pi^*(a|s) = 0$, the likelihood ratio becomes infinite, leading to trivial instantaneous detection. To verify the asymptotic scaling law $O(1/\varepsilon^2)$,

which relies on the curvature of the KL divergence, we must ensure the baseline distribution has full support. We achieve this by applying global smoothing to the computed Nash policy:

$$\tilde{\pi}^*(a|s) = (1 - \beta)\pi^*(a|s) + \beta\frac{1}{|A|},$$

where $\beta = 0.05$ is a noise floor parameter. This ensures every action has at least probability $\beta/5$, guaranteeing that the likelihood ratios remain bounded and the martingale grows diffusively rather than jumping to infinity.

**Alternative strategy.** The deviation strategy is designed to simulate a "timid" attacker. For any state, we reduce the probability mass assigned by the Nash equilibrium to moving East (towards the goal) by 90% and redistribute the removed mass to moving West (retreating) or Wait. Specifically, if $\pi^*(East) > 0$, we define the reduction $\delta = 0.9 \times \pi^*(East)$ and update the probabilities as follows:

$$\pi^{\text{afraid}}(East) \leftarrow \pi^*(East) - \delta, \quad \pi^{\text{afraid}}(West) += 0.5 \times \delta, \quad \pi^{\text{afraid}}(Wait) += 0.5 \times \delta.$$

The actual strategy played by the suspect agent is the mixture $\pi^{(\varepsilon)} = (1 - \varepsilon)\tilde{\pi}^* + \varepsilon\pi^{\text{afraid}}$.

**Martingale test parameters.** We evaluate the detection sensitivity across a range of deviation magnitudes $\varepsilon \in \{0.05, 0.1, 0.2, 0.3, 0.5\}$. The sequential test uses a fixed threshold of $b = 20$, which corresponds to controlling the type-1 error at level $\alpha = 1/b = 0.05$. To obtain stable estimates of the expected stopping time $\mathbb{E}[\tau]$, we perform $N = 150$ independent Monte Carlo trials for each epsilon value.

We test the smoothed Nash equilibrium $\tilde{\pi}^*$ (as the null hypothesis) against the true alternative mixture $\pi^{(\varepsilon)}$. For the theoretical comparison shown in Figure 2, we anchored the theoretical curve at the largest $\varepsilon$ value (0.5) to focus on the asymptotic rate and discard multiplicative factors.

### H.3.2. GRID SOCCER GAMEPLAY VISUALIZATION

To provide qualitative insight into the agent behaviors, we visualize sample trajectories from the Grid Soccer environment alongside the corresponding action probabilities. For visualization clarity, only the first 12 steps of each episode are shown.

**Nash equilibrium play.** Figure 9 displays a sample episode where Player A follows the smoothed Nash equilibrium strategy $\tilde{\pi}^*$. The accompanying Table 2 details the complete step-by-step probabilities for this episode. Notice that Player A maintains a high probability of moving East ($\approx 0.96$) when the path is clear.

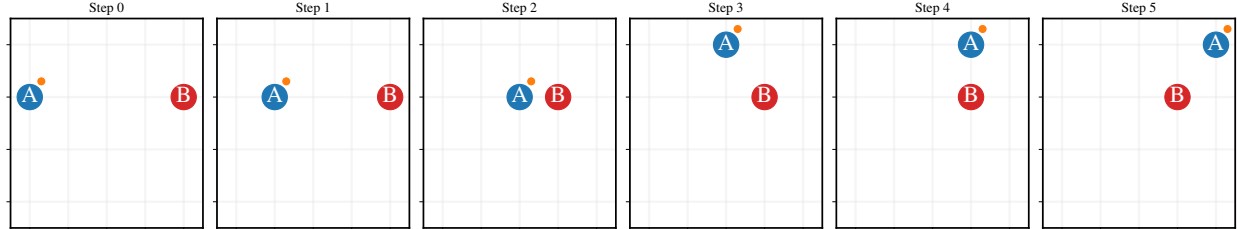

*Figure 9.* **Nash equilibrium gameplay sample.** The Attacker (A) plays the Nash strategy $\tilde{\pi}^*$. The agent actively moves East to goal, capitalizing on the Defender's slip at Step 0.

*Table 2.* **Complete log for Nash episode.** Detailed probabilities and events for the episode corresponding to Figure 9. Note the slip events (Steps 0, 3, and 4) where Player B intends to move but the real action becomes X.

| Step | A Pos | B Pos | Act A | Act B | Real B | Slip? | Poss | Player A Probabilities | | | | | Player B Probabilities | | | | |
| --- | --- | --- | --- | --- | --- | --- | --- | --- | --- | --- | --- | --- | --- | --- | --- | --- | --- |
| | (Ax,Ay) | (Bx,By) | | | | | | N | S | E | W | X | N | S | E | W | X |
| 0 | (1,0) | (1,4) | E | W | X | Yes | A | 0.01 | 0.01 | **0.96** | 0.01 | 0.01 | 0.01 | 0.01 | 0.01 | **0.96** | 0.01 |
| 1 | (1,1) | (1,4) | E | W | W | No | A | 0.01 | 0.01 | **0.96** | 0.01 | 0.01 | 0.01 | 0.01 | 0.01 | **0.96** | 0.01 |
| 2 | (1,2) | (1,3) | N | X | X | No | A | **0.69** | 0.28 | 0.01 | 0.01 | 0.01 | 0.01 | 0.01 | 0.01 | 0.01 | **0.96** |
| 3 | (0,2) | (1,3) | E | N | X | Yes | A | 0.01 | 0.01 | **0.96** | 0.01 | 0.01 | **0.96** | 0.01 | 0.01 | 0.01 | 0.01 |
| 4 | (0,3) | (1,3) | E | W | X | Yes | A | 0.01 | 0.01 | **0.96** | 0.01 | 0.01 | 0.01 | 0.01 | 0.01 | **0.96** | 0.01 |
| 5 | (0,4) | (1,3) | END | END | END | No | A | | | | | | | | | | |

**Afraid Strategy Play.** In contrast, Figure 10 illustrates a game where Player A follows the $\pi^{\text{afraid}}$ strategy. As shown in

Table 3, the probability of moving East is systematically suppressed ($\approx 0.1$), with mass redistributed to West and Wait ($\approx 0.44$). This results in a passive behavior where the agent retreats or stagnates despite opportunities to advance.

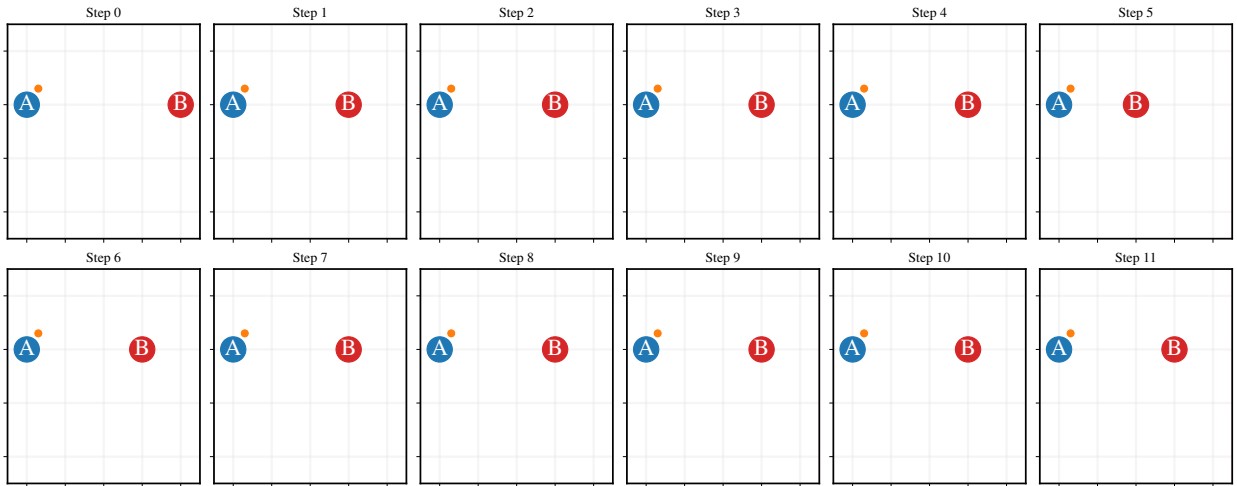

*Figure 10.* **Afraid strategy gameplay sample.** The Attacker plays the deviated strategy $\pi^{\text{afraid}}$. The agent systematically avoids moving East, resulting in retreat (West) or stalling (Wait).

*Table 3.* **Partial log for afraid episode.** Partial action history and probability distributions for both players for the episode shown in Figure 10. Player A's probability for East is manually suppressed to $\approx 0.1$, shifting mass to West/Wait ($\approx 0.44$). Even when unblocked, the agent hesitates.

| Step | A Pos (Ax,Ay) | B Pos (Bx,By) | Act A | Act B | Real B | Slip? | Poss | Player A Probabilities | | | | | Player B Probabilities | | | | |
|---|---|---|---|---|---|---|---|---|---|---|---|---|---|---|---|---|---|
| | | | | | | | | N | S | E | W | X | N | S | E | W | X |
| 0 | (1,0) | (1,4) | W | W | W | No | A | 0.01 | 0.01 | 0.096 | **0.441** | **0.441** | 0.01 | 0.01 | 0.01 | **0.96** | 0.01 |
| 1 | (1,0) | (1,3) | X | X | X | No | A | 0.01 | 0.01 | 0.096 | **0.441** | **0.441** | 0.01 | 0.01 | 0.01 | 0.01 | **0.96** |
| 2 | (1,0) | (1,3) | X | X | X | No | A | 0.01 | 0.01 | 0.096 | **0.441** | **0.441** | 0.01 | 0.01 | 0.01 | 0.01 | **0.96** |
| 3 | (1,0) | (1,3) | X | X | X | No | A | 0.01 | 0.01 | 0.096 | **0.441** | **0.441** | 0.01 | 0.01 | 0.01 | 0.01 | **0.96** |
| 4 | (1,0) | (1,3) | X | W | W | No | A | 0.01 | 0.01 | 0.096 | **0.441** | **0.441** | 0.01 | 0.01 | 0.01 | 0.01 | **0.96** |
| 5 | (1,0) | (1,2) | W | E | E | No | A | 0.01 | **0.492** | 0.0478 | 0.2251 | 0.2251 | 0.01 | 0.01 | **0.600** | 0.370 | 0.01 |
| 6 | (1,0) | (1,3) | W | X | X | No | A | 0.01 | 0.01 | 0.096 | **0.441** | **0.441** | 0.01 | 0.01 | 0.01 | 0.01 | **0.96** |
| 7 | (1,0) | (1,3) | W | X | X | No | A | 0.01 | 0.01 | 0.096 | **0.441** | **0.441** | 0.01 | 0.01 | 0.01 | 0.01 | **0.96** |
| 8 | (1,0) | (1,3) | X | X | X | No | A | 0.01 | 0.01 | 0.096 | **0.441** | **0.441** | 0.01 | 0.01 | 0.01 | 0.01 | **0.96** |
| 9 | (1,0) | (1,3) | W | X | X | No | A | 0.01 | 0.01 | 0.096 | **0.441** | **0.441** | 0.01 | 0.01 | 0.01 | 0.01 | **0.96** |
| 10 | (1,0) | (1,3) | X | X | X | No | A | 0.01 | 0.01 | 0.096 | **0.441** | **0.441** | 0.01 | 0.01 | 0.01 | 0.01 | **0.96** |
| 11 | (1,0) | (1,3) | X | X | X | No | A | 0.01 | 0.01 | 0.096 | **0.441** | **0.441** | 0.01 | 0.01 | 0.01 | 0.01 | **0.96** |

### H.3.3. IMPLEMENTATION DETAILS—PREDATOR-PREY GAME

Our Predator-Prey experiment tests the robustness of the framework in a multi-agent setting where the specific deviation parameter is unknown.

**Game dynamics and parameters.** The game is played on a grid of size $W \times H = 10 \times 10$.

- **Agents:** 3 Predators (1 Suspect, 2 Honest) vs 1 Prey.

- **Action space:** Each agent chooses from 5 actions: {Stay, Up, Down, Right, Left}.

- **Objective:** Predators win if any predator captures the prey (occupies the same cell).

- **Horizon:** Episodes are limited to $T_{\max} = 5000$ steps to ensure termination.

While the theoretical framework is framed around Nash equilibria, the martingale test is fundamentally a statistical test between a baseline policy and an alternative policy. This experiment demonstrates the flexibility of our framework by testing deviations from a simple heuristic baseline rather than a computational Nash equilibrium.

- **Null hypothesis ($\mathcal{H}_0$).** The baseline strategy is a random walk, where the agent selects actions uniformly at random ($p = 0.2$ for each of the 5 actions).

- **Alternative strategy.** We first define a heuristic "chasing" strategy $\pi^{\text{opt}}$ that assigns high probability ($p \propto 10$) to moves that reduce the Euclidean distance to the prey, medium probability ($p \propto 1$) to neutral moves, and low probability ($p \propto 0.1$) to moves that increase distance. The actual strategy played by the suspect agent is the mixture $\pi(a|s) = (1 - \varepsilon_{\text{true}})\frac{1}{|A|} + \varepsilon_{\text{true}}\pi^{\text{opt}}(a|s)$.

**Mixture martingale.** The suspect agent plays a strategy with parameter $\varepsilon_{\text{true}} \in \{0.05, 0.1, 0.2, 0.3, 0.4, 0.6, 0.8\}$. The monitor, unaware of $\varepsilon_{\text{true}}$, constructs a mixture martingale by averaging likelihood ratios over a discretized grid of candidate epsilons $\varepsilon_{\text{mix}} \in \mathcal{E}_{\text{mix}} = \{0.1, 0.3, 0.5, 0.7, 0.9\}$. The rejection threshold is set to $b = 20$, which corresponds to controlling the type-1 error at level $\alpha = 1/b = 0.05$. The results (Figure 3) confirm that this averaging approach maintains detection power proportional to the true informational distance (KL divergence), validating the practical applicability of the method when the exact magnitude of the deviation is unknown.

### H.3.4. PREY-PREDATOR GAMEPLAY VISUALIZATION

To provide qualitative insight into the detection mechanism in the Prey-Predator environment, we visualize a sample trajectory where the Suspect agent uses a mixed strategy ($\varepsilon_{\text{true}} = 0.6$). Figure 11 displays the first 12 steps of the episode, tracking the positions of the Suspect (Red), Honest agents (Blue), and the Prey (Green).

**Martingale progression.** The monitor evaluates the Suspect's actions against a hypothesis mixture. As shown in Figure 11 and detailed in Table 4, the martingale value grows significantly when the Suspect chooses actions that are highly probable under the chasing heuristic but unlikely under the null (random walk) hypothesis. For instance, at Step 3, the Martingale jumps from 2.71 to 4.97 as the Suspect moves "Down" towards the prey, a move heavily favored by the chasing component of the mixture strategy.

*Table 4.* **Partial log for suspect agent.** Detailed action probabilities and martingale updates for the initial steps of the episode shown in Figure 11. Note how selecting high-probability heuristic actions (like "Down" or "Right") drives the martingale up.

| Step | Row | Col | Action | Martingale | Stay | Up | Down | Left | Right |
|------|-----|-----|--------|------------|------|------|------|------|-------|
| 0 | 0 | 0 | Down | 1.0 | 0.1061 | 0.1061 | **0.3409** | 0.1061 | **0.3409** |
| 1 | 1 | 0 | Right | 1.5869 | 0.1071 | 0.0827 | **0.3515** | 0.1071 | **0.3515** |
| 2 | 1 | 1 | Right | 2.7072 | 0.1083 | 0.0828 | **0.3630** | 0.0828 | **0.3630** |
| 3 | 1 | 2 | Down | 4.9720 | 0.1083 | 0.0828 | **0.3630** | 0.0828 | **0.3630** |
| 4 | 2 | 2 | Stay | 9.4762 | 0.1083 | 0.0828 | **0.3630** | 0.0828 | **0.3630** |
| 5 | 2 | 2 | Right | 4.3418 | 0.1083 | 0.0828 | **0.3630** | 0.0828 | **0.3630** |
| 6 | 2 | 3 | Right | 8.0398 | 0.1083 | 0.0828 | **0.3630** | 0.0828 | **0.3630** |
| 7 | 2 | 4 | Right | 15.3790 | 0.1083 | 0.0828 | **0.3630** | 0.0828 | **0.3630** |
| 8 | 2 | 5 | Up | 30.2033 | 0.1083 | 0.0828 | **0.3630** | 0.0828 | **0.3630** |
| 9 | 1 | 5 | Stay | 8.3651 | 0.1083 | 0.0828 | **0.3630** | 0.0828 | **0.3630** |
| 10 | 1 | 5 | Down | 4.4740 | 0.1083 | 0.0828 | **0.3630** | 0.0828 | **0.3630** |
| 11 | 2 | 5 | Right | 7.7457 | 0.1083 | 0.0828 | **0.3630** | 0.0828 | **0.3630** |

### H.3.5. SCALING OF DETECTION TIME FOR DEVIATIONS

In our experiments, we observe that the average detection time $\mathbb{E}[\tau]$ scales as $O(1/\varepsilon^2)$. According to Theorem 3.4, the expected detection time is bounded by the inverse of the state-averaged Kullback-Leibler divergence. Therefore, the empirical observation implies that the Kullback-Leibler divergence is $\propto 1/\varepsilon^2$. Here, we provide a formal proof that the Kullback-Leibler divergence between a base strategy $\pi^*$ and a mixture strategy $\pi^{(\varepsilon)} = (1 - \varepsilon)\pi^* + \varepsilon\pi'$ indeed scales quadratically with $\varepsilon$ as $\varepsilon \to 0$, provided the base strategy satisfies the support condition.

**Proposition H.1.** *Let $P$ and $Q$ be two probability distributions over a finite action set $\mathcal{A}$ such that $P(a) > 0$ for all $a \in \mathcal{A}$. Let $P_\varepsilon$ be the mixture distribution defined by $P_\varepsilon = (1 - \varepsilon)P + \varepsilon Q$ for $\varepsilon \in [0, 1]$. Then, as $\varepsilon \to 0$:*

$$\text{KL}(P_\varepsilon \| P) = \varepsilon^2 \chi^2(Q \| P) + O(\varepsilon^3),$$

*where $\chi^2(Q \| P) = \frac{1}{2}\sum_{a \in \mathcal{A}} \frac{(Q(a) - P(a))^2}{P(a)}$ is the Chi-square divergence.*

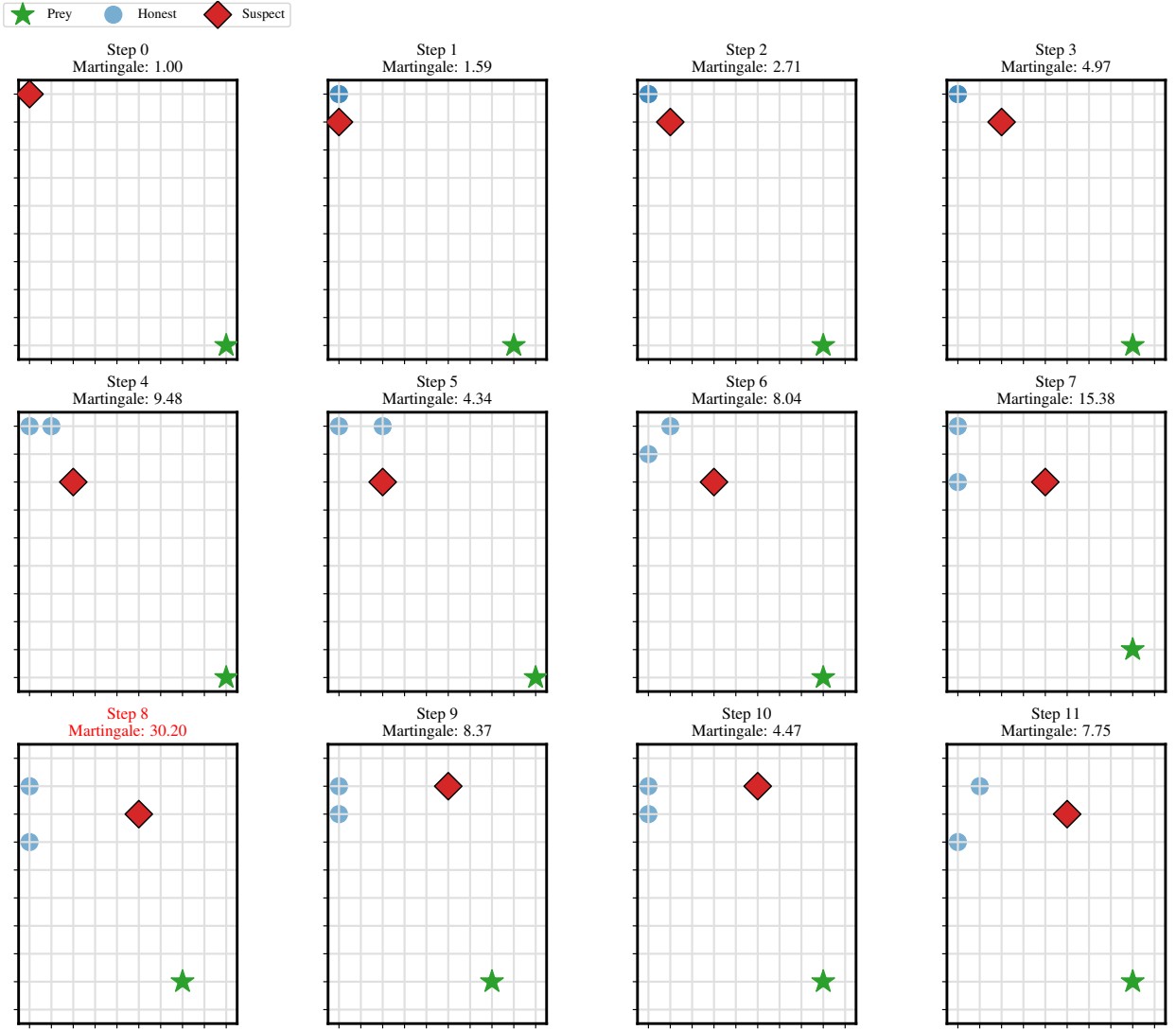

*Figure 11.* **Prey-Predator gameplay sample.** Film-strip visualization of the first 12 steps. The martingale value (shown above each frame) rises as the Suspect systematically pursues the Prey, triggering a detection (red title) when the value exceeds the threshold $b = 20$.

*Proof.* By definition, the KL divergence is given by:

$$\mathrm{KL}(P_\varepsilon \| P) = \sum_{a \in \mathcal{A}} P_\varepsilon(a) \log \left( \frac{P_\varepsilon(a)}{P(a)} \right).$$

Let $\delta(a) = Q(a) - P(a)$. We can rewrite the mixture probability as:

$$P_\varepsilon(a) = (1 - \varepsilon)P(a) + \varepsilon Q(a) = P(a) + \varepsilon(Q(a) - P(a)) = P(a) + \varepsilon\delta(a).$$

Substituting this into the log term:

$$\frac{P_\varepsilon(a)}{P(a)} = \frac{P(a) + \varepsilon\delta(a)}{P(a)} = 1 + \varepsilon\frac{\delta(a)}{P(a)}.$$

The KL divergence becomes:

$$\mathrm{KL}(P_\varepsilon \| P) = \sum_{a \in \mathcal{A}} (P(a) + \varepsilon\delta(a)) \log \left( 1 + \varepsilon\frac{\delta(a)}{P(a)} \right).$$

Using the Taylor expansion $\log(1 + x) = x - \frac{x^2}{2} + O(x^3)$ for small $x$ (which holds for small $\varepsilon$ since $\frac{\delta(a)}{P(a)}$ is bounded due to the full support assumption on $P$), we expand the log term:

$$\log\left(1 + \varepsilon\frac{\delta(a)}{P(a)}\right) = \varepsilon\frac{\delta(a)}{P(a)} - \frac{\varepsilon^2}{2}\left(\frac{\delta(a)}{P(a)}\right)^2 + O(\varepsilon^3).$$

Substituting this back into the sum:

$$\mathrm{KL}(P_\varepsilon\|P) = \sum_{a\in\mathcal{A}}[P(a) + \varepsilon\delta(a)]\left[\varepsilon\frac{\delta(a)}{P(a)} - \frac{\varepsilon^2}{2}\frac{\delta(a)^2}{P(a)^2} + O(\varepsilon^3)\right]$$

$$= \sum_{a\in\mathcal{A}}\left[\underbrace{P(a)\cdot\varepsilon\frac{\delta(a)}{P(a)}}_{\text{Term 1}} \underbrace{- P(a)\cdot\frac{\varepsilon^2}{2}\frac{\delta(a)^2}{P(a)^2} + \varepsilon\delta(a)\cdot\varepsilon\frac{\delta(a)}{P(a)}}_{\text{Term 2}} + O(\varepsilon^3)\right].$$

We analyze the terms individually. The first term is given by:

$$\sum_{a\in\mathcal{A}}\varepsilon\delta(a) = \varepsilon\sum_{a\in\mathcal{A}}(Q(a) - P(a)) = \varepsilon\left(\sum_a Q(a) - \sum_a P(a)\right) = \varepsilon(1 - 1) = 0.$$

The linear term vanishes, which is expected as KL is minimized at $\varepsilon = 0$.

Now, adding the second-order terms:

$$\sum_{a\in\mathcal{A}}\left(\varepsilon^2\frac{\delta(a)^2}{P(a)} - \frac{\varepsilon^2}{2}\frac{\delta(a)^2}{P(a)}\right) = \frac{\varepsilon^2}{2}\sum_{a\in\mathcal{A}}\frac{\delta(a)^2}{P(a)}.$$

Substituting $\delta(a) = Q(a) - P(a)$, we obtain:

$$\mathrm{KL}(P_\varepsilon\|P) = \frac{\varepsilon^2}{2}\sum_{a\in\mathcal{A}}\frac{(Q(a) - P(a))^2}{P(a)} + O(\varepsilon^3).$$

The summation term is precisely the Chi-square divergence $\chi^2(Q\|P)$. $\qquad\square$

