# OpenReview forum: "Anytime Detection of Strategic Deviations in Multi-Agent Systems"
_ICML.cc/2026/Conference — ICML 2026 regular_

### Official Review · Reviewer_b4kA · 2026-02-23

**Soundness:** 4
**Presentation:** 3
**Significance:** 3
**Originality:** 3
**Overall Recommendation:** 5
**Confidence:** 4

**Summary:**

The paper proposes a testing framework, for online monitoring, for multi-agent systems across two settings. First, in repeated normal-form games, it provides a strategy-independent test for Nash Equilibrium by monitoring regret-based e-values. Second, in infinite-horizon stochastic games, it tests adherence to a specific target policy using likelihood-ratio e-values. In both settings, the method constructs a test based on stochastic sequences forming supermartingales and martingales under the null hypothesis. The approach requires an observer who can compute deviating rewards for all possible deviations. The authors verify the established results with numerical experiments.

**Compliance With Llm Reviewing Policy:**

Affirmed.

**Key Questions For Authors:**

Title Alignment: Given that Section 3 focuses on monitoring adherence to a specific policy rather than testing the strategic properties of an equilibrium (as is done in Section 2), do the authors believe it would be appropriate to adapt the title of the article to better reflect the scope of the proposed work?

Equilibrium Monitoring in Stochastic Games: Would it be possible to propose an algorithm, perhaps based on a well-explained heuristic, to monitor deviations from equilibrium in stochastic games without requiring prior knowledge of the target policy? And to illustrate it with numerical experiments ?

Scalability and Empirical Validation: Could the authors demonstrate the scalability of the proposed method by evaluating it in larger environments with a significantly larger number of agents than the current two-player experiments?

**Limitations:**

yes

**Strengths And Weaknesses:**

Strengths:

Mathematical Rigor: The paper is technically good and highly rigorous. It is well-written, and the comprehensive appendices provide thorough proofs and additional context for the presented theorems.

Novel Online Framework for Normal-Form Games: The authors introduce a new framework for testing equilibrium in repeated normal-form games. The ability to monitor online these systems is a valuable contribution to the field.

Novel Online Framework for Stochastic Game Policy Consistency: It presents a new framework for testing coherence with a given policy in a stochastic game setting, answering the question to monitor potential deviation of agents for a given policy.

Weaknesses:

Misleading Title: The title does not fit the work, as the second problem addressed in the paper does not actually tackle the question of deviating from an equilibrium, but rather deviating from a given policy. To use the presented method for testing equilibrium in a stochastic game, the observer would need to know the equilibrium in advance. For stochastic games, the results do not utilize the specific properties of a Nash equilibrium.

Numerical Scalability Concerns: The proposed methods appear computationally and statistically difficult to scale as the number of agents increases. For even a medium-sized game, the monitor must compute and track all possible deviating actions for every agent. The paper would be strengthened by the inclusion of a more scalable algorithm, even a heuristic one without formal proofs, to demonstrate how this framework might be applied to many-agent systems.

Restrictive Information Requirements: It is assumed that the observer knows the reward functions, which may be very restrictive in practice. It would be interesting to know if it is possible to leverage the properties of a Nash equilibrium to develop a supermartingale that does not require direct knowledge of the rewards (perhaps using only observed past rewards), even without knowing the reward matrix.

Lack of Equilibrium Testing in Stochastic Games: There is no actual test provided for equilibrium in the stochastic game section.

Conclusion:

I recommend a Accept because the paper introduces a mathematically elegant and rigorous framework for online equilibrium monitoring using e-values, which is a timely and novel application of sequential testing. The online nature of the test is a strong contribution to the community. I am hesitant to give a higher score due to the disconnect between the title and the content of Section 3, as well as the restrictive assumptions regarding reward knowledge and the small number of agents used in the experiments.

---

> ### Author Rebuttal · Authors · 2026-03-30
>
> We thank the reviewer for their detailed and comprehensive review. We are happy to address the points they raised.
>
> **On scalability to larger systems:**
>
> This point was also raised by reviewer kSSF, and we refer the reviewer to our detailed response there. In short, we conducted an additional large-scale experiment and introduced a simple data-splitting heuristic: using the first $T_0$ steps to identify a small subset of promising supermartingales, and then running the testing procedure only on this subset. This approach both keeps the detection procedure computationally light and further improves the observed speedup.
>
> **On the title and applicability to equilibrium testing in stochastic games:**
>
> We agree that extending our framework to testing equilibria in stochastic games is an important direction. We outline a heuristic for monitoring deviations from equilibrium in stochastic games. Let
> $$V_i^{\pi}(s) = \mathbb{E}\left[\sum_{t=0}^{\infty} \gamma^t r_i(s_t,a_t) \mid s_0=s\right]$$
> be the value function, and $$Q_i^{\pi}(s,a) = r_i(s,a) + \gamma \mathbb{E}_{s'\sim P(. \mid s,a)}[V_i^{\pi}(s')]$$
>
> the action-value function. We define a Nash Equilibrium (Markov Perfect Equilibrium) as a policy $\pi^{\star}$ such that $$V_i^{\pi^{\star}}(s) \ge V_i^{(\pi_i', \pi^{\star}_{-i})}(s), \quad \forall \pi_i', \forall i, \forall s.$$
>
> Equivalently (omitting the details):
>
> $$V_i^{\pi^{\star}}(s) \ge Q_i^{\pi^{\star}}(s,a_i,\pi^{\star}_{-i}), \quad \forall a_i, \forall i, \forall s$$
>
> where $Q_i^{\pi^{\star}}(s,a_i,\pi_{-i}^{\star}) := \mathbb{E}\left[Q_i^{\pi^{\star}}(s,(a_i,a_{-i}))\right]$ and the expectation is taken over $a_{-i} \sim \pi_{-i}^\star(\cdot | s)$. In the setting where $\pi^{\star}$ is unknown and we observe a trajectory $(s_t,a_t,r_t)$, a natural approach is to construct a plug-in sequential test based on empirical regret. Let $\hat \pi_{t-1}$, $\hat Q_{i,t-1}$, and $\hat V_{i,t-1}$ be predictable estimators. We can define a variable
> $$
> X_{i,t}^{(a_i)}=\hat Q_{i,t−1}(s_t,a_i, \hat \pi_{−i,t−1})- (r_{i,t}+\gamma \hat V_{i,t-1}(s_{t+1})),
> $$
> and then a process for each player $i$ and potential deviation $a_i$:
> $$
> E_{i,t}^{(a_i)}=E_{i,t-1}^{(a_i)}(1+\lambda_{i,t}X_{i,t}^{(a_i)}).
> $$
> Under a true equilibrium and exact value functions, $(E_{i,t})_t$ would form a supermartingale. However, with plug-in estimators, finite-sample bias may appear, and the resulting procedure is not guaranteed to be anytime-valid. It should therefore be viewed as an asymptotically valid heuristic under some consistency assumptions.
>
> While promising, this approach is more involved and would require a dedicated treatment. We therefore believe it is preferable to leave a full development to future work, although we would be happy to include a brief discussion in the paper. Following the reviewer’s suggestion, we also agree that it would be more accurate to adjust the title to emphasize that our focus is more broadly on monitoring strategic behavior in multi-agent systems (rather than equilibrium).
>
> **On the assumptions regarding reward knowledge:**
>
> This point was also raised by reviewer WvdG, and we refer to our response there. To summarize, these assumptions are standard in the game theory literature. Importantly, compared to prior work, our methods require strictly less information: depending on the setting, we only assume access to either the reward function or the equilibrium policy, but not both.
>
> We thank the reviewer again for their detailed feedback and for their positive evaluation of our paper. We hope we have addressed their questions and remain available for any further clarifications.

---

> > ### Author Rebuttal · Reviewer_b4kA · 2026-03-31
> >
> > I thank the authors for the answers. If the authors indeed add a larger experiment and adapt the title of their article, I am happy to confirm my grade of 5. Thank you very much

---

### Official Review · Reviewer_kSSF · 2026-03-05

**Soundness:** 2
**Presentation:** 3
**Significance:** 3
**Originality:** 3
**Overall Recommendation:** 4
**Confidence:** 2

**Summary:**

This paper proposes an online sequential testing framework based on e-values to monitor whether behaviors in multi-agent systems deviate from game-theoretic equilibria. Unlike traditional offline tests that rely on fixed sample sizes, this approach allows for anytime-valid inference. In Normal-Form Games, the authors construct test supermartingales based on counterfactual payoffs to accumulate evidence of profitable deviations, and they incorporate the e-BH procedure to control both the Family-Wise Error Rate (FWER) and the False Discovery Rate (FDR). Furthermore, the paper attempts to extend this framework to Stochastic Games via likelihood ratios and provides theoretical bounds on the expected detection time.

**Compliance With Llm Reviewing Policy:**

Affirmed.

**Final Justification:**

This paper provides an excellent theoretical framework for focusing on strategic behavior in multi-agent systems. Combined with the author's newly added experiment (although this experiment may still be relatively small compared to the scale of online games, I believe it at least demonstrates the efficiency of the method in larger game scenarios), I think this paper could be accepted.

**Key Questions For Authors:**

1. Why was the mixture methodology abandoned in evaluating the FWER and FDR procedures?

2. Can you provide experimental results for a genuinely large normal-form game comparing the detection delay of FWER versus FDR?

I am not familiar with this field or the corresponding theories, and I am willing to discuss with the authors and peer reviewers during the rebuttal phase. If the authors can resolve my doubts, I am willing to raise my score.

**Limitations:**

yes

**Strengths And Weaknesses:**

Strengths:
1. Introducing e-values and anytime-valid inference to equilibrium monitoring in multi-agent systems is a highly creative and novel perspective. It provides an elegant statistical tool for online monitoring without requiring a predetermined sample size.

2. The construction of supermartingales based on counterfactual payoffs is mathematically rigorous. It unifies the testing of various equilibria and provides a clear theoretical foundation for FWER and FDR control.

Weaknesses:
1. The authors identify that fixing a single parameter $\lambda$ leads to a loss of detection power, and thus propose using a Mixture of e-values in Corollary 2.5 to resolve this. However, the core FWER and FDR experiments completely abandon the proposed Mixture mechanism. Instead, they hardcode a highly conservative $\lambda = 0.05$.

2. The authors repeatedly claim that the e-BH procedure for FDR control is introduced to address the strictness of FWER and boost power in "large games". However, the empirical validation is exclusively conducted on a trivial $2 \times 2$ toy game with only 4 deviation hypotheses. In this minimal setting, the FDR procedure yields only a marginal 1.15x speedup over FWER. Demonstrating a minor speedup in a micro-scale game fails to substantiate the claim that the method significantly improves detection power in large games, rendering the practical significance claims unsupported.
3. The core premise of the paper is "Betting on Equilibrium," which inherently relies on incentive compatibility. While Section 2 correctly uses payoff differences to monitor profitable deviations, Section 3 completely abandons payoffs to ensure mathematical tractability. Instead, it relies purely on the action likelihood ratio to monitor behavior. Consequently, the framework degrades from a game-theoretic equilibrium test to a standard policy compliance or out-of-distribution detection test. If an agent switches to an alternative action that yields the same expected return, it remains within a Nash equilibrium, yet the likelihood-based test in Section 3 would falsely flag it as a deviation. This fundamentally undermines the conceptual soundness of the stochastic game formulation.

---

> ### Author Rebuttal · Authors · 2026-03-27
>
> We thank the reviewer for their insightful comments. Despite not being fully familiar with the field, their feedback is highly relevant, and we are happy to address their questions.
>
> **Regarding the use of a Dirac distribution with $\lambda = 0.05$:**
>
> This choice was primarily made for brevity, as we wanted to keep the appendix not too long and focus on other aspects we deemed more central. We expect similar results for other values of $\lambda$, as well as for mixture-based approaches, which typically do not provide substantially different insights in this setting. Indeed, in this line of work, even a reasonably chosen fixed $\lambda$ (without careful tuning) often achieves empirical performance comparable to that of mixture-based methods.
>
> That said, we agree that including such experiments would improve completeness, and we would be happy to add results with mixtures in the final version.
>
> **Regarding experiments on larger systems:**
>
> We conducted a new experiment on a larger system (20 players, 3 actions each, yielding 60 deviation hypotheses), where the benefits of our approach become even more apparent. Computationally, the method remains efficient when combined with a data-splitting strategy: we use the first $T_0$ steps to identify the supermartingales with the most potential, then apply our FWER or FDR procedures after $T_0$​ steps only on the selected subset of promising supermartingales. For this experiment, we used a simple, unoptimized heuristic of $T_0=50$ steps to select a subset of just 10 supermartingales. Even without tuning these parameters, the empirical results over 200 independent runs show that this splitting strategy keeps the detection system computationally light while drastically reducing the time to detection:
> - *Split-FDR vs. FWER*: Split-FDR detects deviations 2.54x faster on average than the standard FWER baseline (with maximum speedups reaching 3.22x).
> - *Split-FDR vs. Standard FDR*: Split-FDR detects deviations 1.56x faster on average.
>
> We will include this extended experiment and the details of the splitting heuristic in a new section in the appendix.
>
> **Regarding the comment on Section 3 testing a policy rather than an equilibrium:**
>
> This point was also raised by reviewer b4kA, and we refer the reviewer to the corresponding detailed response. Briefly, while we propose a heuristic approach for testing equilibrium, it is more involved and would likely deserve a dedicated treatment. Following the suggestion of reviewer b4kA, we therefore plan to revise the title to better reflect the scope of the paper and focus on the likelihood-ratio-based construction. We leave the equilibrium-testing heuristic as a direction for future work.
>
> We thank the reviewer again for their feedback and hope we have satisfactorily addressed their questions.

---

> > ### Author Rebuttal · Reviewer_kSSF · 2026-04-01
> >
> > I consulted feedback from other reviewers and the authors' rebuttal. I believe that if the authors revised the scope of the topic (i.e., emphasizing the primary focus on strategic behavior (rather than equilibrium) in multi-agent systems) and addressed the questions raised by other reviewers, the paper would reach an acceptable level. Therefore, I raised my score.

---

### Official Review · Reviewer_WvdG · 2026-03-08

**Soundness:** 3
**Presentation:** 3
**Significance:** 3
**Originality:** 3
**Overall Recommendation:** 4
**Confidence:** 4

**Summary:**

The authors consider in my opinion a novel application of sequential testing (safe anytime-valid inference via e-processes) to testing the null hypothesis of Nash equilibrium (and other notions of equilibrium considered in the appendix). This translates to the null hypothesis that no player can increase their expected payoff by unilaterally deviating (where an individual player changes their chosen strategy while all other agents in the game keep their behavior exactly the same). The authors define the following outcome

$X_{i,t}^{a_i'} = u(a_{i,t}, a_{-i,t}) - u(a_i', a_{-i,t})$

where $u$ is a utility/payoff function, the vector $a_t$ are the actions taken by players at time $t$, and $a_i'$ is an action that player $i$ could have played. In Nash equilibrium, the expected value of this outcome is positive, as the player cannot increase their expected payoff by a strategy that is a pure play (constantly taking the static action a_i' at each round).

The authors construct an e-process using these outcomes using standard techniques: $E_{i,t}=1-\lambda X_{i,t}$ is standard for testing the means of bounded outcome RVs.

In subsequent sections the authors switch from normal-form games to stochastic games, dropping the $E_{i,t}$ e-process in favour of a likelihood ratio test-martingale..

**Compliance With Llm Reviewing Policy:**

Affirmed.

**Final Justification:**

There was a mistake in the authors original submission. After identifying it, the authors have identified how to reframe their procedure for correctness, I am satisfied with their revision and found this paper interesting both from methodological and application perspectives.

**Key Questions For Authors:**

At first I thought that testing against a constant static strategy is restrictive, why not test against an alternative strategy \pi_i', instead of a constant strategy (repeatedly taking action a_i'). I've convinced myself that testing against all possible constant strategies is sufficient to tests against *any* mixed strategy \pi_i'. It might help to note this in the paper if it isn't addressed already.

It might be helpful to remind the reader that $X_{it}$ is a bounded outcome, and that the e-process you consider in the first half is explicitly for testing the means of bounded outcomes.

In the first half of the paper, the authors test the null equilibria hypothesis by considering static constant strategies (playing $a_i'$ each round). Although this is sufficient, is it optimal? This is a degenerate strategy which essentially puts probability 1 on a single action, so we must consider many such alternatives for each possible action. Would it not be more powerful to adaptively learn an alternative strategy $\pi_i'$, instead of testing $|A_i|$ fixed degenerate alternative strategies + a multiple testing correction? This is essentially what you propose in equation 16 in the context of stochastic games. In which case, why not apply it to normal-form games also?

**Limitations:**

The framework assumes the payoff function $u(a_i', a_{-i,t})$ is known for every action of player $i$ and all other actions of remaining players $-i$. In practical settings, $u$ needs to be learned.

Similarly for stochastic games, the null equilibrium strategy $\pi$ is assumed known. Computing a true Nash equilibrium in complex, real-world stochastic environments is computationally intractable.

The math relies on the monitor perfectly observing the current state $s_t$ and the exact joint action profile $a_t$ of all players at every single time step. In practice, however, states are often hidden and actions are observed with noise.

For the normal-form game martingales to work, the null hypothesis strictly assumes that players draw their actions identically and independently from a fixed strategy at every round. The authors note in their own introduction that learning agents update their objectives and face changing environments, making the stationarity assumptions underlying equilibrium concepts "unrealistic".

**Strengths And Weaknesses:**

Soundness: [happy to increase my score if the authors can address my following concern]

 I have serious concerns regarding the FDR claim. Regarding Algorithm 2: You use Procedure 2 (Running maximum e-BH procedure) from Tavyrikov et al. (2025), but this result only guarantees FDR control for independent e-processes IIUC. Proposition 1 of  Tavyrikov et al. (2025)  states: "The running maximum e-BH procedure applied to arbitrarily dependent e-processes does not provide FDR control at level α.". You are effectively working with the running maximum, because you are counting the number of process that have "ever" crossed the rejection threshold. Perhaps I have misunderstood, but are your e-processes not dependent? If so, you would need the additional "Adjuster" of Tavyrikov et al. (2025) in the later sections. Looking at your proof of Theorem C.8, I think the issue is the conditional application of Ville's inequality. In the proof, they write
$$\mathbb{P}\left(
\exists s \le \tau :
M_{i,s}^{(a_i')}(\nu)
\ge \frac{1}{k_\tau \alpha \gamma_{i,a_i'}}
\,\middle|\, k_\tau, H_0
\right)
\le k_\tau \alpha \gamma_{i,a_i'}.
$$

I don't think this step is justified. Ville's inequality applies to a null e-process with a fixed threshold $c$, giving
$$
\mathbb{P}\left( \sup_{s \ge 0} M_s \ge c \right) \le \frac{1}{c}.
$$
But here the threshold
$
\frac{1}{k_\tau \alpha \gamma_{i,a_i'}}
$
is random, since $k_\tau$ is itself determined by the collection of threshold-crossing events across all hypotheses. In particular, $k_\tau$ is a function of the same running suprema that define the rejection set. Therefore one cannot simply condition on $k_\tau$ and apply Ville's inequality as though the threshold were exogenous.

This is exactly where the dependence problem enters. The argument would be valid if $k_\tau$ were independent of the null e-process under consideration, but in Algorithm 2 it is not: all the e-processes are constructed from the same observed play sequence, and $k_\tau$ is chosen from their joint threshold crossings. Thus the proof implicitly treats the running supremum like an ordinary stopped e-value, which is the step that fails.

Presentation:
The paper is written very well. My only issue is that the main body is the paper introduces things in abstract terms, while a lot of the interesting meat is in the appendix! I realize this is difficult given the 8-page limit.

Significance:
I think this paper develops little by way of safe anytime-valid inference machinery. What is novel though is the application to testing equilibria in games. Coming from the former, with little exposure to the latter, this paper was an enjoyable entry point to studying equilibria in games. This is actually very much on-brand for safe anytime-valid inference, as it is often positioned as game-theoretic statistics.

---

> ### Author Rebuttal · Authors · 2026-03-26
>
> We thank the reviewer for their very clear and detailed review, and especially for their careful reading, which allowed them to identify an important and indeed obvious mistake in our paper.
>
> **Regarding the FDR claim:**
>
> The reviewer is absolutely correct, and we are very grateful for pointing out this issue. We will revise this part and instead rely on the standard stopped e-BH procedure, without using the running supremum or the adjusters proposed in Tavyrikov et al. [2025].
>
> Importantly, we reran the experiments from the paper (normal-form games setting) using the corrected procedure, and obtained an expected detection time of $1546.3 \pm 212.3$, which is essentially identical to the previously reported $1545.4\pm 212.8$. This negligible difference suggests that, in our setting, the practical impact of the issue is minimal. Intuitively, e-processes that grow enough to cross the e-BH threshold are typically associated with true deviations and are unlikely to decrease significantly afterward. As a result, the difference between using the running supremum and the standard stopped e-values is empirically negligible in our experiments.
>
> We also experimented with Adjusters 1 and 2 from Tavyrikov et al. [2025], but observed substantially larger expected stopping times ($3774.2 \pm 220.3$ and $3198.6 \pm 312.9$, respectively). For this reason, we believe that using the standard stopped e-BH procedure is the most appropriate and practical choice.
>
> **On testing against fixed actions instead of mixed strategies:**
>
> This is a standard and well-known fact in game theory, which underlies the definition of Nash equilibrium. Because the utility is linear in the mixed strategy, if a mixed strategy is profitable, then there necessarily exists a pure action $ a_i' $ that is at least as profitable. This follows from the fact that the set of mixed strategies forms a probability simplex, and a linear function achieves its maximum at an extreme point. As this is classical and typically implicit in the literature, we did not elaborate on it, but we are happy to add a clarification.
>
> **On the choice of the e-process:**
>
> The e-process we use is indeed explicitly for testing the means of bounded outcomes, and we will add a sentence emphasizing this point immediately after its definition.
>
> **Regarding the use of adaptive or learned alternatives:**
>
> We have also considered likelihood-ratio-based constructions instead of coin-betting e-values, and provide the corresponding derivation in Appendix G. While such approaches can be optimal when the true alternative is known, they require knowledge of the equilibrium strategy $\pi$, which is not assumed in the main body. When the true alternative is unknown, there are two natural approaches: one can either learn an alternative strategy via plug-in methods (as suggested by the reviewer) or use mixtures, as is also standard in the literature and in our work. For brevity, we focused on the mixture approach in the paper, but we would be happy to expand the discussion and include additional experiments comparing both approaches in the final version.
>
> **Regarding the limitations:**
>
> We would like to emphasize that these limitations are largely inherent to the game theory literature, where it is typically assumed that the payoff function $u$ or the equilibrium strategy $\pi$ is known, and that actions are i.i.d. For example, prior work such as Babichenko et al. [2017] (which studies testing Nash equilibria in a batch setting) assumes i.i.d. data and knowledge of both $u$ and $\pi$. In contrast, our method in the main body requires only knowledge of $u$, while the approach in Appendix G requires only knowledge of $\pi$. In this sense, our methods improve over existing approaches in terms of the information required (see Remark 2.10).
>
> We thank the reviewer again for identifying the issue regarding the FDR claim, and we are reassured to see that it has minimal practical impact, as the corrected stopped e-BH procedure performs almost identically in our experiments. We will update the final version accordingly and are grateful for this valuable feedback. We hope we have addressed the reviewer’s concerns and would be happy to provide further clarifications if needed.

---

> > ### Author Rebuttal · Reviewer_WvdG · 2026-04-01
> >
> > Thanks you for partially addressing my concerns. I am still a little confused about your subsequent FDR claim. The standard e-BH procedure applies only to *stopped* e-values, yet it sounds like you are still using the e-BH machinery to define the stopping rule, which is my concern - I don't think this is permitted either, can you please provide formal justification or citation. Procedure 1 of Tavyrikov et al. [2025] assumes that the e-processes have already been stopped (via some other stopping rule). To be clear, this is the primary hurdle for me to switch from weak reject to weak accept. Can you please clarify very formally what your stopping rule is and the claims you are making about the e-values at that stopping time, and the formal proof this controls FDR? Please distinguish: are all e-processes (across all hypotheses) stopped simultaneously? Or do you allow each e-process to have it's own stopping time? Please define the stopping rule.
> >
> > Moreover, if we stop the entire experiment at the first time a single e-process crosses a threshold, then aren't we always just going to have a single rejection? I can't imagine a scenario where at t=n-1 we have no rejections, but at t=n we have several rejections.

---

> > > ### Author Response · Authors · 2026-04-02
> > >
> > > We thank the reviewer for their prompt follow-up and for highlighting this point. The reviewer is completely correct that applying the e-BH procedure to sequentially stopped e-processes requires extreme care, as the original e-BH procedure was designed for static e-values.
> > >
> > > To ensure there is no ambiguity, we provide a formal statement and proof:
> > >
> > >
> > > **Statement:** Let $(\Omega, \mathcal{F}, \lbrace \mathcal{F}_t \rbrace_t,\mathbb{P})$ be the filtered proability space considered in the paper. Let $\mathcal{H} = \lbrace H_1,...,H_m \rbrace$ be a set of null hypotheses, with $\mathcal{H}_0 \subseteq \mathcal{H}$ denoting the subset of true nulls.
> > >
> > >
> > > For each $i \in \lbrace 1,...,m \rbrace$, let $M_i = (M_{i,t})_t$ be an $\mathcal{F}_t$-adapted process.
> > >
> > >
> > > Assume that for every true null $i \in \mathcal{H}_0$:
> > >
> > >
> > > $M_i$ is a non-negative supermartingale (wrt the filtration) with $M_{i,0} \le 1$.
> > >
> > >
> > > Let $\gamma_1,...,\gamma_m$ be fixed, non-negative weights summing to 1.
> > >
> > >
> > > At any time $t\ge0$, the e-BH procedure evaluates $(M_{1,t},...,M_{m,t})$ and returns a rejection set $R_t \subseteq \mathcal{H}$ defined by:
> > >
> > >
> > > $k_t = \max \lbrace k \in \lbrace 1,...,m\rbrace : \sum_{j} 1\lbrace M_{j,t} \ge 1/(k\alpha\gamma_j) \rbrace  \ge k \rbrace$,
> > >
> > >
> > > $R_t = \lbrace i \in \lbrace 1,..., m \rbrace : M_{i,t} \ge 1/(k_t\alpha\gamma_i) \rbrace$.
> > >
> > >
> > > We define the **global** stopping time $\tau$ as the first round the e-BH procedure rejects at least one hypothesis:
> > >
> > >
> > > $\tau = \inf\lbrace t\ge0 : |R_t|>0 \rbrace.$
> > >
> > >
> > > If we stop the experiment at $\tau$ and reject the set $R_\tau$, the FDR is controlled at level $\alpha$:
> > >
> > >
> > > $FDR = \mathbb{E}\left[ \frac{|R_\tau \cap \mathcal{H}_0|}{|R(\tau)| \vee 1} \right] \le \alpha$.
> > >
> > >
> > > (We sometimes deliberately write $R(\tau)$ instead of $R_\tau$ due to LaTeX rendering issues in Markdown. Similarly, we may write $M_i(\tau)$ instead of $M_{i,\tau}$.)
> > >
> > >
> > > **Proof:** First of all, it is clear that $\lbrace \tau \le t \rbrace \in \mathcal{F}_t$, so $\tau$ is a valid stopping time.
> > >
> > > For clarity of exposition, we first present the proof under the assumption that the experiment eventually stops ($\tau < \infty$ almost surely), and then relax this to the fully general case.
> > >
> > > By the Optional Stopping Theorem we know that $\mathbb{E}[M_{i,\tau}] \le \mathbb{E}[M_{i,0}] \le 1$ for all $i \in \mathcal{H}_0$.
> > >
> > >
> > > We can now bound the FDR. We have $|R_\tau| = k_\tau \ge 1$. The e-BH condition ensures that for every $i \in R_\tau$, we have:
> > >
> > >
> > > $M_{i,\tau} \ge 1/(|R_\tau| \alpha \gamma_i)$ so $1/|R_\tau| \le \alpha \gamma_i M_{i,\tau}$.
> > >
> > >
> > > Then:
> > >
> > >
> > > $FDR = \mathbb{E}\left[ \sum_{i \in \mathcal{H}_0} \frac{1 \lbrace i \in R(\tau) \rbrace}{|R\tau|} \right]$, so:
> > >
> > >
> > > $FDR \le \mathbb{E}\left[ \sum_{i \in \mathcal{H}_0} M_i(\tau) \alpha \gamma_i 1 \lbrace i \in R(\tau) \rbrace \right] $
> > >
> > >
> > > Since the indicator is upper bonded by one, and applying the fact that $\mathbb{E}[M_{i,\tau}] \le 1$ for all $i \in \mathcal{H}_0$, we deduce that:
> > >
> > >
> > > $FDR \le \alpha \sum_{i \in \mathcal{H}_0} \gamma_i$
> > >
> > >
> > > $FDR  \le  \alpha \sum_{i=1}^m \gamma_i = \alpha$.
> > >
> > > This concludes the proof. In the general case, the proof is similar; we can apply the OST to $\tau \wedge t$ for any fixed $t \ge 0$, and obtain that:
> > >
> > > $\mathbb{E}[M_{i,\tau} 1\lbrace \tau < \infty \rbrace ] \le 1$ for all $i \in \mathcal{H}_0$ by the Monotone Convergence Theorem.
> > >
> > > Substituting this into the FDR calculation yields $FDR \le \alpha$.
> > >
> > > We hope this clarifies our approach. Of course, this may lead to multiple hypotheses being rejected simultaneously (see, for example, panel (a) of Figure 5 in Appendix H). We thank the reviewer again for their engagement and hope we have adequately addressed their question.

---

### Official Review · Reviewer_yMi7 · 2026-03-18

**Soundness:** 4
**Presentation:** 3
**Significance:** 3
**Originality:** 3
**Overall Recommendation:** 5
**Confidence:** 3

**Summary:**

The present paper considers the problem of detecting deviations from expected strategic play in normal-form games and stochastic games. The authors employ the e-values paradigm to enable monitoring in an online fashion. They propose different formulations of e-values to detect deviations with corresponding theoretical worst-case analysis of the stopping times under suitable $\mathcal{H}_1$ hypotheses. They demonstrate empirical applicability in different exemplary settings and show that it matches their theoretical results.

**Compliance With Llm Reviewing Policy:**

Affirmed.

**Final Justification:**

While conceptually not very novel, I think this is a solid contribution to the field and, therefore, vote for acceptance. (Further clarification of reviewer WvdG's concerns pending.)

**Key Questions For Authors:**

Did you perform experiments with a larger variety of the mixture over candidate strategies? Or even one where the perfect deviation strategy is not contained exactly. How does that impact your results?

Please expand on your choice of the linear e-values. Why are these superior to other choices?

**Limitations:**

yes

**Strengths And Weaknesses:**

The paper considers an important and timely problem of detecting deviations from expected or desired play. Monitoring this in an online fashion is becoming increasingly relevant due to the widespread deployment of automated algorithms designed for single-agent problems in game-theoretic environments. Whether these systems are stable and remain close to a desired play is unclear.
The authors leverage the paradigm of e-values to enable monitoring in an online fashion. The results presented seem solid. The paper is well-written and easy to follow. Despite no code being published, I am confident that the paper contains all relevant information to reproduce the presented results.
Nevertheless, I have some concerns regarding this work, which I will explain in more detail below. In particular, the overall novelty and impact of the work appear to be limited. Overall, I see this as solid work with clear contribution but limited impact.

## Comments

1. The applied statistical tools (e-values, mixture martingales, etc.) are standard. The conceptual novelty lies primarily in their application to monitoring deviations from equilibrium. While this is a natural application and well-executed, the conceptual contribution is limited. Furthermore, while this holds value as it fills a gap in the literature and the problem of online monitoring is relevant, the paper misses out to addressing several limitations. Some of which are acknowledged, e.g., the high computational costs, perfect knowledge of target (and deviation) policy or utility functions, and only testing for stationary deviations. Practical considerations, e.g., how to choose the mixture distribution $\nu$ (Algorithm 1) or what happens if we select the wrong set of plausible deviations mentioned in Remark 3.5, are missing. I think adding a discussion or acknowledgement about this would be helpful.

2. The specific choice of linear e-values ($1 - \lambda X_t$) seems natural but raise the question about high variance and stability. The comment about “optimal in the sense of Clerico (2024; 2025); Larsson et al. (2025b)” on page 3 is unclear to me. A discussion comparing alternative e-process constructions (e.g., likelihood-ratio or exponential tilting) would strengthen the contribution.

3. The empirical experiments are designed to verify the theoretical results but give little insight into the methodologies limitations. One example would be to explore what happens if the mixture over candidate strategies is not estimated as well as for Figure 3 (see page 36 for the made choices here).

4. The relevance for real-world settings are limited. The limitations, some of which mentioned above, are prohibitive in several ways for larger systems. Furthermore, agents are deploying learning algorithms in practical systems that underlie fluctuations that do not impact overall performance. Therefore, testing against a fixed policy is too strict.

---

> ### Author Rebuttal · Authors · 2026-03-26
>
> We would like to thank the reviewer for their clear and structured feedback, as well as for the positive evaluation of our work. We would like to address the comments raised, particularly regarding the potential impact of our work.
>
> **Regarding the case when the mixture does not contain the true alternative:**
>
> As noted in Remark 3.5, our paper explicitly details what happens in this situation. This is further elaborated in Appendix F, in particular Theorems F.5 and F.7. Theorem F.7 provides a precise upper bound on the expected detection time in the general case, which depends, among other factors, on how much mass the mixture assigns to balls containing the true alternative. We also conducted an experiment in the main body of the paper showing that, even when the mixture does not include the true alternative, the detection rate remains practically optimal (see Figure 3). In this predator-prey experiment, the mixture does not necessarily contain the true parameter (see Section H.2.3 in the appendix for experimental details). Due to space constraints, we did not provide extensive details on this experiment in the main body, but it indeed shows that the detection time still scales optimally even when the mixture does not contain the true alternative, and we would be happy to emphasize this point in the final version of the paper.
>
> **On the choice of the e-value:**
>
> It is true, as the reviewer notes, that other forms (such as exponential tilting or likelihood ratios) are also valid. We chose $1-\lambda X$ as a proof of concept, but our method works equally well with other e-values. Each has its pros and cons; for example, using a likelihood ratio would require knowledge of the alternative and the Nash equilibrium $\pi$. We have detailed this choice of e-value in Appendix G. We can also add an appendix on exponential tilting if the reviewer finds it useful, or alternatively, include a remark clarifying that the method works with any supermartingale and highlight the most common alternatives.
>
> **On scaling to larger systems:**
>
> This point was also raised by reviewer kSSF, and we refer the reviewer to our detailed response there. In short, we conducted an additional large-scale experiment that shows a more pronounced speedup. Moreover, we introduce a simple data-splitting heuristic: using the first $T_0$ steps to identify a small subset of promising supermartingales, and then running the testing procedure only on this subset. This approach both keeps the detection procedure computationally light and further improves the observed speedup.
>
> We thank the reviewer again for their time and detailed comments. We would also like to note that our code is available, included as supplementary material. We remain at the reviewer’s disposal for any further questions and hope we have clarified the significance and scope of our work.

---

> > ### Author Rebuttal · Reviewer_yMi7 · 2026-04-02
> >
> > The authors adequately addressed my comments.
> >
> > I think it would be beneficial to more actively discuss the choice of linear e-values and how and to what extend others can be used as well.
> >
> > Including the larger experiments will also strengthen the paper.
> >
> > The discussion with reviewer WvdG seems to be very relevant. However, my concerns have been addressed and, therefore, I will keep my score.

---

### Decision · Program_Chairs · 2026-04-30

**Decision:**

Accept (regular)

**Comment:**

All reviewers find the paper provides a meaningful contribution.  Through detailed responses, the authors have resolved reviewer concern, given the authors implement the promised changes.  In the words of one reviewer, the provides an "excellent theoretical framework for focusing on strategic behavior in multi-agent systems."  As such, I recommend the paper be accepted.